# BALANCING MIXED LABELS: MIXUP MEETS NEURAL COLLAPSE IN IMBALANCED LEARNING

## ABSTRACT

*Minority collapse*, where minority classes become indistinguishable, is a significant challenge in imbalanced learning, which is addressed by methods such as Mixup with class-balanced sampling. The minority collapse has been mathematically analyzed using the layer-peeled model (LPM), together with the phenomenon of Neural Collapse (NC). Although the LPM has been employed to study NC behavior under Mixup, no prior work has analyzed minority collapse of Mixup, particularly from the perspective of mixed labels. We investigate this overlooked factor and pose the question: *Is the mixed label balance important for alleviating minority collapse?* Our analysis reveals that (i) mixed labels should be balanced, and (ii) in this setting, interpreting mixed labels as singletons is beneficial. Building on the analysis, we propose a *Balanced Mixed Label Sampler* and a *Mixed-Singleton classifier*, which balance mixed labels and treat them as singleton labels. Through theoretical analysis, visualization, and ablation studies, we demonstrate the effectiveness of our approach. Experiments on standard benchmarks further confirm consistent performance gains, highlighting the importance of balancing mixed labels in imbalanced learning.

## 1 INTRODUCTION

In imbalanced learning, severe class imbalance often causes a significant degradation of model accuracy, particularly on the minority classes (Liu et al., 2019). One known cause of this performance drop is the phenomenon termed *minority collapse* (Fang et al., 2021), wherein the class vectors of minority classes converge and become nearly identical. To mitigate this issue, a wide range of strategies has been explored, including data augmentation (Zhang et al., 2018; Verma et al., 2019; Shi et al., 2023), calibration technique (Zhong et al., 2021), mixture-of-experts models (Cai et al., 2021; Zhang et al., 2021; Xiang et al., 2020), and class-balanced loss functions (Cao et al., 2019; Cui et al., 2019) or sampling schemes (Kang et al., 2020; Cao et al., 2019; Zhang et al., 2022; Shen & Lin, 2016). Among these approaches, Mixup (Zhang et al., 2018), especially when combined with class-balanced sampling, has been shown to effectively improve the model performance under class-imbalanced conditions.

Meanwhile, Neural Collapse (NC) (Papyan et al., 2020) has emerged as a key framework for analyzing geometric properties of last-layer features and classifier in classification models at the terminal phase of training. Although NC has been studied in both Mixup (Fisher et al., 2024) and imbalanced learning (Liu et al., 2023; Yang et al., 2022) separately, Mixup in imbalanced settings has not been investigated in conjunction with NC. In particular, the balance of mixed labels has received little attention. The only related finding comes from M-lab NC (Li et al., 2024), which observes that even when multi-label samples are imbalanced, NC occurs at the singleton-class level as long as singleton label samples are balanced, with multi-label class emerging as combinations of singletons. However, whether the balance of input samples still hold for mixed labels under Mixup remains unclear. This motivates our central research question: *Could the balance of mixed labels be a critical factor in minority collapse?*

Building on the proof approach of Fang et al. (2021), we first demonstrate that *minority collapse still occurs under Mixup when the frequency of mixed labels are not balanced* (Theorem 1). Although existing class-balanced samplers partially alleviate the minority collapse of Mixup by balancing the frequency of singleton labels, they fail to address it entirely due to the randomness of Mixup. To

Figure 1: Overview of Balanced Mixed Label Sampler (BMLS) and Mixed-Singleton Classifier (MS)

obtain empirical evidence for this failure, we examined the per-label frequency generated in each epoch and observed an *epoch-wise label imbalance* phenomenon (Figure 2). Furthermore, through a mixed-label frequency control experiment (Figure 3), we empirically verified that this imbalance has a substantial impact on weakening the mitigation of minority collapse under Mixup. To address this issue, we propose **B**alanced **M**ixed **L**abel **S**ampler that balances the frequency of mixed labels across epochs (§3). Both theoretically and empirically, we demonstrate that aligning the frequency of mixed labels across epochs mitigates the minority collapse (Proposition 1 and Figure 4). Furthermore, our analysis uncovers that the minority collapse of Mixup is determined solely by the frequency of singleton and mixed labels, independent of the mixup ratio. Leveraging this insight, we introduce **M**ixed-**S**ingleton classifier, which treats mixed labels as singleton labels when learning class vectors (§3). Compared with a conventional singleton classifier implemented as a fully connected layer, our approach achieves superior performance, particularly improving accuracy on minority classes (Table 1).

## 2 RELATED WORK

In this section, we primarily discuss the novelty of our work. Additional related work that is not mentioned here or requires further detail can be found in Appendix A.

**Mixup-based Method.** Many attempts have been made to address the challenges of imbalanced learning environments using Mixup (Zhang et al., 2018), which increases the diversity of sampled data and alleviates risk of overfitting on tail classes, including data augmentation, architecture improvements, and calibration methods. (See more references in Appendix A.1.) However, no research has specifically studied on the frequency balance of mixed labels in minority collapse.

**Class-balanced Methods.** Various class-balanced samplers have been proposed (see more references in Appendix A.2), yet no work has mainly focused on the frequency balance of mixed labels. Additionally, while Logit Adjustment (Menon et al., 2021) and UniMix (Xu et al., 2021) have concentrated on the effect of the class vectors of singleton labels, they did not interpret mixed labels as singletons.

**Neural Collapse in Mixup and Imbalanced Learning.** NC in imbalanced learning has been studied in Fang et al. (2021). To alleviate the minority collapse, Yang et al. (2022) assumed that the classifier is fixed to the K-simplex ETF and proved that LPM with the classifier satisfies NC properties. Also, the fixed ETF classifier with Mixup has improved the model performance in imbalanced learning. Building on the theorems, Fisher et al. (2024) proved Mixup also satisfies NC properties for both same class and different class. However, Yang et al. (2022) and Fisher et al. (2024) did not consider the minority collapse from the frequency of mixed labels in the LPM with learnable classifiers.

## 3 METHOD

**Notations.** Let $\mathcal{X}$ be the dataset with $N$ samples where the number of singleton label classes is $K$ and $\mathbb{S}$ be the set of their feature vectors $\boldsymbol{h}$. Then, we formulate them as $\mathcal{X} := [(\boldsymbol{x}_i, c_i)]_{i=1}^N$ where $c_i$ is the class label of the $i$-th sample $x_i$ and $\mathbb{S} := \{\boldsymbol{h}_i\}_{i=1}^N$. As a result, we define $\boldsymbol{y}_i = \boldsymbol{e}^{(c_i)}$ as the one-hot vector of $x_i$. Then, we denote the subset of $\mathbb{S}$ which has only $k$-th class feature vectors $\boldsymbol{h}_{k,i}$ as $\mathbb{S}_k := \{\boldsymbol{h}_{k,i}\}_{i=1}^{n_k}$ where $n_k$ is the number of $k$-th class samples and $k \in [K]$. Thus, $N = \sum_{k=1}^K n_k$.

**Overview of Mixup.** Mixup randomly permutes input samples and blends them with the ones before permutation, respectively. Let $\mathcal{I} := [i]_{i=1}^{N}$ be the indices of $\mathcal{X}$ and $\pi(\mathcal{I}) := [\pi(i)]_{i=1}^{N}$ be the permuted one where $\pi(i)$ represents the index number corresponding to $i$-th element of $\mathcal{I}$. Therefore, the index pairs of mixed samples $\mathcal{I}^{\lambda}$ is denoted as $\mathcal{I}^{\lambda} := [(i, \pi(i))]_{i \in \mathcal{I}}$. In this case, we denote $\mathcal{I}_{(a,b)}^{\lambda}$ as the index pairs of $(c_i, c_{\pi(i)}) = (a, b)$, and $\mathbb{S}_{(a,b)}^{\lambda}$ as the mixed feature set of $(a, b)$-label samples. Therefore, $\mathbb{S}_{(a,b)}^{\lambda} := \{\lambda \boldsymbol{h}_{a,i} + (1 - \lambda)\boldsymbol{h}_{b,j} \,|(i, j) \in \mathcal{I}_{(a,b)}^{\lambda}\} = \{\boldsymbol{h}_{(a,b),i}^{\lambda}\}_{i=1}^{n_{(a,b)}}$ where $(a, b) \in \mathbb{K}^2$, $n_{(a,b)} = |\mathcal{I}_{(a,b)}^{\lambda}|$, and $\mathbb{K}^2 = \{(a, b)|1 \le a \le K, \, 1 \le b \le K\}$. Thus, $N = \sum_{(a,b) \in \mathbb{K}^2} n_{(a,b)}$.

Based on the notations, we perform mixup on each pair defined by $\mathcal{I}^{\lambda}$ to create mixed-label samples by linearly interpolating them:

$$\boldsymbol{x}_i^{\lambda} = \lambda \boldsymbol{x}_i + (1 - \lambda)\boldsymbol{x}_{\pi(i)}, \boldsymbol{y}_i^{\lambda} = \lambda \boldsymbol{y}_{c_i} + (1 - \lambda)\boldsymbol{y}_{c_{\pi(i)}}, \forall (i, \pi(i)) \in \mathcal{I}^{\lambda}, \tag{1}$$

where the mixup ratio $\lambda \in (0, 1)$ is sampled from the beta distribution $D_{\lambda}$, i.e., $\lambda \sim D_{\lambda}(\alpha, \alpha)$ and $\alpha$ is a hyperparameter.

**Balanced Mixed Label Sampler.** We propose the Balanced Mixed Label Sampler (BMLS), where the frequency of all mixed-label samples is equal in each epoch as shown in Figure 1. When using BMLS, the probability of sampling of a $(a, b)$-label sample is

$$P_{(i, \pi(i))|(i, \pi(i)) \in \tilde{\mathcal{I}}^{\lambda}} = \frac{1}{N}. \tag{2}$$

$\tilde{\mathcal{I}}^{\lambda}$ is the index pairs of samples where the frequency of mixed labels is balanced, i.e., $n_{(a,b)} = n$ for all $(a, b) \in \mathbb{K}^2$. As done in the class-aware sampler (Shen & Lin, 2016), we remove the randomness by pre-defining $\tilde{\mathcal{I}}^{\lambda}$ for every epoch. After generating $\tilde{\mathcal{I}}^{\lambda}$, we simply replace $\mathcal{I}^{\lambda}$ to $\tilde{\mathcal{I}}^{\lambda}$ in Eq. 1.

*As proven in Theorem 1 and Proposition 1, we show that the minority collapse observed in Mixup arises from the imbalanced frequency of mixed-label samples* (The theorems and proofs are deferred for clarity of exposition). Consequently, the proposed sampler mitigates the minority collapse of Mixup by performing sampling after pre-balancing the frequency of all label samples, including mixed labels, as formulated in Eq. 2.

**Mixed-Singleton Classifier.** Let $\boldsymbol{W} \in \mathbb{R}^{K \times p}$ be a classifier of singleton labels, which is a fully-connected layer. We define the Mixed-Singleton classifier (MS) as

$$\boldsymbol{W}^{\lambda} = [\lambda \boldsymbol{w}_a + (1 - \lambda)\boldsymbol{w}_b]_{(a,b) \in \mathbb{K}^2}, \tag{3}$$

where $p$ is the last-layer feature dimension, as shown in Figure 1. We replace the singleton classifier with MS and perform Mixup with BMLS, where mixed-label samples $\tilde{\boldsymbol{x}}_i^{\lambda}$ and their one-hot vectors $\tilde{\boldsymbol{y}}_i^{\lambda}$ are defined as:

$$\tilde{\boldsymbol{x}}_i^{\lambda} = \lambda \boldsymbol{x}_i + (1 - \lambda)\boldsymbol{x}_{\pi(i)}, \tilde{\boldsymbol{y}}_i^{\lambda} = \boldsymbol{e}^{\mathcal{I}^2(c_i, c_{\pi(i)})}, \forall (i, \pi(i)) \in \tilde{\mathcal{I}}^{\lambda}, \tag{4}$$

where $\mathcal{I}^2$ denotes the index pairs of $\mathbb{K}^2$, and $\mathcal{I}^2(a, b)$ gives the index number of $(a, b) \in \mathbb{K}^2$.

During the proof of Theorem 1, we focused on the observation that *oversampling can mitigate the minority collapse of Mixup regardless of the mixup lambda $\lambda$* in Eq. 19. Motivated by this, *we treated each mixed label as a new singleton class*. As a result, the proposed classifier improves the accuracy on minority classes, thereby strengthening the minority collapse mitigation effect of BMLS.

Building on these methods, we generated mixed labels $(a, b)$ only for the case where $a < b$, ensuring that the existing theorem and proposition still hold, thereby mitigating the limitations of both methods. The limitation and proof are described in $7 and Appendix C.5.

## 4 THEORETICAL ANALYSIS

### 4.1 PROOF SKETCH

We first present a proof sketch that outlines the approach we followed to propose and prove our theorems. Fang et al. (2021) proved that oversampling mitigates minority collapse when singleton label samples are imbalanced, following the sequence outlined below. (Gray indicates the part as defined in Fang et al. (2021).)

(1) Define the Layer-Peeled Model. (Eq. 7)
(2) Prove that NC properties are satisfied when the LPM has global optimality in the case where singleton label samples are balanced. (Theorem 1)
(3) Demonstrate that the LPM suffers from minority collapse in the case where singleton label samples are imbalanced. (Lemma 1 and Theorem 5)
(4) Show that oversampling alleviates minority collapse in the imbalanced case. (Proposition 1)

Our theorem and proof leverages strategies similar to those in Fang et al. (2021), but we extend these concepts to Mixup focusing on the balance of mixed label samples.

In $4.2, (1) we define the Layer-Peeled Model with Mixup (LPM$_\lambda$) and omit step (2), which holds true according to the theorem of Fisher et al. (2024); (3) we prove that in the imbalanced case the LPM$_\lambda$ also suffers from minority collapse; and in closing, (4) we show that the Balanced Mixed Label Sampler (BMLS) alleviates the minority collapse. In $4.3, we extend the LPM$_\lambda$ by modifying the classifier: (1) we newly define the Layer-Peeled Model with Mixup and Mixed-Singleton classifier (LPM$_\lambda$-MS); (2) we prove that when this model achieves global optimality, it also satisfies the NC properties; and finally, following the same reasoning as in $4.2, (3–4) we show that in the imbalanced case the LPM$_\lambda$-MS suffers from minority collapse, and that BMLS is effective to the minority collapse even in this setting.

## 4.2 BALANCING MIXED LABELS MITIGATE THE MINORITY COLLAPSE OF MIXUP

**(1) Problem Settings.** The Layer-Peeled Model (LPM) (Fang et al., 2021) is the optimization program of simplified neural network, modeled by only last-layer features and classifier. Following the definition of LPM, we obtain the Layer-Peeled Model with Mixup (LPM$_\lambda$):

$$\min_{\boldsymbol{W}, \boldsymbol{H}^\lambda} \mathbb{E}_\lambda \frac{1}{N} \sum_{k \in \mathbb{K}^2} \sum_{i=1}^{n_k} \mathcal{L}(\boldsymbol{W}\boldsymbol{h}_{k,i}^\lambda, \boldsymbol{y}_k^\lambda) \text{ s.t. } \frac{1}{K} \sum_{k=1}^{K} \|\boldsymbol{w}_k\|^2 \leq E_W, \ \frac{1}{K^2} \sum_{k \in \mathbb{K}^2} \frac{1}{n_k} \sum_{i=1}^{n_k} \|\boldsymbol{h}_{k,i}^\lambda\|^2 \leq E_H,$$
(5)

where $\boldsymbol{y}_{(a,b)}^\lambda = \lambda \boldsymbol{e}^{(a)} + (1-\lambda)\boldsymbol{e}^{(b)}$. For simplicity, we hereafter denote $\boldsymbol{W} = [\boldsymbol{w}_k]_{k=1}^K \in \mathbb{R}^{K \times p}$ for the weights of the classifier and the positive thresholds $E_W \propto 1/K$ and $E_H \propto 1/K$.

We present a convex optimization program that serves as a relaxation of the non-convex LPM$_\lambda$ (Eq. 5), leveraging the established result that a quadratically constrained quadratic program can be transformed into a semidefinite program (Sturm & Zhang, 2003). This formulation is provided as Eq. 11 in Appendix B.

**(2) Satisfying NC properties.** As proven in Fisher et al. (2024), when LPM$_\lambda$ (Eq. 5) has the global optimality, NC properties are satisfied. We omit this step.

**(3) Minority collapse occurs in LPM$_\lambda$.** Now, we are ready for proving that LPM$_\lambda$ also suffers from minority collapse. Lemma 1 below relates the solutions of Eq. 11 to that of Eq. 5.

**Lemma 1.** *Assume $p \geq K^2 + K$ and the loss function $\mathcal{L}$ is convex in its first argument. Let $\boldsymbol{X}^\star$ be a minimizer of the convex program (Eq. 11). Define $(\boldsymbol{W}^\star, \boldsymbol{H}^\star)$ as*

$$\left[\boldsymbol{h}_{(1,1)}^\star, \boldsymbol{h}_{(1,2)}^\star, \ldots, \boldsymbol{h}_{(K,K)}^\star, (\boldsymbol{W}^\star)^\top\right] = \boldsymbol{P}(\boldsymbol{X}^\star)^{1/2},$$
(6)

$$\boldsymbol{h}_{k,i}^\star = \boldsymbol{h}_k^\star, \ \text{for all } i \in \mathcal{I}_k^\lambda, k \in \mathbb{K}^2,$$

*where $(\boldsymbol{X}^\star)^{1/2}$ denotes the positive square root of $\boldsymbol{X}^\star$ and $\boldsymbol{P} \in \mathbb{R}^{p \times (K^2+K)}$ is any partial orthogonal matrix such that $\boldsymbol{P}^\top \boldsymbol{P} = \boldsymbol{I}_{K^2+K}$. Then, $(\boldsymbol{W}^\star, \boldsymbol{H}^\star)$ is a minimizer of Eq. 5. Moreover, if all $\boldsymbol{X}^\star$'s satisfy $\frac{1}{K^2} \sum_{k=1}^{K^2} \boldsymbol{X}^\star(k,k) = E_H$, then all the solutions of Eq. 5 are in the form of Eq. 6.*

*Proof.* See Appendix C.1 □

**Theorem 1.** *Assume $p \geq K$ and $n_A/n_B \to \infty$, and fix $K_A$ and $K_B$. Let $(\boldsymbol{W}^\star, \boldsymbol{H}^\star)$ be any global minimizer of the LPM$_\lambda$ (Eq. 5). As the imbalance factor $R \equiv n_A/n_B \to \infty$, we have*

$$\lim \boldsymbol{w}_k^\star - \boldsymbol{w}_{k'}^\star = \boldsymbol{0}_p, \ \text{for all } K_A < k < k' \leq K.$$

*Proof.* See Appendix C.3 □

From Lemma 1 and Theorem 1, we demonstrate that LPM$_\lambda$ also exhibits minority collapse.

**(4) Balancing mixed labels mitigates minority collapse in LPM$_\lambda$.** To formalize the behavior of a neural network trained by minimizing a new program with balanced samples including mixed-label ones through BMLS, we propose that it may perform as if it were trained on a larger dataset containing $n_A$ examples in the majority class and $w_r n_B$ examples in the minority class. We begin by analyzing the LPM$_\lambda$ in the context of BMLS:

$$\min_{\boldsymbol{W}, \boldsymbol{H}^\lambda} \frac{1}{N'} \left[ \sum_{k \in \mathbb{K}_A^2} \sum_{i=1}^{n_A} \mathcal{L}(\boldsymbol{W} \boldsymbol{h}_{k,i}^\lambda, \boldsymbol{y}_k^\lambda) + w_r \sum_{k \in \mathbb{K}_B^2} \sum_{i=1}^{n_B} \mathcal{L}(\boldsymbol{W} \boldsymbol{h}_{k,i}^\lambda, \boldsymbol{y}_k^\lambda) \right] \tag{7}$$

$$\text{s.t.} \ \frac{1}{K} \sum_{k=1}^{K} \|\boldsymbol{w}_k\|^2 \leq E_W, \ \frac{1}{|\mathbb{K}_A^2|} \sum_{k \in \mathbb{K}_A^2} \frac{1}{n_A} \sum_{i=1}^{n_A} \|\boldsymbol{h}_{k,i}^\lambda\|^2 + \frac{1}{|\mathbb{K}_B^2|} \sum_{k \in \mathbb{K}_B^2} \frac{1}{n_B} \sum_{i=1}^{n_B} \|\boldsymbol{h}_{k,i}^\lambda\|^2 \leq E_H,$$

where $N' = n_A |\mathbb{K}_A^2| + w_r n_B |\mathbb{K}_B^2|$

The following result supports the intuition that BMLS enhances the size of the minority classes in the LPM$_\lambda$. For simplicity, we omit the superscript $\lambda$ in Proposition 1.

**Proposition 1.** *Assume $p \geq K^2 + K$ and the loss function $\mathcal{L}$ is convex in the first argument. Let $\boldsymbol{X}^\star$ be any minimizer of the convex program (Eq. 11) with $n_{(1,1)} = n_{(1,2)} = \cdots = n_{(K_A, K_A)} = n_A$ and $n_{(K_A+1, K_A+1)} = n_{(K_A+1, K_A+2)} = \cdots = n_{(K,K)} = w_r n_B$. Define $(\boldsymbol{W}^\star, \boldsymbol{H}^\star)$ as*

$$\left[ \boldsymbol{h}_{(1,1)}^\star, \boldsymbol{h}_{(1,2)}^\star, \ldots, \boldsymbol{h}_{(K,K)}^\star, (\boldsymbol{W}^\star)^\top \right] = \boldsymbol{P}(\boldsymbol{X}^\star)^{1/2}, \tag{8}$$

$$\boldsymbol{h}_{k_A, i}^\star = \boldsymbol{h}_{k_A}^\star, \ \text{for all } i \in \mathcal{I}_{k_A}^\lambda, k_A \in \mathbb{K}_A^2, \ \boldsymbol{h}_{k_B, i}^\star = \boldsymbol{h}_{k_B}^\star, \ \text{for all } i \in \mathcal{I}_{k_B}^\lambda, k_B \in \mathbb{K}_B^2,$$

*where $\boldsymbol{P} \in \mathbb{R}^{p \times (K^2 + K)}$ is any partial orthogonal matrix such that $\boldsymbol{P}^\top \boldsymbol{P} = \boldsymbol{I}_{K^2 + K}$. Then, $(\boldsymbol{W}^\star, \boldsymbol{H}^\star)$ is a global minimizer of the mixed-label balanced LPM$_\lambda$ (Eq. 7). Moreover, if all $\boldsymbol{X}^\star$'s satisfy $\frac{1}{K^2} \sum_{k \in \mathbb{K}^2} \boldsymbol{X}^\star(k, k) = E_H$, then all the solutions of Eq. 7 are in the form of Eq. 8.*

*Proof.* See Appendix C.2. □

In conjunction with Lemma 1, Proposition 1 demonstrates that the number of training examples in each minority mixed label is effectively $w_r n_B$ instead of $n_B$ in the LPM$_\lambda$. In the special case where $w_r = n_A / n_B \equiv R$, the results indicate that the angles between any pair of class vectors are equal, regardless of whether they belong to the majority or minority classes.

*Remark* 1. According to Theorem 1, Mixup also experiences the minority collapse. Additionally, as proven in Proposition 1, even when using class-balanced samplers to alleviate label suppression and learn an unbiased classifier, minority collapse is partially mitigated but not fully resolved, as the frequency of mixed labels remains imbalanced. For this reason, when using Mixup in imbalanced learning, the frequency of not only singleton labels but also mixed ones should be balanced.

### 4.3 Enhancing Minority Collapse Mitigation via Singleton Interpretation

Building on Theorem 1 and Proposition 1, we raise a conjecture: *If mixed labels are interpreted as singletons, then the mitigation of minority collapse will be enhanced.*

The rationale for the conjecture can be summarized as follows: *(i) Difference between Mixup loss and mixed feature*. In Proposition 1, minority collapse occurs regardless of the mixup ratio $\lambda$, as illustrated in Eq. 19. This is because the total loss derived from features is equivalent to that obtained without Mixup. However, the behavior of features differs: while the loss is divided between classes according to the mixup ratio $\lambda$, the mixed features are not generally decomposed in this way due to the non-linearity of the model; *(ii) Similar importance of singleton and mixed labels in minority collapse*. In addition, the minority collapse of LPM$_\lambda$ depends not only on the number of singleton label samples but also on that of mixed-label samples, as if the mixed labels were singletons; *(iii) Negative impact of Mixup loss on classifier learning*. Furthermore, it has been reported that Mixup primarily facilitates representation learning while exerting a minimal or adverse effect on classifier

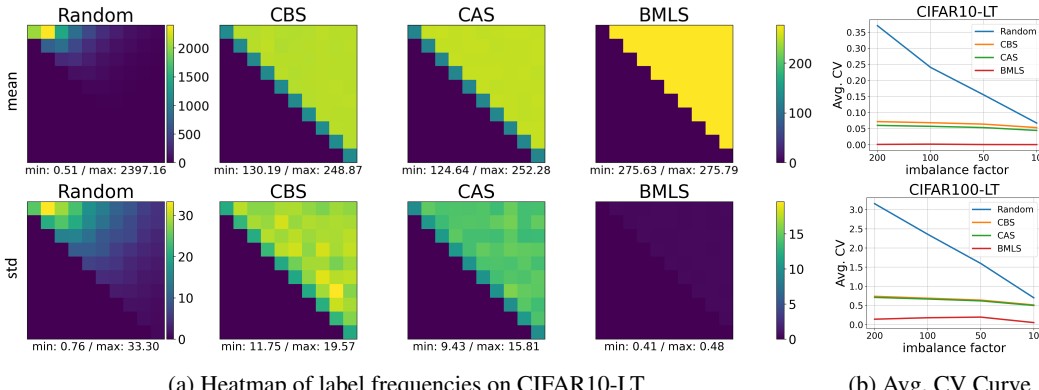

(a) Heatmap of label frequencies on CIFAR10-LT      (b) Avg. CV Curve

Figure 2: Mean and standard deviation of label frequencies including mixed label across epochs. (a) Higher imbalance factor means higher imbalanced, and (b) the closer Avg. CV is to 0, the more evenly the labels appear across epochs

learning (Zhong et al., 2021). For this reason, *would it not be more effective in alleviating minority collapse to interpret mixed labels as singletons, as this reduces the adverse effect of Mixup?*

**(1) Problem Settings.** By replacing the classifier as Mixed-Singleton classifier defined in §3, we obtain the LPM$_\lambda$ with Mixed-Singleton classifier (LPM$_\lambda$-MS):

$$\min_{\boldsymbol{W}^\lambda, \boldsymbol{H}^\lambda} \mathbb{E}_\lambda \frac{1}{N} \sum_{k \in \mathbb{K}^2} \sum_{i=1}^{n_k} \mathcal{L}(\boldsymbol{W}^\lambda \boldsymbol{h}_{k,i}^\lambda, \boldsymbol{y}_k^\lambda)$$

$$\text{s.t.} \quad \frac{1}{|\mathbb{K}^2|} \sum_{k \in \mathbb{K}^2} \left\| \boldsymbol{w}_k^\lambda \right\|^2 \leq E_W, \quad \frac{1}{K^2} \sum_{k \in \mathbb{K}^2} \frac{1}{n_k} \sum_{i=1}^{n_k} \left\| \boldsymbol{h}_{k,i}^\lambda \right\|^2 \leq E_H, \tag{9}$$

where the only differences are $\boldsymbol{W}^\lambda = [\lambda \boldsymbol{w}_a + (1-\lambda) \boldsymbol{w}_b]_{(a,b) \in \mathbb{K}^2}$.

**(2) Satisfying NC properties.** In this setting, LPM$_\lambda$-MS has the same global minimum with that of the LPM in balanced case where the number of classes is $K$ due to the linear interpolation property of $W_{(a,b)}^\lambda$. (See Eq. 46 proven in Theorem 3.) As a result, the LPM$_\lambda$-MS also satisfies NC properties.

**(3-4)** Therefore, we omit steps (3-4) and conclude Theorem 2.

For simplicity, we remove the superscript $\lambda$ in Theorem 2.

**Theorem 2.** *Assume $p \geq 2K^2$ and the loss function $\mathcal{L}$ is convex in the first argument. Let $\boldsymbol{X}^\star$ be any minimizer of the convex program with $n_{(1,1)} = n_{(1,2)} = \cdots = n_{(K_A, K_A)} = n_A$ and $n_{(K_A+1, K_A+1)} = n_{(K_A+1, K_A+2)} = \cdots = n_{(K,K)} = w_r n_B$. Define $(\boldsymbol{W}^\star, \boldsymbol{H}^\star)$ as*

$$\left[ \boldsymbol{h}_{(1,1)}^\star, \boldsymbol{h}_{(1,2)}^\star, \ldots, \boldsymbol{h}_{(K,K)}^\star, (\boldsymbol{W}^\star)^\top \right] = \boldsymbol{P}(\boldsymbol{X}^\star)^{1/2}, \tag{10}$$

$$\boldsymbol{h}_{k,i}^\star = \boldsymbol{h}_k^\star, \text{ for all } i \in \mathcal{I}_k^\lambda, k \in \mathbb{K}_A^2, \quad \boldsymbol{h}_{k,i}^\star = \boldsymbol{h}_k^\star, \text{ for all } i \in \mathcal{I}_k^\lambda, k \in \mathbb{K}_B^2,$$

*where $\boldsymbol{P} \in \mathbb{R}^{p \times 2K^2}$ is any partial orthogonal matrix such that $\boldsymbol{P}^\top \boldsymbol{P} = \boldsymbol{I}_{2K^2}$. Then $(\boldsymbol{W}^\star, \boldsymbol{H}^\star)$ is a global minimizer of the mixed-label balanced LPM$_\lambda$-MS.*

*Proof.* Theorem 2 follows directly from the same arguments applied to oversampling-adjusted LPM in imbalanced case, which has already been proven in Fang et al. (2021). We omit the proof here. □

*Remark* 2. As proven in Theorem 2, balancing mixed labels and interpreting them as singletons allows the LPM$_\lambda$-MS to operate in the same manner of the LPM. At the same time, it is expected to preserve the strong feature learning effect of Mixup while potentially reducing its negligible influence on classifier learning by maintaining mixed-label samples but removing the mixup loss.

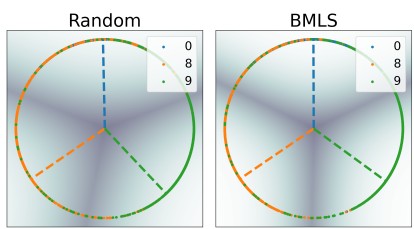

| $\mathcal{D}$ | Ctgy. | Coverage ratio | | | | | |
|---|---|---|---|---|---|---|---|
| | | 0.0 | 0.2 | 0.4 | 0.6 | 0.8 | 1.0 |
| C10 | few | 58.74 | 58.56 | 58.50 | 58.51 | 60.06 | 65.60 |
| | med | 68.46 | 71.09 | 74.63 | 77.85 | 77.10 | 75.21 |
| | many | 86.93 | 92.13 | 91.79 | 91.49 | 91.91 | 89.59 |
| | all | 71.08 | 73.64 | 74.94 | 76.14 | 76.43 | 76.64 |
| C100 | few | 7.72 | 10.28 | 12.17 | 12.66 | 13.16 | 13.20 |
| | med | 34.49 | 39.13 | 42.95 | 44.47 | 46.21 | 46.97 |
| | many | 61.26 | 65.64 | 66.49 | 67.85 | 68.93 | 67.60 |
| | all | 36.37 | 40.31 | 42.50 | 43.66 | 44.81 | 44.60 |

Figure 3: Mixed-label frequency control experiments on CIFAR10/100 LT datasets. Coverage ratio represents the proportion of mixed labels used in training during one epoch compared to the total number of mixed labels. (e.g., when coverage ratio is 0.6 in CIFAR100-LT, the model trains on mixed labels consisting of combinations of 60 different classes, which change with each epoch.) (figure) Test Acc. (%) and Avg. CV over coverage ratio (table) Comparison of test accuracies

| Sampler | Test Acc. (%) ↑ | $U_G$ ↑ | $U$ ↑ |
|---|---|---|---|
| Random | 72.91 | 14.6404 | 4.2325 |
| CBS | 75.86 | 15.2521 | 4.4848 |
| CAS | 76.60 | 15.3319 | 4.4999 |
| BMLS | 78.71 | 15.3379 | 4.5651 |

Figure 4: Experiments on CIFAR10-LT dataset for the effectiveness of BMLS to minority collapse. (figure) Visualization of 2D-projection of class vectors about Many class $\{0\}$ and Few classes $\{8, 9\}$. Dashed line indicates each class vector and contrast of background means the confidence value, i.e., a confidence close to 0.5 indicates that the model is confused between the two classes for the given sample, and this is represented by darker colors in the figure. (table) Quantitative comparison results. ($U_G$: Uniformity of all classes, $U$: Uniformity of $\{0, 8, 9\}$ classes)

## 5 EXPERIMENTAL RESULTS

To empirically validate the effectiveness of our analysis and proposed solutions, we conducted experiments in various imbalanced environments. We used CIFAR10/100-LT, Places-LT, ImageNet-LT and iNaturalist2018, with five repeated experiments with random seeds in CIFAR10/100-LT and three in others. The tables presenting the experimental results show the average of test accuracies. Detailed criteria and descriptions of the evaluation results reported in the table are provided in Appendix E In all tables, *imb* refers to the imbalance factor, *C10/100* represents the CIFAR10/100-LT datasets, *Clf.* refers to the classifier, and BMLS$_{MS}$ denotes the method using both BMLS and MS. Unless otherwise specified, all experiments include Mixup. Best in bold. Implementation details are illustrated in Appendix D.

### 5.1 EMPIRICAL VALIDATION

**Epoch-wise Label Imbalance.** To demonstrate the empirical evidence of *Remark* 1, we examine the mean and standard deviation of label frequencies from various sampler: random sampler, class-balanced sampler (CBS) (Kang et al., 2020), class-aware sampler (CAS) (Shen & Lin, 2016), and ours (BMLS), as shown in Figure 2. We use the average of Coefficient of Variation ($\overline{\text{CV}}$) (Dodge, 2008) as the metric to measure the dispersion of each label frequency distributions: $\overline{\text{CV}} = \frac{1}{C} \sum_{c=1}^{C} \frac{\sigma_c}{\mu_c}$, where the lower $\overline{\text{CV}}$, the less dispersion, which means labels evenly appear across epochs. After training, the mean of label frequencies is almost balanced across all samplers, but epoch-wise balance is not. To empirically validate that the epoch-wise label imbalance is a problem in imbalanced learning, we do mixed-label frequency control experiments. As shown in Figure 3, the more imbalanced mixed label appears from epoch to epoch, the lower the performance of models.

**The Effect of Balanced Mixed Label Sampler.** As shown in Figure 3, epoch-wise imbalance not

Table 1: Experiments on CIFAR10/100-LT datasets with imbalance factor 200 and 100 for effectiveness of Multi-Singleton Classifier (higher imbalance factor is more imbalanced)

| Sampler | Dataset | Clf. | imb200 | | | | imb100 | | | |
|---|---|---|---|---|---|---|---|---|---|---|
| | | | many | med | few | all | many | med | few | all |
| BMLS | C10 | FC | **90.49** | **74.12** | 54.43 | 73.13 | 88.53 | **77.84** | 70.53 | 78.85 |
| | | MS | 88.94 | 72.97 | **62.77** | **74.70** | **89.14** | 76.34 | **74.63** | **79.67** |
| | | diff. | -1.55 | -1.15 | +8.34 | +1.57 | +0.61 | -1.50 | +4.10 | +0.82 |
| BMLS | C100 | FC | **65.77** | 41.73 | 7.19 | 40.36 | **68.98** | 46.13 | 14.98 | 45.32 |
| | | MS | 63.24 | **44.86** | **11.19** | **41.71** | 66.31 | **49.80** | **21.80** | **47.62** |
| | | diff. | -2.53 | +3.13 | +4.00 | +1.35 | -2.67 | +3.67 | +6.82 | +2.30 |

Table 2: Experiments on CIFAR10/100-LT datasets with various imbalance factors. (†: the reported values are taken from each reference paper. More references in Table 7)

| Method | CIFAR10-LT | | | | CIFAR100-LT | | | |
|---|---|---|---|---|---|---|---|---|
| | imbalance factor | | | | imbalance factor | | | |
| | 200 | 100 | 50 | 10 | 200 | 100 | 50 | 10 |
| ERM+CAS[†] | N/A | 68.40 | N/A | 86.90 | N/A | 31.90 | N/A | 55.00 |
| Mixup[†] | 67.30 | 72.80 | 78.60 | 87.70 | 38.70 | 43.00 | 48.10 | 58.20 |
| LOM[†] | N/A | 74.20 | N/A | 89.40 | N/A | 41.50 | N/A | 59.90 |
| ETF+DR[†] | 71.90 | 76.50 | 81.00 | 87.70 | 40.90 | 45.30 | 50.40 | N/A |
| Remix[†] | N/A | 73.00 | N/A | 88.50 | N/A | 41.40 | N/A | 59.50 |
| DBN-mix[†] | 79.58 | 83.47 | 86.82 | 90.87 | **46.21** | **51.04** | 54.93 | 64.98 |
| Mixup | 66.77 | 72.94 | 78.64 | 88.05 | 39.06 | 42.88 | 48.31 | 63.03 |
| +LOM | 70.17 | 76.63 | 81.15 | 89.24 | 39.61 | 44.24 | 49.99 | 63.90 |
| +CAS | 69.90 | 76.43 | 81.42 | 89.24 | 40.28 | 44.65 | 50.07 | 63.57 |
| +BMLS$_{MS}$ | 74.70 | 79.67 | 83.46 | 88.51 | 41.71 | 47.62 | 52.74 | 64.47 |
| diff. | +7.93 | +6.73 | +4.82 | +0.46 | +2.65 | +4.74 | +4.43 | +1.44 |
| ETF+DR | 71.58 | 76.82 | 81.25 | 87.59 | 41.20 | 45.07 | 50.71 | 63.08 |
| BMLS+WETF$_{MS}$+CE | 77.73 | 80.31 | 84.22 | 88.26 | 42.73 | 47.10 | 52.44 | 64.10 |
| diff. | +6.15 | +3.49 | +2.97 | +0.67 | +1.53 | +2.03 | +1.73 | +1.02 |
| Remix | 69.58 | 75.15 | 80.41 | 88.61 | 41.03 | 44.95 | 50.19 | 63.45 |
| +BMLS | 73.95 | 80.10 | 83.92 | 88.62 | 39.95 | 46.34 | 51.53 | 64.42 |
| +BMLS$_{MS}$ | 73.18 | 78.00 | 83.70 | 88.20 | 40.25 | 46.82 | 49.78 | 63.54 |
| diff. | +3.60 | +2.85 | +3.29 | -0.41 | -0.78 | +1.87 | -0.41 | +0.09 |
| DBN-mix | 77.40 | 82.40 | 86.05 | **91.01** | 40.71 | 45.52 | 50.47 | 62.68 |
| +BMLS$_{MS}$ | **79.73** | **84.30** | **87.28** | 90.93 | 44.42 | 49.08 | **55.41** | **65.42** |
| diff. | +2.33 | +1.90 | +1.23 | -0.08 | +3.71 | +3.56 | +4.94 | +2.74 |

only of singleton labels but also of mixed ones affects model performance. While class-balanced sampling methods such as CBS and CAS oversamples singleton label samples within each mini-batch, Mixup ruins the balance of both singleton labels and mixed ones by randomly permuting input samples and blending them each other. Empirically, we observe that enforcing balance among mixed labels through BMLS improves model performance, promoting more balanced classifier, as demonstrated on Figure 4.

**The Effect of Mixed-Singleton Classifier.** To validate the Mixed-Singleton classifier and support the conjecture in §4.3, we compared a singleton classifier (FC) and ours (MS). As shown in Table 1, MS further boosts performance, particularly for few classes. This improvement indicates that MS facilitates less minority collapse in few classes, and the effect still maintains even though the degree of imbalance increases.

Table 3: Experiments on large datasets. (*:use pre-trained model) (More detail results in Table 5)

| Method | Places-LT | Places-LT* | ImageNet-LT | iNaturalist18 |
|---|---|---|---|---|
| random | 22.06 | 25.90 | 45.19 | 64.62 |
| CBS | 24.79 | 37.32 | 47.49 | 67.06 |
| CAS | 24.26 | 37.44 | 47.31 | **67.55** |
| BMLS | 27.33 | 37.39 | **48.83** | 66.98 |
| BMLS$_{MS}$ | **27.95** | **37.81** | 47.54 | 56.60 |

## 5.2 STANDARD IMBALANCED LEARNING BENCHMARKS

**Results and Analysis on Small Datasets.** To evaluate the performance of our method, we selected Mixup, CAS, and LOM—the latter being the most similar to our approach—as baselines. As shown in Table 2, our proposed method achieves the highest performance on CIFAR10-LT and CIFAR100-LT across all settings, except for the case with an imbalance factor of 10, where class imbalance is relatively mild. Furthermore, when classes are categorized into *many*, *medium*, and *few* based on their sample frequency, and test accuracy is measured accordingly (see Table 8 in Appendix E), BMLS demonstrates the largest improvement for *few* classes compared to other baselines. These results indicate that BMLS mitigates minority collapse more effectively than other class-balanced samplers.

**Integration with ETF classifier, Remix, and DBN-mix.** To validate the generality of our approach, its effectiveness across diverse settings, and its compatibility with other Mixup-based methods, we reproduced several representative techniques: (i) ETF+DR (Yang et al., 2022), an NC-inspired method that fixes the classifier to a simplex ETF form; (ii) Remix (Chou et al., 2020), which re-balances the Mixup lambda according to class sample counts; and (iii) DBN-mix (Baik et al., 2024), which substantially improves imbalanced learning performance through bilateral Mixup and a double-branch architecture. Then, we applied our proposed method to each of them. All experimental settings are identical to ours, and detailed descriptions of the reproducibility process and the integration of our method with each baseline are provided in Appendix D. As shown in Table 2, our proposed methods significantly improve the performance of prior mixup-based methods by seamlessly integrating them. Even in DBN-mix experiments, our proposed methods achieve performance that is competitive with state-of-the-art methods. Through integration experiments with a range of Mixup-based methods, we demonstrate that our proposed method has the potential to serve as an effective sampler and classifier, facilitating the development of new state-of-the-art methods. More detailed comparative results for ETF+DR and Remix can be found in Table 6 (Appendix E) and Table 14 (Appendix F.1), respectively.

**Results and Analysis on Large Datasets.** In practical experimental settings, both BMLS and MS exhibit limitations depending on the number of classes $K$. First, BMLS struggles when $K^2$ is bigger than the dataset size, as it fails to generate mixed samples uniformly across all mixed-labels in each epoch. This leads to the same issue seen in traditional class-balanced samplers, we already introduced, epoch-wise label imbalance. MS, in addition to the issues faced by BMLS, suffers from an exponential increase in the number of class vectors for mixed labels as $K$ grows. Concurrently, the number of samples available for learning each class vector decreases significantly, raising the potential for underfitting. As shown in the results in Table 3, the effect of BMLS$_{MS}$ diminishes as the number of classes increases (*i.e.*, $K_{PL} = 365 < K_{IN} = 1000 < K_{iNat18} = 8142$). However, despite these limitations, BMLS$_{MS}$ demonstrates superior performance compared to other class-balanced samplers on Place-LT, and when only BMLS is used on ImageNet-LT, it achieves the highest performance, while improving the accuracy on few classes. (See Table 9 and Table 10 in Appendix E.) Even in the most challenging case, iNaturalist2018, using only BMLS still results in competitive performance compared to other class-balanced samplers.

## 5.3 ABLATION STUDY

To empirically validate whether our proposed methods effectively address the minority collapse issue and improve model performance in imbalanced learning environments, we conducted an ablation study. As shown in Table 4, applying both BMLS and MS together resulted in the largest performance improvement. Moreover, in scenarios where the number of samples in *few* classes is extremely small (e.g., imbalance factors of 200 and 100 in CIFAR100-LT), where both MS and FC face the most challenging imbalanced condition, MS alone actually outperforms.

Table 4: Ablation study on CIFAR10/100-LT datasets with various imbalance factors including $K^2$ classifier (notated as $K^2$ on the table). The results are the mean of five repeated experiments with random seeds. Best in bold (CBS: Class-Balanced Sampler, CAS: Class-Aware Sampler, BMLS: Balanced Mixed Label Sampler)

| Sampler | Clf. | CIFAR10-LT | | | | CIFAR100-LT | | | |
|---|---|---|---|---|---|---|---|---|---|
| | | imbalance factor | | | | imbalance factor | | | |
| | | 200 | 100 | 50 | 10 | 200 | 100 | 50 | 10 |
| *Sampler* | | | | | | | | | |
| random | FC | 66.77 | 72.94 | 78.64 | 88.05 | 39.06 | 42.88 | 48.31 | 63.03 |
| BMLS | FC | **73.13** | **78.85** | **83.07** | **89.46** | **40.03** | **45.20** | **51.99** | **65.72** |
| *Classifier* | | | | | | | | | |
| random | MS | 53.11 | 64.08 | 68.56 | 80.56 | 33.42 | 36.87 | 41.66 | 56.71 |
| random | $K^2$ | 34.86 | 39.01 | 42.20 | 51.60 | 7.90 | 8.72 | 9.22 | 16.41 |
| BMLS | MS | **74.70** | **79.67** | **83.46** | **88.51** | **41.71** | **47.62** | **52.74** | **64.47** |

$K^2$ **Classifier.** As shown in the results, the $K^2$ classifier performs worse than MS alone, and even worse than when MS is combined with a random sampler. This degradation occurs because the use of a $K^2$ classifier drastically reduces the number of samples available to learn each class vector, leading to underfitting due to insufficient class-vector learning. Through this experiment, we empirically confirm that the performance improvement of MS is not attributable to increased classifier capacity, but rather to the effect of the linear interpolation between class vectors induced by mixup ratio $\lambda$.

# 6 CONCLUSION

The research problem targeted in this study is the issue of minority collapse in imbalanced learning environments, where class imbalance negatively impacts model performance, particularly for minority classes. We analyzed the impact of Mixup on this problem and identified two key findings: first, minority collapse is influenced by the frequency balance of mixed labels, and second, when mixed labels are balanced, interpreting them as singletons enhances reducing the minority collapse. Based on these findings, we proposed BMLS and MS as solutions. BMLS balanced mixed-label frequencies more effectively, while MS leveraged the singleton interpretation to further enhance classifier performance. These methods demonstrated significant effectiveness in mitigating minority collapse and improving model performance, particularly for minority class¡ samples. Through experiments, we validated the utility and versatility of the proposed methods, showing that both BMLS and MS consistently improved performance compared to existing baselines and demonstrated their applicability across different datasets and imbalance factors.

# 7 LIMITATIONS AND FUTURE WORK

**Scalability.** As observed in the experimental results and analysis for large datasets, both BMLS and MS suffer from issues related to epoch-wise label imbalance and underfitting class vectors due to the exponential increase in the number of mixed labels, which is proportional to the number of singleton labels $K$. Additionally, in this study, to ensure a fair comparison, we matched the number of samples learned per epoch to those generated by a random sampler (*e.g.*, in iNaturalist2018, we used 437,513 images, while the number of mixed labels was $K^2 = 66,292,164$ with $K = 8,142$). As explained in $3, this paper partially addresses the issue by reducing the diversity of mixed labels. However, if the number of training samples is sufficiently increased without considering the constraint, it could also serve as a technical solution.

**Integration with other methods.** In this study, we extend our methods to Remix, ETF+DR, and DBN-mix. However, both BMLS and MS are methods that can be used in conjunction with other Mixup-based methods for imbalanced learning. Through the experiments with the previous methods, we demonstrated the potential for integration with other methods. We anticipate that future research will explore these integrations to more effectively mitigate minority collapse.

## REPRODUCIBILITY STATEMENT

We summarize the reproducibility statement of this paper as follow.

- $3. To reproduce BMLS and MS, we define notations and provide helpful preliminaries with a theoretical support in Appendix C.5.
- $4. To prove our theorems such as Theorem 1, Proposition 1, and Theorem 2, we demonstrate the detailed proofs of them in Appendix C.
- $5. All experiments can be reproduced using our text supplementary materials (Appendix D), which provide dataset descriptions, model architectures, and hyperparameter settings, as well as our code including configuration files for each experiment. Additionally, experimental requirements, such as necessary libraries, are specified in the README files included with the code.

In addition, our codes can be accessed at *link* (T.B.A)

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

# APPENDIX

## DETAILS ABOUT LARGE LANGUAGE MODELS IN PAPER WRITING

In this paper, the authors used LLMs solely for the purpose of checking mistranslations or grammar.

## A  ADDITIONAL RELATED WORK

### A.1  MIXUP-BASED METHOD

**Data augmentation.** Mixup (Zhang et al., 2018) generates mixed-label samples by interpolating between input samples, extending training distribution support. Manifold Mixup (Verma et al., 2019) applies this technique to intermediate layers, regularizing the network by encouraging less confident predictions. CP-Mix, or Confusion-Pairing Mixup (Yoon et al., 2025), augments samples based on confusion pairs, addressing data deficiency by enhancing the model's ability to distinguish frequently misclassified class pairs. ExtraMix (Kwon et al., 2023) introduces a mixup technique capable of extrapolation, broadening both feature and label distributions, which minimizes label imbalance more effectively than traditional methods. CutMix (Yun et al., 2019; Zhao & Lei, 2021; Pan et al., 2024) focuses on mixed-label sample generation by cutting and pasting image patches, creating a regional dropout effect. CMO (Park et al., 2021) extends this idea by pasting minority class images onto majority class backgrounds, enriching minority class samples with context from majority class images. OTMix (Gao et al., 2023) improves upon this by using Optimal Transport to adaptively combine majority class backgrounds with minority class foregrounds, ensuring semantically reasonable mixed images.

**Architecture.** BBN (Zhou et al., 2020), SBN, and DBN (Baik et al., 2024) utilize different architectures to enhance both representation and classifier learning. These methods incorporate bilateral mixup or decoupling strategies to optimize performance for imbalanced datasets. OTLR (Liu et al., 2019) uses dynamic meta-embedding and modulated attention to map images into a feature space

that respects both closed-world classification and the novelty of the open world, improving the generalization of imbalanced datasets.

**Calibration or two-stage.** UniMix (Xu et al., 2021) balances class distributions by introducing a novel mixing factor and sampler that favors the minority class. MiSLAS (Zhong et al., 2021) decouples representation and classifier learning, improving both calibration and performance in imbalanced data scenarios.

While many attempts have been made to address the challenges of imbalanced learning environments using Mixup, including data augmentation, architecture improvements, and calibration methods, no research has specifically focused on the balance of mixed labels in such contexts.

## A.2 CLASS-BALANCED METHODS

**Re-balance.** Remix (Chou et al., 2020) applies a higher mixup ratio to minority classes, rebalancing the data without sampling. Re-weighting (Elkan, 2001; Byrd & Lipton, 2019; Cui et al., 2019) adjusts the loss function by tuning class weights, with methods like Balanced SoftMax (Ren et al., 2020) explicitly considering label distribution shifts during optimization. Logit Adj (Menon et al., 2021) adjusts logits based on label frequencies, promoting a larger margin between rare positive and dominant negative labels. $\tau$-Norm (Kang et al., 2020) normalizes classifier weight norms according to class size, rebalancing decision boundaries. LDAM loss (Cao et al., 2019) improves generalization by replacing standard cross-entropy with a margin-based approach, tailored to handle imbalanced datasets. cRT (Kang et al., 2020) re-trains the classifier using class-balanced sampling, improving the model's generalization ability. LWS (Kang et al., 2020) focuses on re-scaling classifier weights to ensure a balanced learning process for imbalanced datasets.

**Re-/Over-Sampling.** M2M (Kim et al., 2020) augments minority classes by translating samples from majority classes, enhancing generalization for minority class features. MixBoost (Kabra et al., 2020) iteratively selects and combines majority and minority class instances to create hybrid samples, improving model performance. The Meta Sampler (Ren et al., 2020), built on balanced SoftMax, adapts the sampling rate through meta-learning to alleviate over-balancing issues. CB Sampling (Kang et al., 2020) ensures that each class has an equal probability of being selected, balancing the dataset during training. Class-Aware Sampler (CAS) (Shen & Lin, 2016) is more specific method of CB Sampling, which explicitly ensures the class frequency balance on each mini-batch. Label-Occurrence Mixup (LOM) (Zhang et al., 2022) uses two CB samplers to sample input pairs, respectively. CSA (Shi et al., 2023) generates diverse training images for tail classes by maintaining a context bank from head-class images.

Various class-balanced samplers have been proposed, yet no research has specifically focused on the balance of mixed labels. Additionally, while methods such as Logit Adjustment and UniMix have concentrated on singleton-labels, they did not interpret mixed labels as singletons.

## A.3 NEURAL COLLAPSE IN MIXUP AND IMBALANCED LEARNING

NC in imbalanced learning has been studied in Fang et al. (2021). To alleviate the minority collapse, Yang et al. (2022) assumed that the classifier is fixed to the K-simplex ETF and proved that LPM with the classifier satisfies NC properties. Also, the fixed ETF classifier with Mixup has improved the model performance in imbalanced learning. Building on the theorems, Fisher et al. (2024) proved Mixup also satisfies NC properties for both same class and different class. However, Yang et al. (2022) and Fisher et al. (2024) did not consider the minority collapse from the mixed label balance in the LPM with learnable classifiers.

# B CONVEX OPTIMIZATION PROGRAM

To begin with, defining $\boldsymbol{h}_k^\lambda = \frac{1}{n_k} \sum_{i=1}^{n_k} \boldsymbol{h}_{k,i}^\lambda$ as the feature mean of the $\mathbb{S}_k^\lambda$ where $k \in \mathbb{K}^2$, we introduce a new decision variable $\boldsymbol{X} = [\boldsymbol{h}_{(1,1)}^\lambda, \boldsymbol{h}_{(1,2)}^\lambda, \ldots, \boldsymbol{h}_{(K,K)}^\lambda, \boldsymbol{W}^\top]^\top [\boldsymbol{h}_{(1,1)}^\lambda, \boldsymbol{h}_{(1,2)}^\lambda, \ldots, \boldsymbol{h}_{(K,K)}^\lambda, \boldsymbol{W}^\top] \in \mathbb{R}^{(K^2+K) \times (K^2+K)}$. By definition, $\boldsymbol{X}$ is positive semi-definite and satisfies

$$\frac{1}{K^2} \sum_{k=1}^{K^2} \boldsymbol{X}(k,k) = \frac{1}{K^2} \sum_{k \in \mathbb{K}^2} \|\boldsymbol{h}_k^\lambda\|^2 \overset{a}{\leq} \frac{1}{K^2} \sum_{k \in \mathbb{K}^2} \frac{1}{n_k} \sum_{i=1}^{n_k} \|\boldsymbol{h}_{k,i}^\lambda\|^2 \leq E_H$$

and

$$\frac{1}{K} \sum_{k=K^2+1}^{K^2+K} \boldsymbol{X}(k,k) = \frac{1}{K} \sum_{k=1}^{K} \|\boldsymbol{w}_k\|^2 \leq E_W,$$

where $\overset{a}{\leq}$ follows from the Cauchy-Schwarz inequality. Thus, we consider the following semi-definite programming problem:

$$\min_{\boldsymbol{X} \in \mathbb{R}^{(K^2+K) \times (K^2+K)}} \sum_{k \in \mathbb{K}^2} \frac{n_k}{N} \mathcal{L}(\boldsymbol{z}(k)^\lambda, \boldsymbol{y}_k^\lambda)$$

$$\text{s.t. } \boldsymbol{X} \succeq 0, \tag{11}$$

$$\frac{1}{K^2} \sum_{k=1}^{K^2} \boldsymbol{X}(k,k) \leq E_H, \quad \frac{1}{K} \sum_{k=K^2+1}^{K^2+K} \boldsymbol{X}(k,k) \leq E_W,$$

$$\text{for all } 1 \leq k \leq K^2,$$

$$\boldsymbol{z}_k = \left[ \boldsymbol{X}(k, K^2+1), \boldsymbol{X}(k, K^2+2), \ldots, \boldsymbol{X}(k, K^2+K) \right]^\top.$$

When $\mathcal{L}$ is the cross-entropy loss with softmax function,

$$\mathcal{L}(\boldsymbol{z}^\lambda(k), \boldsymbol{y}_k^\lambda) = -\lambda \log \left( \frac{\exp(\boldsymbol{z}^\lambda(a))}{\sum_{k'=1}^{K} \exp(\boldsymbol{z}^\lambda(k'))} \right) - (1-\lambda) \log \left( \frac{\exp(\boldsymbol{z}^\lambda(b))}{\sum_{k'=1}^{K} \exp(\boldsymbol{z}^\lambda(k'))} \right),$$

where $\boldsymbol{z}^\lambda(k')$ denotes the $k'$-th entry of the logit $\boldsymbol{z}_i^\lambda = \boldsymbol{W} \boldsymbol{h}_{k,i}^\lambda$, and $k = (a, b)$.

## C  PROOFS

### C.1  PROOF OF LEMMA 1

**Restated Lemma 1.** Assume $p \geq K^2 + K$ and the loss function $\mathcal{L}$ is convex in its first argument. Let $\boldsymbol{X}^\star$ be a minimizer of the convex program (Eq. 11). Define $(\boldsymbol{W}^\star, \boldsymbol{H}^\star)$ as

$$\left[\boldsymbol{h}^\star_{(1,1)}, \boldsymbol{h}^\star_{(1,2)}, \ldots, \boldsymbol{h}^\star_{(K,K)}, \, (\boldsymbol{W}^\star)^\top\right] = \boldsymbol{P}(\boldsymbol{X}^\star)^{1/2},$$

$$\boldsymbol{h}^\star_{k,i} = \boldsymbol{h}^\star_k, \quad \text{for all } i \in \mathcal{I}^\lambda_k, k \in \mathbb{K}^2,$$

where $(\boldsymbol{X}^\star)^{1/2}$ denotes the positive square root of $\boldsymbol{X}^\star$ and $\boldsymbol{P} \in \mathbb{R}^{p \times (K^2 + K)}$ is any partial orthogonal matrix such that $\boldsymbol{P}^\top \boldsymbol{P} = \boldsymbol{I}_{K^2 + K}$. Then, $(\boldsymbol{W}^\star, \boldsymbol{H}^\star)$ is a minimizer of Eq. 5. Moreover, if all $\boldsymbol{X}^\star$'s satisfy $\frac{1}{K^2} \sum_{k=1}^{K^2} \boldsymbol{X}^\star(k, k) = E_H$, then all the solutions of Eq. 5 are in the form of Eq. 6.

*Proof.* For any feasible solution $(\boldsymbol{W}, \boldsymbol{H}^\lambda)$ for the original program Eq. 5, we define

$$\boldsymbol{h}^\lambda_k := \frac{1}{n_k} \sum_{i=1}^{n_k} \boldsymbol{h}_{k,i}, \ k \in \mathbb{K}^2,$$

and

$$\boldsymbol{X} := \left[\boldsymbol{h}^\lambda_{(1,1)}, \boldsymbol{h}^\lambda_{(1,2)}, \ldots, \boldsymbol{h}^\lambda_{(K,K)}, \boldsymbol{W}^\top\right]^\top \left[\boldsymbol{h}^\lambda_{(1,1)}, \boldsymbol{h}^\lambda_{(1,2)}, \ldots, \boldsymbol{h}^\lambda_{(K,K)}, \boldsymbol{W}^\top\right].$$

Clearly, $\boldsymbol{X} \succeq 0$. For the other two constraints of Eq. 11, we have

$$\frac{1}{K^2} \sum_{k=1}^{K^2} \boldsymbol{X}(k, k) = \frac{1}{K^2} \sum_{k \in \mathbb{K}^2} \|\boldsymbol{h}^\lambda_k\|^2 \overset{a}{\leq} \frac{1}{K^2} \sum_{k \in \mathbb{K}^2} \frac{1}{n_k} \sum_{i=1}^{n_k} \|\boldsymbol{h}^\lambda_{k,i}\|^2 \overset{b}{\leq} E_H$$

and

$$\frac{1}{K} \sum_{k=K^2+1}^{K^2+K} \boldsymbol{X}(k, k) = \frac{1}{K} \sum_{k=1}^{K} \|\boldsymbol{w}_k\|^2 \overset{c}{\leq} E_W,$$

where $\overset{a}{\leq}$ applies Jensen's inequality and $\overset{b}{\leq}$ and $\overset{c}{\leq}$ use that $(\boldsymbol{W}, \boldsymbol{H}^\lambda)$ is a feasible solution. So $\boldsymbol{X}$ is a feasible solution for the convex program Eq. 11. Letting $L_0$ be the global minimum of Eq. 11, for any feasible solution $(\boldsymbol{W}, \boldsymbol{H}^\lambda)$, we obtain

$$\frac{1}{N} \sum_{k \in \mathbb{K}^2} \sum_{i=1}^{n_k} \mathcal{L}(\boldsymbol{W}\boldsymbol{h}^\lambda_{k,i}, \boldsymbol{y}^\lambda_k) = \sum_{k \in \mathbb{K}^2} \frac{n_k}{N} \left[\frac{1}{n_k} \sum_{i=1}^{n_k} \mathcal{L}(\boldsymbol{W}\boldsymbol{h}^\lambda_{k,i}, \boldsymbol{y}^\lambda_k)\right]$$

$$\overset{a}{\geq} \sum_{k \in \mathbb{K}^2} \frac{n_k}{N} \mathcal{L}(\boldsymbol{W}\boldsymbol{h}^\lambda_k, \boldsymbol{y}^\lambda_k) = \sum_{k \in \mathbb{K}^2} \frac{n_k}{N} \mathcal{L}(\boldsymbol{z}(k)^\lambda, \boldsymbol{y}^\lambda_k) \geq L_0, \quad (12)$$

where in $\overset{a}{\geq}$, we use $\mathcal{L}$ is convex on the first argument, and so $\mathcal{L}(\boldsymbol{W}\boldsymbol{h}^\lambda, \boldsymbol{y}^\lambda_k)$ is convex on $\boldsymbol{h}$ given $\boldsymbol{W}$ and $k \in \mathbb{K}^2$.

For the simplicity of our expressions, we hereafter remove the superscript $\lambda$ of $\boldsymbol{H}^\lambda$, $\boldsymbol{h}^\lambda$ and $\boldsymbol{z}^\lambda$.

On the other hand, considering the solution $(\boldsymbol{W}^\star, \boldsymbol{H}^\star)$ defined in Eq. 6 with $\boldsymbol{X}^\star$ being a minimizer of Eq. 11, we have $\left[\boldsymbol{h}^\star_{(1,1)}, \boldsymbol{h}^\star_{(1,2)}, \ldots, \boldsymbol{h}^\star_{(K,K)}, \boldsymbol{W}^\top\right]^\top \left[\boldsymbol{h}^\star_{(1,1)}, \boldsymbol{h}^\star_{(1,2)}, \ldots, \boldsymbol{h}^\star_{(K,K)}, \boldsymbol{W}^\top\right] = \boldsymbol{X}^\star$ ($p \geq K^2 + K$ guarantees the existence of $\left[\boldsymbol{h}^\star_{(1,1)}, \boldsymbol{h}^\star_{(1,2)}, \ldots, \boldsymbol{h}^\star_{(K,K)}, (\boldsymbol{W}^\star)^\top\right]$). We can verify that $(\boldsymbol{W}^\star, \boldsymbol{H}^\star)$ is a feasible solution for Eq. 5 and have

$$\frac{1}{N} \sum_{k \in \mathbb{K}^2} \sum_{i=1}^{n_k} \mathcal{L}(\boldsymbol{W}^\star \boldsymbol{h}^\star_{k,i}, \boldsymbol{y}^\lambda_k) = \sum_{k \in \mathbb{K}^2} \frac{n_k}{N} \mathcal{L}(\boldsymbol{z}(k)^\star, \boldsymbol{y}^\lambda_k) = L_0, \quad (13)$$

where $\boldsymbol{z}(k)^\star = \left[\boldsymbol{X}^\star(k, K^2 + 1), \boldsymbol{X}^\star(k, K^2 + 2), \ldots, \boldsymbol{X}^\star(k, K^2 + K)\right]^\top$ for $k \in \mathbb{K}^2$.

Combining Eq. 12 and Eq. 13, we conclude that $L_0$ is the global minimum of Eq. 5 and $(\boldsymbol{W}^\star, \boldsymbol{H}^\star)$ is a minimizer.

Suppose there is a minimizer $(\boldsymbol{W}', \boldsymbol{H}')$ that cannot be written as Eq. 6. Let

$$\boldsymbol{h}'_k = \frac{1}{n_k} \sum_{i=1}^{n_k} \boldsymbol{h}'_{k,i}, \ k \in \mathbb{K}^2,$$

and

$$\boldsymbol{X}' = \left[\boldsymbol{h}'_{(1,1)}, \boldsymbol{h}'_{(1,2)}, \ldots, \boldsymbol{h}'_{(K,K)}, (\boldsymbol{W}')^\top\right]^\top \left[\boldsymbol{h}'_{(1,1)}, \boldsymbol{h}'_{(1,2)}, \ldots, \boldsymbol{h}'_{(K,K)}, (\boldsymbol{W}')^\top\right].$$

Eq. 12 implies that $\boldsymbol{X}'$ is a minimizer of Eq. 11. As $(\boldsymbol{W}', \boldsymbol{H}')$ cannot be written as Eq. 6 with $\boldsymbol{X}^\star = \boldsymbol{X}'$, then there is a $k' \in \mathbb{K}^2$, $i, j \in [n_{k'}]$ with $i \neq j$ such that $\boldsymbol{h}_{k',i} \neq \boldsymbol{h}_{k',j}$. We have

$$\frac{1}{K^2} \sum_{k=1}^{K^2} \boldsymbol{X}'(k,k) = \frac{1}{K^2} \sum_{k \in \mathbb{K}^2} \|\boldsymbol{h}'_k\|^2$$

$$= \frac{1}{K^2} \sum_{k \in \mathbb{K}^2} \frac{1}{n_k} \sum_{i=1}^{n_k} \|\boldsymbol{h}'_{k,i}\|^2 - \frac{1}{K^2} \sum_{k \in \mathbb{K}^2} \frac{1}{n_k} \sum_{i=1}^{n_k} \|\boldsymbol{h}'_{k,i} - \boldsymbol{h}'_k\|^2$$

$$\leq \frac{1}{K^2} \sum_{k \in \mathbb{K}^2} \frac{1}{n_k} \sum_{i=1}^{n_k} \|\boldsymbol{h}'_{k,i}\|^2 - \frac{1}{K^2} \frac{1}{n_{k'}} (\|\boldsymbol{h}'_{k',i} - \boldsymbol{h}'_{k'}\|^2 + \|\boldsymbol{h}'_{k',j} - \boldsymbol{h}'_{k'}\|^2)$$

$$\leq \frac{1}{K^2} \sum_{k \in \mathbb{K}^2} \frac{1}{n_k} \sum_{i=1}^{n_k} \|\boldsymbol{h}'_{k,i}\|^2 - \frac{1}{K^2} \frac{1}{2n_{k'}} \|\boldsymbol{h}'_{k',i} - \boldsymbol{h}'_{k',j}\|^2$$

$$< E_H.$$

By contraposition, if all $\boldsymbol{X}^\star$ satisfy that $\frac{1}{K^2} \sum_{k=1}^{K^2} \boldsymbol{X}^\star(k,k) = E_H$, then all the solutions of Eq. 5 are in the form of Eq. 6. We complete the proof. $\qquad\square$

## C.2 PROOF OF PROPOSITION 1

**Restated Proposition 1.** Assume $p \geq K^2 + K$ and the loss function $\mathcal{L}$ is convex in the first argument. Let $\boldsymbol{X}^\star$ be any minimizer of the convex program (Eq. 11) with $n_{(1,1)} = n_{(1,2)} = \cdots = n_{(K_A,K_A)} = n_A$ and $n_{(K_A+1,K_A+1)} = n_{(K_A+1,K_A+2)} = \cdots = n_{(K,K)} = w_r n_B$. Define $(\boldsymbol{W}^\star, \boldsymbol{H}^\star)$ as

$$\left[ \boldsymbol{h}^\star_{(1,1)}, \boldsymbol{h}^\star_{(1,2)}, \ldots, \boldsymbol{h}^\star_{(K,K)}, (\boldsymbol{W}^\star)^\top \right] = \boldsymbol{P}(\boldsymbol{X}^\star)^{1/2},$$

$$\boldsymbol{h}^\star_{k_A,i} = \boldsymbol{h}^\star_{k_A}, \ \text{for all } i \in \mathcal{I}^\lambda_{k_A}, k_A \in \mathbb{K}^2_A, \ \boldsymbol{h}^\star_{k_B,i} = \boldsymbol{h}^\star_{k_B}, \ \text{for all } i \in \mathcal{I}^\lambda_{k_B}, k_B \in \mathbb{K}^2_B,$$

where $\boldsymbol{P} \in \mathbb{R}^{p \times (K^2+K)}$ is any partial orthogonal matrix such that $\boldsymbol{P}^\top \boldsymbol{P} = \boldsymbol{I}_{K^2+K}$. Then, $(\boldsymbol{W}^\star, \boldsymbol{H}^\star)$ is a global minimizer of the mixed-label balanced LPM$_\lambda$ (Eq. 7). Moreover, if all $\boldsymbol{X}^\star$'s satisfy $\frac{1}{K^2} \sum_{k \in \mathbb{K}^2} \boldsymbol{X}^\star(k,k) = E_H$, then all the solutions of Eq. 7 are in the form of Eq. 8.

*Proof.* For any feasible solution $(\boldsymbol{W}, \boldsymbol{H}^\lambda)$ for the original program Eq. 5, we define

$$\boldsymbol{h}^\lambda_{k_A} := \frac{1}{n_A} \sum_{i=1}^{n_A} \boldsymbol{h}_{k_A,i}, \ k_A \in \mathbb{K}^2_A, \ \text{and } \boldsymbol{h}^\lambda_{k_B} := \frac{1}{w_r n_B} \sum_{i=1}^{w_r n_B} \boldsymbol{h}_{k_B,i}, \ k_B \in \mathbb{K}^2_B,$$

and

$$\boldsymbol{X} := \left[ \boldsymbol{h}^\lambda_{(1,1)}, \boldsymbol{h}^\lambda_{(1,2)}, \ldots, \boldsymbol{h}^\lambda_{(K,K)}, \boldsymbol{W}^\top \right]^\top \left[ \boldsymbol{h}^\lambda_{(1,1)}, \boldsymbol{h}^\lambda_{(1,2)}, \ldots, \boldsymbol{h}^\lambda_{(K,K)}, \boldsymbol{W}^\top \right].$$

Clearly, $\boldsymbol{X} \succeq 0$. For the other two constraints of Eq. 11, we have

$$\frac{1}{K^2} \sum_{k=1}^{K^2} \boldsymbol{X}(k,k) = \frac{1}{K^2} \sum_{k \in \mathbb{K}^2} \|\boldsymbol{h}^\lambda_k\|^2$$

$$\overset{a}{\leq} \frac{1}{K^2} \left( \sum_{k_A \in \mathbb{K}^2_A} \frac{1}{n_A} \sum_{i=1}^{n_A} \|\boldsymbol{h}^\lambda_{k_A,i}\|^2 + \sum_{k_B \in \mathbb{K}^2_B} \frac{1}{w_r n_B} \sum_{i=1}^{w_r n_B} \|\boldsymbol{h}^\lambda_{k_B,i}\|^2 \right)$$

$$\overset{b}{\leq} E_H$$

and

$$\frac{1}{K} \sum_{k=K^2+1}^{K^2+K} \boldsymbol{X}(k,k) = \frac{1}{K} \sum_{k=1}^{K} \|\boldsymbol{w}_k\|^2 \overset{c}{\leq} E_W,$$

where $\overset{a}{\leq}$ applies Jensen's inequality and $\overset{b}{\leq}$ and $\overset{c}{\leq}$ use that $(\boldsymbol{W}, \boldsymbol{H}^\lambda)$ is a feasible solution. So $\boldsymbol{X}$ is a feasible solution for the convex program Eq. 11. Letting $L_0$ be the global minimum of Eq. 11, for any feasible solution $(\boldsymbol{W}, \boldsymbol{H}^\lambda)$, we obtain

$$\frac{1}{N} \sum_{k \in \mathbb{K}^2} \sum_{i=1}^{n_k} \mathcal{L}(\boldsymbol{W} \boldsymbol{h}^\lambda_{k,i}, \boldsymbol{y}^\lambda_k)$$

$$= \sum_{k_A \in \mathbb{K}^2_A} \frac{n_A}{N} \left[ \frac{1}{n_A} \sum_{i=1}^{n_A} \mathcal{L}(\boldsymbol{W} \boldsymbol{h}^\lambda_{k_A,i}, \boldsymbol{y}^\lambda_{k_A}) \right] + \sum_{k_B \in \mathbb{K}^2_B} \frac{w_r n_B}{N} \left[ \frac{1}{w_r n_B} \sum_{i=1}^{w_r n_B} \mathcal{L}(\boldsymbol{W} \boldsymbol{h}^\lambda_{k_B,i}, \boldsymbol{y}^\lambda_{k_B}) \right]$$

$$\overset{a}{\geq} \sum_{k_A \in \mathbb{K}^2_A} \frac{n_A}{N} \mathcal{L}(\boldsymbol{W} \boldsymbol{h}^\lambda_{k_A}, \boldsymbol{y}^\lambda_{k_A}) + \sum_{k_B \in \mathbb{K}^2_B} \frac{w_r n_B}{N} \mathcal{L}(\boldsymbol{W} \boldsymbol{h}^\lambda_{k_B}, \boldsymbol{y}^\lambda_{k_B})$$

$$= \sum_{k_A \in \mathbb{K}^2_A} \frac{n_A}{N} \mathcal{L}(\boldsymbol{z}(k_A)^\lambda, \boldsymbol{y}^\lambda_{k_A}) + \sum_{k_B \in \mathbb{K}^2_B} \frac{w_r n_B}{N} \mathcal{L}(\boldsymbol{z}(k_B)^\lambda, \boldsymbol{y}^\lambda_{k_B}) \geq L_0, \tag{14}$$

where in $\overset{a}{\geq}$, we use $\mathcal{L}$ is convex on the first argument, and so $\mathcal{L}(\boldsymbol{W} \boldsymbol{h}^\lambda, \boldsymbol{y}^\lambda_k)$ is convex on $\boldsymbol{h}$ given $\boldsymbol{W}$ and $k \in \mathbb{K}^2$.

For the simplicity of our expressions, we hereafter remove the superscript $\lambda$ of $\boldsymbol{H}^\lambda, \boldsymbol{h}^\lambda$ and $\boldsymbol{z}^\lambda$.

On the other hand, considering the solution $(\boldsymbol{W}^\star, \boldsymbol{H}^\star)$ defined in Eq. 6 with $\boldsymbol{X}^\star$ being a minimizer of Eq. 11, we have $\left[\boldsymbol{h}^\star_{(1,1)}, \boldsymbol{h}^\star_{(1,2)}, \ldots, \boldsymbol{h}^\star_{(K,K)}, \boldsymbol{W}^\top\right]^\top \left[\boldsymbol{h}^\star_{(1,1)}, \boldsymbol{h}^\star_{(1,2)}, \ldots, \boldsymbol{h}^\star_{(K,K)}, \boldsymbol{W}^\top\right] = \boldsymbol{X}^\star$ ($p \geq K^2 + K$ guarantees the existence of $\left[\boldsymbol{h}^\star_{(1,1)}, \boldsymbol{h}^\star_{(1,2)}, \ldots, \boldsymbol{h}^\star_{(K,K)}, (\boldsymbol{W}^\star)^\top\right]$). We can verify that $(\boldsymbol{W}^\star, \boldsymbol{H}^\star)$ is a feasible solution for Eq. 5 and have

$$\frac{1}{N} \sum_{k \in \mathbb{K}^2} \sum_{i=1}^{n_k} \mathcal{L}(\boldsymbol{W}^\star \boldsymbol{h}^\star_{k,i}, \boldsymbol{y}^\lambda_k) = \sum_{k_A \in \mathbb{K}^2_A} \frac{n_A}{N} \mathcal{L}(\boldsymbol{z}(k_A)^\star, \boldsymbol{y}^\lambda_{k_A}) + \sum_{k_B \in \mathbb{K}^2_B} \frac{w_r n_B}{N} \mathcal{L}(\boldsymbol{z}(k_B)^\star, \boldsymbol{y}^\lambda_{k_B}) = L_0,$$

(15)

where $\boldsymbol{z}(k_A)^\star = \left[\boldsymbol{X}^\star(k_A, K^2+1), \boldsymbol{X}^\star(k_A, K^2+2), \ldots, \boldsymbol{X}^\star(k_A, K^2+K_A)\right]^\top$ for $k_A \in \mathbb{K}^2_A$ and $\boldsymbol{z}(k_B)^\star = \left[\boldsymbol{X}^\star(k_B, K^2+K_A+1), \boldsymbol{X}^\star(k_B, K^2+K_A+2), \ldots, \boldsymbol{X}^\star(k_B, K^2+K)\right]^\top$ for $k_B \in \mathbb{K}^2_B$.

Combining Eq. 14 and Eq. 15, we conclude that $L_0$ is the global minimum of Eq. 5 and $(\boldsymbol{W}^\star, \boldsymbol{H}^\star)$ is a minimizer.

Suppose there is a minimizer $(\boldsymbol{W}', \boldsymbol{H}')$ that cannot be written as Eq. 6. Let

$$\boldsymbol{h}'_{k_A} = \frac{1}{n_A} \sum_{i=1}^{n_A} \boldsymbol{h}'_{k_A,i}, \ k_A \in \mathbb{K}^2_A, \text{ and } \boldsymbol{h}'_{k_B} = \frac{1}{w_r n_B} \sum_{i=1}^{w_r n_B} \boldsymbol{h}'_{k_B,i}, k_B \in \mathbb{K}^2_B$$

and

$$\boldsymbol{X}' = \left[\boldsymbol{h}'_{(1,1)}, \boldsymbol{h}'_{(1,2)}, \ldots, \boldsymbol{h}'_{(K,K)}, (\boldsymbol{W}')^\top\right]^\top \left[\boldsymbol{h}'_{(1,1)}, \boldsymbol{h}'_{(1,2)}, \ldots, \boldsymbol{h}'_{(K,K)}, (\boldsymbol{W}')^\top\right].$$

Eq. 14 implies that $\boldsymbol{X}'$ is a minimizer of Eq. 11. As $(\boldsymbol{W}', \boldsymbol{H}')$ cannot be written as Eq. 6 with $\boldsymbol{X}^\star = \boldsymbol{X}'$, then there is a $k' \in \mathbb{K}^2$, $i,j \in [n'_k]$ with $i \neq j$ such that $\boldsymbol{h}_{k',i} \neq \boldsymbol{h}_{k',j}$. We have

$$\frac{1}{K^2} \sum_{k=1}^{K^2} \boldsymbol{X}'(k,k) = \frac{1}{K^2} \sum_{k \in \mathbb{K}^2} \|\boldsymbol{h}'_k\|^2$$

$$= \frac{1}{K^2} \sum_{k_A \in \mathbb{K}^2_A} \frac{1}{n_A} \sum_{i=1}^{n_A} \|\boldsymbol{h}'_{k_A,i}\|^2 - \frac{1}{K^2} \sum_{k_A \in \mathbb{K}^2_A} \frac{1}{n_A} \sum_{i=1}^{n_A} \|\boldsymbol{h}'_{k_A,i} - \boldsymbol{h}'_{k_A}\|^2$$

$$+ \frac{1}{K^2} \sum_{k_B \in \mathbb{K}^2_B} \frac{1}{w_r n_B} \sum_{i=1}^{w_r n_B} \|\boldsymbol{h}'_{k_B,i}\|^2 - \frac{1}{K^2} \sum_{k_B \in \mathbb{K}^2_B} \frac{1}{w_r n_B} \sum_{i=1}^{w_r n_B} \|\boldsymbol{h}'_{k_B,i} - \boldsymbol{h}'_{k_B}\|^2$$

$$\leq \frac{1}{K^2} \sum_{k_A \in \mathbb{K}^2} \frac{1}{n_A} \sum_{i=1}^{n_A} \|\boldsymbol{h}'_{k_A,i}\|^2 - \frac{1}{K^2} \frac{1}{n_{k'_A}} (\|\boldsymbol{h}'_{k'_A,i} - \boldsymbol{h}'_{k'_A}\|^2 + \|\boldsymbol{h}'_{k'_A,j} - \boldsymbol{h}'_{k'_A}\|^2)$$

$$+ \frac{1}{K^2} \sum_{k_B \in \mathbb{K}^2_B} \frac{1}{w_r n_B} \sum_{i=1}^{w_r n_B} \|\boldsymbol{h}'_{k_B,i}\|^2 - \frac{1}{K^2} \frac{1}{n_{k'_B}} (\|\boldsymbol{h}'_{k'_B,i} - \boldsymbol{h}'_{k'_B}\|^2 + \|\boldsymbol{h}'_{k'_B,j} - \boldsymbol{h}'_{k'_B}\|^2)$$

$$\leq \frac{1}{K^2} \sum_{k_A \in \mathbb{K}^2_A} \frac{1}{n_A} \sum_{i=1}^{n_A} \|\boldsymbol{h}'_{k_A,i}\|^2 - \frac{1}{K^2} \frac{1}{2n_{k'_A}} \|\boldsymbol{h}'_{k'_A,i} - \boldsymbol{h}'_{k'_A,j}\|^2$$

$$+ \frac{1}{K^2} \sum_{k_B \in \mathbb{K}^2_B} \frac{1}{w_r n_B} \sum_{i=1}^{w_r n_B} \|\boldsymbol{h}'_{k_B,i}\|^2 - \frac{1}{K^2} \frac{1}{2n_{k'_B}} \|\boldsymbol{h}'_{k'_B,i} - \boldsymbol{h}'_{k'_B,j}\|^2$$

$$< E_H.$$

By contraposition, if all $\boldsymbol{X}^\star$ satisfy that $\frac{1}{K^2} \sum_{k=1}^{K^2} \boldsymbol{X}^\star(k,k) = E_H$, then all the solutions of Eq. 5 are in the form of Eq. 6. We complete the proof. $\square$

## C.3 PROOF OF THEOREM 1

**Restated Theorem 1.** Assume $p \geq K$ and $n_A/n_B \to \infty$, and fix $K_A$ and $K_B$. Let $(\boldsymbol{W}^\star, \boldsymbol{H}^\star)$ be any global minimizer of the LPM$_\lambda$ (Eq. 5). As the imbalance factor $R \equiv n_A/n_B \to \infty$, we have

$$\lim \boldsymbol{w}_k^\star - \boldsymbol{w}_{k'}^\star = \boldsymbol{0}_p, \quad \text{for all } K_A < k < k' \leq K.$$

To prove Theorem 1, we first study a limit case where we only learn the classification for partial classes. We solve the optimization program:

$$\min_{\boldsymbol{W}, \boldsymbol{H}^\lambda} \mathbb{E}_{\lambda \sim D_\lambda} \quad \frac{1}{|\mathbb{K}_A^2| \cdot n_A} \sum_{k \in \mathbb{K}_A^2} \sum_{i=1}^{n_A} \mathcal{L}(\boldsymbol{W}\boldsymbol{h}_{k,i}^\lambda, \boldsymbol{y}_k^\lambda)$$

$$\text{s.t.} \quad \frac{1}{K} \sum_{k=1}^{K} \|\boldsymbol{w}_k\|^2 \leq E_W, \tag{16}$$

$$\frac{1}{|\mathbb{K}_\cup^2|} \sum_{k \in \mathbb{K}_\cup^2} \frac{1}{n_k} \sum_{i=1}^{n_k} \|\boldsymbol{h}_{k,i}^\lambda\|^2 \leq E_H,$$

where $\boldsymbol{y}_{(a,b)}^\lambda = \lambda \boldsymbol{y}_a + (1-\lambda)\boldsymbol{y}_b$, $\mathbb{K}_A^2 = \{(a,b) | 1 \leq a \leq K_A \wedge 1 \leq b \leq K_A\}$, $\mathbb{K}_B^2 = \{(a,b) | K_A + 1 \leq a \leq K \wedge K_A + 1 \leq b \leq K\}$, $\mathbb{K}_\cup^2 = \mathbb{K}_A^2 \cup \mathbb{K}_B^2$ and

$$n_k = \begin{cases} n_A & \text{if } k = (a,b) \in \mathbb{K}_A^2 \\ n_B & \text{if } k = (a,b) \in \mathbb{K}_B^2 \\ 0 & \text{otherwise} \end{cases}.$$

For the simplicity of our expressions, we remove the superscript $\lambda$ of $\boldsymbol{H}^\lambda$ and $\boldsymbol{h}^\lambda$.

Lemma 2 characterizes useful properties for the minimizer of Eq. 16.

**Lemma 2.** *Let $(\boldsymbol{W}, \boldsymbol{H})$ be a minimizer of Eq. 16. We have $\boldsymbol{h}_{k,i}^\lambda = \boldsymbol{0}_p$ for all $k \in \mathbb{K}_B^2$ and $i \in [n_B]$. Let $L_0$ be the global minimum of Eq. 16. We have*

$$L_0 = \frac{1}{|\mathbb{K}_A^2| \cdot n_A} \sum_{k \in \mathbb{K}_A^2} \sum_{i=1}^{n_A} \mathcal{L}(\boldsymbol{W}\boldsymbol{h}_{k,i}, \boldsymbol{y}_k^\lambda).$$

*Then $L_0$ only depends on $K_A$, $n_A$, $E_H$, and $E_W$. Moreover, for any feasible solution $(\boldsymbol{W}', (\boldsymbol{H})')$, if there exist $k, k' \in \mathbb{K}_B^2$ such that $\|\boldsymbol{w}_k - \boldsymbol{w}_{k'}\| = \epsilon > 0$, we have*

$$\frac{1}{|\mathbb{K}_A^2| \cdot n_A} \sum_{k \in \mathbb{K}_A^2} \sum_{i=1}^{n_A} \mathcal{L}(\boldsymbol{W}\boldsymbol{h}_{k,i}, \boldsymbol{y}_k^\lambda) \geq L_0 + \epsilon',$$

*where $\epsilon' > 0$ depends on $\epsilon$, $|\mathbb{K}_A^2|$, $n_A$, $E_H$, and $E_W$.*

Now we are ready to prove Theorem 1. The proof is based on the contradiction.

*Proof of Theorem 1.* Consider sequences $n_A^\ell$ and $n_B^\ell$ with $R^\ell := n_A^\ell/n_B^\ell$ for $\ell = 1, 2, \ldots$. We have $R^\ell \to \infty$. For each optimization program indexed by $\ell \in \mathbb{N}_+$, we introduce $(\boldsymbol{W}^{\ell,\star}, \boldsymbol{H}^{\ell,\star})$ as a minimizer and separate the objective function into two parts. We consider

$$\mathcal{L}^\ell(\boldsymbol{W}^\ell, \boldsymbol{H}^\ell) = \frac{|\mathbb{K}_A^2| \cdot n_A^\ell}{|\mathbb{K}_A^2| \cdot n_A^\ell + |\mathbb{K}_B^2| \cdot n_B^\ell} \mathcal{L}_A^\ell(\boldsymbol{W}^\ell, \boldsymbol{H}^\ell) + \frac{|\mathbb{K}_B^2| \cdot n_B^\ell}{|\mathbb{K}_A^2| \cdot n_A^\ell + |\mathbb{K}_B^2| \cdot n_B^\ell} \mathcal{L}_B^\ell(\boldsymbol{W}^\ell, \boldsymbol{H}^\ell),$$

with

$$\mathcal{L}_A^\ell(\boldsymbol{W}^\ell, \boldsymbol{H}^\ell) := \frac{1}{|\mathbb{K}_A^2| \cdot n_A^\ell} \sum_{k \in \mathbb{K}_A^2} \sum_{i=1}^{n_A^\ell} \mathcal{L}(\boldsymbol{W}^\ell \boldsymbol{h}_{k,i}^\ell, \boldsymbol{y}_k^\lambda)$$

and

$$\mathcal{L}_B^\ell \left( \boldsymbol{W}^\ell, \boldsymbol{H}^\ell \right) := \frac{1}{|\mathbb{K}_B^2| \cdot n_B^\ell} \sum_{k \in \mathbb{K}_B^2} \sum_{i=1}^{n_B^\ell} \mathcal{L} \left( \boldsymbol{W}^\ell \boldsymbol{h}_{k,i}^\ell, \boldsymbol{y}_k^\lambda \right).$$

We define $\left( \boldsymbol{W}^{\ell,A}, \boldsymbol{H}^{\ell,A} \right)$ as a minimizer of the optimization program:

$$\min_{\boldsymbol{W}^\ell, \boldsymbol{H}^\ell} \quad \mathcal{L}_A^\ell \left( \boldsymbol{W}^\ell, \boldsymbol{H}^\ell \right)$$

$$\text{s.t.} \quad \frac{1}{K} \sum_{k=1}^K \left\| \boldsymbol{w}_k^\ell \right\|^2 \le E_W, \tag{17}$$

$$\frac{1}{|\mathbb{K}_A^2|} \sum_{k \in \mathbb{K}_A^2} \frac{1}{n_A^\ell} \sum_{i=1}^{n_A^\ell} \left\| \boldsymbol{h}_{k,i}^\ell \right\|^2 + \frac{1}{|\mathbb{K}_B^2|} \sum_{k \in \mathbb{K}_B^2} \frac{1}{n_B^\ell} \sum_{i=1}^{n_B^\ell} \left\| \boldsymbol{h}_{k,i}^\ell \right\|^2 \le E_H,$$

and $\left( \boldsymbol{W}^{\ell,B}, \boldsymbol{H}^{\ell,B} \right)$ as a minimizer of the optimization program:

$$\min_{\boldsymbol{W}^\ell, \boldsymbol{H}^\ell} \quad \mathcal{L}_B^\ell \left( \boldsymbol{W}^\ell, \boldsymbol{H}^\ell \right)$$

$$\text{s.t.} \quad \frac{1}{K} \sum_{k=1}^K \left\| \boldsymbol{w}_k^\ell \right\|^2 \le E_W, \tag{18}$$

$$\frac{1}{|\mathbb{K}_A^2|} \sum_{k \in \mathbb{K}_A^2} \frac{1}{n_A^\ell} \sum_{i=1}^{n_A^\ell} \left\| \boldsymbol{h}_{k,i}^\ell \right\|^2 + \frac{1}{|\mathbb{K}_B^2|} \sum_{k \in \mathbb{K}_B^2} \frac{1}{n_B^\ell} \sum_{i=1}^{n_B^\ell} \left\| \boldsymbol{h}_{k,i}^\ell \right\|^2 \le E_H.$$

Note that Programs Eq. 17 and Eq. 18 and their minimizers have been studied in Lemma 2. We define:

$$L_A := \mathcal{L}_A^\ell \left( \boldsymbol{W}^{\ell,A}, \boldsymbol{H}^{\ell,A} \right) \quad \text{and} \quad L_B := \mathcal{L}_B^\ell \left( \boldsymbol{W}^{\ell,B}, \boldsymbol{H}^{\ell,B} \right).$$

Then Lemma 2 implies that $L_A$ and $L_B$ only depend on $|\mathbb{K}_A^2|$, $K_B$, $E_H$, and $E_W$, and are independent of $\ell$. Moreover, since $\boldsymbol{h}_{k,i}^{\ell,A} = \boldsymbol{0}_p$ for all $k \in \mathbb{K}_B^2$ and $i \in [n_B]$, we have

$$\mathcal{L}_B^\ell \left( \boldsymbol{W}^{\ell,A}, \boldsymbol{H}^{\ell,A} \right) = \lambda \cdot \log(K) + (1-\lambda) \cdot \log(K) = \log(K). \tag{19}$$

Now we prove Theorem 1 by contradiction. Suppose there exists a pair $(k, k')$ such that $\lim_{\ell \to \infty} \boldsymbol{w}_k^{\ell,\star} - \boldsymbol{w}_{k'}^{\ell,\star} \ne \boldsymbol{0}_p$. Then there exists $\epsilon > 0$ such that for a subsequence $\{ (\boldsymbol{w}^{a_\ell,\star}, \boldsymbol{h}^{a_\ell,\star}) \}_{\ell=1}^\infty$ and an index $\ell_0$ when $\ell \ge \ell_0$, we have $\left\| \boldsymbol{W}_k^{a_\ell,\star} - \boldsymbol{W}_{k'}^{a_\ell,\star} \right\| \ge \epsilon$. Now we figure out a contradiction by estimating the objective function value on $(\boldsymbol{W}^{a_\ell,\star}, \boldsymbol{H}^{a_\ell,\star})$. In fact, because $(\boldsymbol{W}^{a_\ell,\star}, \boldsymbol{H}^{a_\ell,\star})$ is a minimizer of $\mathcal{L}^\ell(\boldsymbol{W}^\ell, \boldsymbol{H}^\ell)$, we have

$$\mathcal{L}^{a_\ell} \left( \boldsymbol{W}^{a_\ell,\star}, \boldsymbol{H}^{a_\ell,\star} \right) \le \mathcal{L}^{a_\ell} \left( \boldsymbol{W}^{a_\ell,A}, \boldsymbol{H}^{a_\ell,A} \right)$$

$$\overset{Eq.\ 19}{=} \frac{|\mathbb{K}_A^2| \cdot n_A^{a_\ell}}{|\mathbb{K}_A^2| \cdot n_A^{a_\ell} + |\mathbb{K}_B^2| \cdot n_B^{a_\ell}} L_A + \frac{|\mathbb{K}_B^2| \cdot n_B^{a_\ell}}{|\mathbb{K}_A^2| \cdot n_A^{a_\ell} + |\mathbb{K}_B^2| \cdot n_B^{a_\ell}} \log(K)$$

$$= L_A + \frac{1}{K_R R^{a_\ell} + 1} \left( \log(K) - L_A \right) \overset{\ell \to \infty}{\to} L_A, \tag{20}$$

where we define $K_R := |\mathbb{K}_A^2| / |\mathbb{K}_B^2|$ and use $R^\ell = n_A^\ell / n_B^\ell$.

However, when $\ell > \ell_0$, because $\left\| \boldsymbol{w}_k^{a_\ell,\star} - \boldsymbol{w}_{k'}^{a_\ell,\star} \right\| \ge \epsilon > 0$, Lemma 2 implies that

$$\mathcal{L}_A^{a_\ell} \left( \boldsymbol{W}^{a_\ell,\star}, \boldsymbol{H}^{a_\ell,\star} \right) \ge L_A + \epsilon_2,$$

where $\epsilon_2 > 0$ only depends on $\epsilon$, $|\mathbb{K}_A^2|$, $K_B$, $E_H$, and $E_W$, and is independent of $\ell$. We obtain

$$
\begin{aligned}
\mathcal{L}^{a_\ell}\left(\boldsymbol{W}^{a_\ell,\star}, \boldsymbol{H}^{a_\ell,\star}\right) &= \frac{|\mathbb{K}_A^2| \cdot n_A^{a_\ell}}{|\mathbb{K}_A^2| \cdot n_A^{a_\ell} + |\mathbb{K}_B^2| \cdot n_B^{a_\ell}} \mathcal{L}_A^{a_\ell}\left(\boldsymbol{W}^{a_\ell,\star}, \boldsymbol{H}^{a_\ell,\star}\right) \\
&\quad + \frac{|\mathbb{K}_B^2| \cdot n_B^{a_\ell}}{|\mathbb{K}_A^2| \cdot n_A^{a_\ell} + |\mathbb{K}_B^2| \cdot n_B^{a_\ell}} \mathcal{L}_B^{a_\ell}\left(\boldsymbol{W}^{a_\ell,\star}, \boldsymbol{H}^{a_\ell,\star}\right) \\
&\overset{a}{\geq} \frac{|\mathbb{K}_A^2| \cdot n_A^{a_\ell}}{|\mathbb{K}_A^2| \cdot n_A^{a_\ell} + |\mathbb{K}_B^2| \cdot n_B^{a_\ell}} \mathcal{L}_A^{a_\ell}\left(\boldsymbol{W}^{a_\ell,\star}, \boldsymbol{H}^{a_\ell,\star}\right) \\
&\quad + \frac{|\mathbb{K}_B^2| \cdot n_B^{a_\ell}}{|\mathbb{K}_A^2| \cdot n_A^{a_\ell} + |\mathbb{K}_B^2| \cdot n_B^{a_\ell}} \mathcal{L}_B^{a_\ell}\left(\boldsymbol{W}^{a_\ell,B}, \boldsymbol{H}^{a_\ell,B}\right) \\
&= \frac{|\mathbb{K}_A^2| \cdot n_A^{a_\ell}}{|\mathbb{K}_A^2| \cdot n_A^{a_\ell} + |\mathbb{K}_B^2| \cdot n_B^{a_\ell}}(L_A + \epsilon_2) + \frac{|\mathbb{K}_B^2| \cdot n_B^{a_\ell}}{|\mathbb{K}_A^2| \cdot n_A^{a_\ell} + |\mathbb{K}_B^2| \cdot n_B^{a_\ell}} L_B \\
&= L_A + \epsilon_2 + \frac{1}{K_R R^{a_\ell} + 1}(L_B - L_A - \epsilon_2) \overset{\ell \to \infty}{\to} L_A + \epsilon_2, \qquad (21)
\end{aligned}
$$

where $\overset{a}{\geq}$ uses $\left(\boldsymbol{W}^{a_\ell,B}, \boldsymbol{H}^{a_\ell,B}\right)$ is the minimizer of Eq. 18. Thus we meet contradiction by comparing Eq. 20 with Eq. 21 and achieve Theorem 1. □

*Proof of Lemma 2.* For any constants $C_a > 0$, $C_b > 0$, and $C_c > 0$, define

$$
\begin{aligned}
C_a' &:= \frac{C_a}{C_a + (K_A - 1)C_b + K_B C_c} \in (0,1) \\
C_b' &:= \frac{C_b}{C_a + (K_A - 1)C_b + K_B C_c} \in (0,1) \\
C_c' &:= \frac{C_c}{C_a + (K_A - 1)C_b + K_B C_c} \in (0,1) \\
C_d &:= -C_a' \log(C_a') - C_b'(K_A - 1)\log(C_b') - K_B C_c' \log(C_c') \\
C_e &:= \frac{K_A C_b}{K_A C_b + K_B C_c} \in (0,1) \\
C_f &:= \frac{K_B C_b}{K_A C_b + K_B C_c} \in (0,1) \\
C_g &:= \frac{K_A C_b + K_B C_c}{C_a + (K_A - 1)C_b + K_B C_c} > 0.
\end{aligned}
$$

Using a similar argument as Theorem 3, we show in Lemma 3 (see the end of the proof), for any feasible solution $(\boldsymbol{W}, \boldsymbol{H})$ of Eq. 16, the objective value of Eq. 16 can be bounded from below by:

$$
\frac{1}{|\mathbb{K}_A^2| n_A} \sum_{k \in \mathbb{K}_A^2} \sum_{i=1}^{n_A} \mathcal{L}(\boldsymbol{W}\boldsymbol{h}_{(a,b),i}, \boldsymbol{y}_{(a,b)})
$$

$$
\overset{a}{\geq} -\frac{C_g}{K_A}\sqrt{KE_H}\sqrt{\sum_{k=1}^{K_A} \|C_e \boldsymbol{w}_A + C_f \boldsymbol{w}_B - \boldsymbol{w}_k\|^2 + C_d}
$$

$$
\overset{b}{\geq} -\frac{C_g}{K_A}\sqrt{KE_H}\sqrt{KE_W - K_A\left(1/K_R - C_f^2 - \frac{C_f^4}{C_e(2 - C_e)}\right)\|\boldsymbol{w}_B\|^2 - \sum_{k=K_A+1}^{K}\|\boldsymbol{w}_k - \boldsymbol{w}_B\|^2 + C_d}
$$

$$
(22)
$$

where $\boldsymbol{w}_A := \frac{1}{K_A}\sum_{k=1}^{K_A} \boldsymbol{w}_k$, $\boldsymbol{w}_B := \frac{1}{K_B}\sum_{k=K_A+1}^{K} \boldsymbol{w}_k$, and $K_R := \frac{K_A}{K_B}$. Moreover, the equality in $\overset{a}{\geq}$ holds only if $\boldsymbol{h}_{k,i} = \boldsymbol{0}_p$ for all $k \in [K_A + 1 : K]$ and $i \in [n_B]$.

Though $C_a$, $C_b$, and $C_c$ can be any positive numbers, we need to carefully pick them to exactly reach the global minimum of Eq. 16. In the following, we separately consider three cases according to the values of $K_A$, $K_B$, and $E_H E_W$.

**(Case 1)** Consider the case when $K_A = 1$. We pick $C_a := \exp\left(\sqrt{K_B(1+K_B)E_H E_W}\right)$, $C_b := 1$, and $C_c := \exp\left(-\sqrt{(1+K_B)E_H E_W/K_B}\right)$.

Then, from $\overset{a}{\geq}$ in Eq. 22, we have

$$\frac{1}{|\mathbb{K}_A^2|n_A} \sum_{k \in \mathbb{K}_A^2} \sum_{i=1}^{n_A} \mathcal{L}(\boldsymbol{W}\boldsymbol{h}_{(a,b),i}, \boldsymbol{y}_{(a,b)})$$

$$\overset{a}{\geq} -C_g C_f \sqrt{KE_H}\sqrt{\|\boldsymbol{w}_1 - \boldsymbol{w}_B\|^2} + C_d$$

$$= -C_g C_f \sqrt{KE_H}\sqrt{\|\boldsymbol{w}_1\|^2 - 2\boldsymbol{w}_1^\mathsf{T}\boldsymbol{w}_B + \|\boldsymbol{w}_B\|^2} + C_d$$

$$\overset{b}{\geq} -C_g C_f \sqrt{KE_H}\sqrt{(1+1/K_B)(\|\boldsymbol{w}_1\|^2 + K_B\|\boldsymbol{w}_B\|^2)} + C_d$$

$$\overset{c}{\geq} -C_g C_f \sqrt{KE_H}\sqrt{(1+1/K_B)\left(KE_W - \sum_{k=2}^{K}\|\boldsymbol{w}_k - \boldsymbol{w}_B\|^2\right)} + C_d$$

$$\overset{c}{\geq} -C_g C_f \sqrt{KE_H}\sqrt{(1+1/K_B)KE_W} + C_d := L_1$$

(23)

where $\overset{a}{\geq}$ uses $C_e + C_f = 1$, $\overset{b}{\geq}$ follows from $-2ab \leq a^2 + b^2$, i.e., $-2\boldsymbol{w}_1^\mathsf{T}\boldsymbol{w}_B \leq (1/K_B)\|\boldsymbol{w}_1\|^2 + K_B\|\boldsymbol{w}_B\|^2$, and $\overset{c}{\geq}$ follows from $\sum_{k=2}^{K}\|\boldsymbol{w}_k\|^2 = K_B\|\boldsymbol{w}_B\|^2 + \sum_{k=2}^{K}\|\boldsymbol{w}_k - \boldsymbol{w}_B\|^2$ and the constraint that $\sum_{k=1}^{K}\|\boldsymbol{w}_k\|^2 \leq KE_W$.

On the other hand, when $(\boldsymbol{M}, \boldsymbol{H})$ satisfies that

$$\boldsymbol{w}_1 = \sqrt{K_B E_W}\boldsymbol{u}, \quad \boldsymbol{w}_k = -\sqrt{\frac{1}{K_B E_W}}\boldsymbol{u}, \quad k \in [2:K],$$

$$\boldsymbol{h}_{1,i} = \sqrt{(1+K_B)E_H}\boldsymbol{u}, \quad i \in [n_A], \quad \boldsymbol{h}_{k,i} = \boldsymbol{0}_p \quad k \in [2:K], \quad i \in [n_B],$$

where $\boldsymbol{u}$ is any unit vector, the inequalities in Eq. 23 reduces to equalities. So, $L_1$ is the global minimum of Eq. 16. Moreover, $L_1$ is achieved only if $\overset{a}{\geq}$ in Eq. 22 reduces to equality. From Lemma 3, we have that any minimizer satisfies that $\boldsymbol{h}_{k,i} = \boldsymbol{0}_p$ for all $k \in [K_A + 1 : K]$ and $i \in [n_B]$.

Finally, for any feasible solution $(\boldsymbol{W}', \boldsymbol{H}')$, if there exist $k, k' \in [K_A + 1 : K]$ such that $\|\boldsymbol{w}_k - \boldsymbol{w}_{k'}\| = \varepsilon > 0$, we have

$$\sum_{k=K_A+1}^{K}\|\boldsymbol{w}_k - \boldsymbol{w}_B\|^2 \geq \|\boldsymbol{w}_k - \boldsymbol{w}_B\|^2 + \|\boldsymbol{w}_{k'} - \boldsymbol{w}_B\|^2 \geq \frac{\|\boldsymbol{w}_k - \boldsymbol{w}_{k'}\|^2}{2} = \varepsilon^2/2. \quad (24)$$

It follows from $\overset{c}{\geq}$ in Eq. 23 that

$$\frac{1}{|\mathbb{K}_A^2|n_A} \sum_{k \in \mathbb{K}_A^2} \sum_{i=1}^{n_A} \mathcal{L}(\boldsymbol{W}\boldsymbol{h}_{(a,b),i}, \boldsymbol{y}_{(a,b)})$$

$$\geq -C_g C_f \sqrt{KE_H}\sqrt{(1+1/K_B)(KE_W - \varepsilon^2/2)} + C_d := L_1 + \varepsilon_1, \quad (25)$$

with $\varepsilon_1 > 0$ depending on $\varepsilon$, $K_A$, $K_B$, $E_H$, and $E_W$.

**(Case 2)** Consider the case when $K_A > 1$ and $\exp\left((1+1/K_R)\sqrt{E_H E_W}/(K_A - 1)\right) < \sqrt{1 + K_R} + 1$. Let us pick $C_a := \exp\left((1+1/K_R)\sqrt{E_H E_W}\right)$, $C_a := \exp\left(-\frac{1}{K_A-1}(1+1/K_R)\sqrt{E_H E_W}\right)$, and $C_c := 1$.

Following from $\overset{b}{\geq}$ in Eq. 22, we know if $1/K_R - C_f^2 - \frac{C_f^4}{C_e(2-C_e)} > 0$, then

$$\frac{1}{|\mathbb{K}_A^2|n_A} \sum_{k \in \mathbb{K}_A^2} \sum_{i=1}^{n_A} \mathcal{L}(\boldsymbol{W}\boldsymbol{h}_{(a,b),i}, \boldsymbol{y}_{(a,b)}) \geq -C_g(1+1/K_R)\sqrt{E_H E_W} + C_d := L_2 \qquad (26)$$

In fact, we do have $1/K_R - C_f^2 - \frac{C_f^4}{C_e(2-C_e)} > 0$ because

$$1/K_R > C_f^2 + \frac{C_f^4}{C_e(2-C_e)}$$

$$\overset{a}{\Longleftrightarrow} C_f < \sqrt{\frac{1}{1+K_R}}$$

$$\overset{b}{\Longleftrightarrow} \frac{C_b}{C_c} > \frac{1}{\sqrt{1+K_R}+1}$$

$$\Longleftrightarrow \exp\left((1+1/K_R)\sqrt{E_H E_W}/(K_A-1)\right) < \sqrt{1+K_R}+1.$$

where in $\overset{a}{\Longleftrightarrow}$, $C_e + C_f = 1$, and in $\overset{b}{\Longleftrightarrow}$, $C_f = \frac{K_B C_c}{K_A C_b + K_B C_c}$.

On the other hand, when $(\boldsymbol{W}, \boldsymbol{H})$ satisfies that

$$[\boldsymbol{w}_1, ..., \boldsymbol{w}_{K_A}] = \sqrt{\frac{E_W}{E_H}} [\boldsymbol{h}_1, ...\boldsymbol{h}_{K_A}]^\intercal = \sqrt{(1+1/K_R)E_W}(\boldsymbol{M}_A^*)^\intercal,$$

$$\boldsymbol{h}_{k,i} = \boldsymbol{h}_k, \quad k \in [K_A], \ i \in [n_A],$$

$$\boldsymbol{h}_{k,i} = \boldsymbol{w}_k = \boldsymbol{0}_p, \quad k \in [K_A+1:K], \ i \in [n_B],$$

where $(\boldsymbol{M}_A^*)$ is a $K_A$-simplex ETF, Eq. 26 reduces to equality. So, $L_2$ is the global minimum of Eq. 16. Moreover, $L_2$ is achieved only if $\overset{a}{\geq}$ in Eq. 22 reduces to equality. From Lemma 3, we have that any minimizer satisfies that $\boldsymbol{h}_{k,i} = \boldsymbol{0}_p$ for all $k \in [K_A+1:K]$ and $i \in [n_B]$.

Finally, for any feasible solution $(\boldsymbol{W}', \boldsymbol{H}')$, if there exist $k, k' \in [K_A+1:K]$ such that $\|\boldsymbol{w}_k - \boldsymbol{w}_{k'}\| = \varepsilon > 0$, plugging Eq. 24 into $\overset{b}{\geq}$ in Eq. 22, we have

$$\frac{1}{|\mathbb{K}_A^2|n_A} \sum_{k \in \mathbb{K}_A^2} \sum_{i=1}^{n_A} \mathcal{L}(\boldsymbol{W}\boldsymbol{h}_{(a,b),i}, \boldsymbol{y}_{(a,b)})$$

$$\geq -\frac{C_g}{K_A}\sqrt{KE_H}\sqrt{KE_W - \varepsilon^2/2)} + C_d := L_2 + \varepsilon_2, \qquad (27)$$

with $\varepsilon_2 > 0$ depending on $\varepsilon$, $K_A$, $K_B$, $E_H$, and $E_W$.

**(Case 3)** Consider the case when $K_A > 1$ and $\exp\left((1+1/K_R)\sqrt{E_H E_W}/(K_A-1)\right) \geq \sqrt{1+K_R}+1$. Let $C_f' := \frac{1}{\sqrt{1+K_R}}$ and $C_e' = 1 - C_f'$. For $x \in [0,1]$, we define:

$$g_N(x) := \sqrt{\frac{(1+K_R)E_W}{K_R x^2 + K_R(1+K_R)(1-x)^2}},$$

$$g_a(x) := \exp\left(\frac{g_N(x)\sqrt{(1+K_R)E_H/K_R}}{x^2 + \left(1+\frac{C_e'}{C_f'}\right)^2(1-x)^2}\left[x^2 + \left(1+\frac{C_e'}{C_f'}\right)(1-x)^2\right]\right),$$

$$g_b(x) := \exp\left(\frac{g_N(x)\sqrt{(1+K_R)E_H/K_R}}{x^2 + \left(1+\frac{C_e'}{C_f'}\right)^2(1-x)^2}\left[-\frac{1}{K_A-1}x^2 + \left(1+\frac{C_e'}{C_f'}\right)(1-x)^2\right]\right),$$

$$g_b(x) := \exp\left(\frac{g_N(x)\sqrt{(1+K_R)E_H/K_R}}{x^2 + \left(1+\frac{C_e'}{C_f'}\right)^2(1-x)^2}\left[-\left(1+\frac{C_e'}{C_f'}\right)K_R(1-x)^2\right]\right),$$

Let $x_0 \in [0, 1]$ be a root of the equation

$$g_b(x)/g_c(x) = \frac{1/C_f' - 1}{K_R}.$$

We first show that the solution $x_0$ exists. First of all, one can directly verify then $x \in [0, 1]$, $g_b(x)/g_c(x)$ is continuous. It suffices to prove that (A) $g_b(0)/g_c(0) \geq \frac{1/C_f'-1}{K_R}$ and (B) $g_b(1)/g_c(1) \leq \frac{1/C_f'-1}{K_R}$.

(A) When $x = 0$, we have $g_b(x)/g_c(x) \geq \exp(0) = 1$. At the same time, $\frac{1/C_f'-1}{K_R} = \frac{\sqrt{1+K_R}-1}{K_R} = \frac{1}{\sqrt{1+K_R}+1} \leq 1$. Thus, $g_b(0)/g_c(0) \geq \exp(0) = 1 \geq \frac{1/C_f'-1}{K_R}$ is achieved.

(B) When $x = 1$, we have $g_N(1) = \sqrt{(1 + 1/K_R)E_W}$. So,

$$g_b(1)/g_c(1) = \exp\left(-(1 + 1/K_R)\sqrt{E_H E_W}/(K_A - 1)\right) \overset{a}{\leq} \frac{1}{\sqrt{1 + K_R} + 1} = \frac{1/C_f' - 1}{K_R},$$

where $\overset{a}{\leq}$ is obtained by the condition that

$$\exp\left((1 + 1/K_R)\sqrt{E_H E_W}/(K_A - 1)\right) \geq \sqrt{1 + K_R} + 1.$$

Now, we pick $C_a := g_a(x_0)$, $C_b := g_b(x_0)$, and $C_c := g_c(x_0)$, because $\frac{C_b}{C_c} = \frac{1/C_f'-1}{K_R}$, we have $C_e = C_e'$, $C_f = C_f'$, and $1/K_R = C_f^2 + \frac{C_f^4}{C_e(2-C_e)}$. Then, it follows from $\overset{b}{\geq}$ in Eq. 22 that

$$\frac{1}{|\mathbb{K}_A^2|n_A} \sum_{k \in \mathbb{K}_A^2} \sum_{i=1}^{n_A} \mathcal{L}(\boldsymbol{W}\boldsymbol{h}_{(a,b),i}, \boldsymbol{y}_{(a,b)}) \geq -C_g(1 + 1/K_R)\sqrt{E_H E_W} + C_d := L_2. \tag{28}$$

On the other hand, consider the solution $(\boldsymbol{W}, \boldsymbol{H})$ that satisfies

$$\boldsymbol{w}_k = g_N(x_0)\boldsymbol{P}_A\left[\frac{x_0}{\sqrt{(K_A - 1)K_A}}(K_A\boldsymbol{y}_k - \mathbf{1}_{K_A} + \frac{1 - x_0}{\sqrt{K_A}}\mathbf{1}_{K_A}\right], \quad k \in [K_A],$$

$$\boldsymbol{w}_k = -\frac{C_e(2 - C_e)}{C_f^2 K_A}\boldsymbol{P}_A\sum_{k=1}^{K_A}\boldsymbol{w}_k, \quad k \in [K_A + 1 : K],$$

$$\boldsymbol{h}_{k,i} = \frac{\sqrt{(1 + 1/K_R)E_H}}{\|\boldsymbol{w}_i + \frac{C_e}{C_f K_A}\sum_{k=1}^{K_A}\boldsymbol{w}_k\|}\boldsymbol{P}_A\left[\boldsymbol{w}_i\frac{C_e}{C_f K_A}\sum_{k=1}^{K_A}\boldsymbol{w}_k\right], \quad k \in [K_A], \, i \in [n_A]$$

$$\boldsymbol{h}_{k,i} = \mathbf{0}_p, \quad k \in [K_A + 1 : K], \, i \in [n_B],$$

where $\boldsymbol{y}_k \in \mathbb{R}^K$ is the one-hot vector of the $k$-th class label and $\boldsymbol{P}_A \in \mathbb{R}^{p \times K_A}$ is a partial orthogonal matrix such that $\boldsymbol{P}_A^\top\boldsymbol{P}_A = \boldsymbol{I}_{K_A}$. We have $\exp(\boldsymbol{h}_{k,i}^\top\boldsymbol{w}_k) = g_a(x_0)$ for $i \in [n_A]$ and $k \in [K_A]$, $\exp(\boldsymbol{h}_{k,i}^\top\boldsymbol{w}_{k'}) = g_b(x_0)$ for $i \in [n_A]$ and $k, k' \in [K_A]$ such that $k \neq k'$, and $\exp(\boldsymbol{h}_{k,i}^\top\boldsymbol{w}_{k'}) = g_c(x_0)$ for $i \in [n_A]$, $k \in [K_A]$, and $k' \in [K_A + 1 : K]$. Moreover, $(\boldsymbol{W}, \boldsymbol{H})$ can achieve the equality in Eq. 28. Finally, following the same argument as (Case 2), we have that (1) $L_2$ is the global minimum of Eq. 16; (2) any minimizer satisfies that $\boldsymbol{h}_{k,i} = \mathbf{0}_p$ for all $k \in [K_A + 1 : K]$ and $i \in [n_B]$; (3) for any feasible solution $(\boldsymbol{W}', \boldsymbol{H}')$, if there exist $k, k' \in [K_A + 1 : K]$ such that $\|\boldsymbol{w}_k - \boldsymbol{w}_{k'}\| = \varepsilon > 0$, then Eq. 26 holds.

Combining the three cases, we obtain Lemma 2, completing the proof. $\qquad\square$

**Lemma 3.** *For any constants $C_a > 0$, $C_b > 0$, and $C_c > 0$, define*

$$C_a' := \frac{C_a}{C_a + (K_A - 1)C_b + K_B C_c} \in (0, 1)$$

$$C_b' := \frac{C_b}{C_a + (K_A - 1)C_b + K_B C_c} \in (0, 1)$$

$$C_c' := \frac{C_c}{C_a + (K_A - 1)C_b + K_B C_c} \in (0, 1)$$

$$C_d := -C_a' \log(C_a') - C_b'(K_A - 1) \log(C_b') - K_B C_c' \log(C_c')$$

$$C_e := \frac{K_A C_b}{K_A C_b + K_B C_c} \in (0, 1)$$

$$C_f := \frac{K_B C_b}{K_A C_b + K_B C_c} \in (0, 1)$$

$$C_g := \frac{K_A C_b + K_B C_c}{C_a + (K_A - 1)C_b + K_B C_c} > 0.$$

*For any feasible solution $(\boldsymbol{W}, \boldsymbol{H})$ of Eq. 16, the objective value of Eq. 16 can be bounded from below by:*

$$\frac{1}{|\mathbb{K}_A^2| n_A} \sum_{k \in \mathbb{K}_A^2} \sum_{i=1}^{n_A} \mathcal{L}(\boldsymbol{W}\boldsymbol{h}_{(a,b),i}, \boldsymbol{y}_{(a,b)})$$

$$\overset{a}{\geq} -\frac{C_g}{K_A} \sqrt{K E_H} \sqrt{\sum_{k=1}^{K_A} \|C_e \boldsymbol{w}_A + C_f \boldsymbol{w}_B - \boldsymbol{w}_k\|^2 + C_d}$$

$$\overset{b}{\geq} -\frac{C_g}{K_A} \sqrt{K E_H} \sqrt{K E_W - K_A \left(1/K_R - C_f^2 - \frac{C_f^4}{C_e(2 - C_e)}\right) \|\boldsymbol{w}_B\|^2 - \sum_{k=K_A+1}^{K} \|\boldsymbol{w}_k - \boldsymbol{w}_B\|^2 + C_d}$$

$$(29)$$

*where $\boldsymbol{w}_A := \frac{1}{K_A} \sum_{k=1}^{K_A} \boldsymbol{w}_k$, $\boldsymbol{w}_B := \frac{1}{K_B} \sum_{k=K_A+1}^{K} \boldsymbol{w}_k$, and $K_R := \frac{K_A}{K_B}$. Moreover, the equality in $\overset{a}{\geq}$ holds only if $\boldsymbol{h}_{k,i} = \boldsymbol{0}_p$ for all $k \in [K_A + 1 : K]$ and $i \in [n_B]$.*

*Remark 3. Note that the case $\boldsymbol{h}_{k,i} = \boldsymbol{0}_p$ does not imply that network activations all die for the classes $k \in [K_A + 1 : K]$. This is because our analysis does not include the bias term for simplicity.*

*Proof of Lemma 3.* For $(a, b) \in \mathbb{K}_A^2$ and $i \in [n_{(a,b)}]$, we introduce $\boldsymbol{z}_{(a,b),i} = \boldsymbol{W}\boldsymbol{h}_{(a,b),i}^{\lambda}$. Because that $C_a' + (K_A - 1)C_b' + K_B C_c' = 1$, $C_a' > 0$, $C_b' > 0$, and $C_c' > 0$, by the concavity of $\log(\cdot)$, we

have

$$
- \lambda \log \left( \frac{\exp(\boldsymbol{z}_{(a,b),i}(a))}{\sum_{k'=1}^{K} \exp(\boldsymbol{z}_{(a,b),i}(k'))} \right) - (1 - \lambda) \log \left( \frac{\exp(\boldsymbol{z}_{(a,b),i}(b))}{\sum_{k'=1}^{K} \exp(\boldsymbol{z}_{(a,b),i}(k'))} \right)
$$

$$
= - \lambda \boldsymbol{z}_{(a,b),i}(a) - (1 - \lambda)\boldsymbol{z}_{(a,b),i}(b)
$$

$$
+ \lambda \log \left( C_a' \left( \frac{\exp(\boldsymbol{z}_{(a,b),i}(a))}{C_a'} \right) + \sum_{k'=1, k' \neq a}^{K_A} C_b' \left( \frac{\exp(\boldsymbol{z}_{k,i}(k'))}{C_b'} \right) + \sum_{k'=K_A+1}^{K} C_c' \left( \frac{\exp(\boldsymbol{z}_{k,i}(k'))}{C_c'} \right) \right)
$$

$$
+ (1 - \lambda) \log \left( C_a' \left( \frac{\exp(\boldsymbol{z}_{(a,b),i}(b))}{C_a'} \right) + \sum_{k'=1, k' \neq b}^{K_A} C_b' \left( \frac{\exp(\boldsymbol{z}_{k,i}(k'))}{C_b'} \right) + \sum_{k'=K_A+1}^{K} C_c' \left( \frac{\exp(\boldsymbol{z}_{k,i}(k'))}{C_c'} \right) \right)
$$

$$
\geq - \lambda \boldsymbol{z}_{(a,b),i}(a) - (1 - \lambda)\boldsymbol{z}_{(a,b),i}(b) + C_a' \left( \lambda \boldsymbol{z}_{(a,b),i}(a) + (1 - \lambda)\boldsymbol{z}_{(a,b),i}(b) \right)
$$

$$
+ C_b' \left( \lambda \sum_{k'=1, k' \neq a}^{K_A} \boldsymbol{z}_{(a,b),i}(k') + (1 - \lambda) \sum_{k'=1, k' \neq b}^{K_A} \boldsymbol{z}_{(a,b),i}(k') \right) + C_c' \sum_{k'=K_A+1}^{K} \boldsymbol{z}_{(a,b),i}(k') + C_d
$$

$$
= C_g C_e \left( \frac{1}{K_A} \sum_{k'=1}^{K_A} \boldsymbol{z}_{(a,b),i}(k') - \lambda \boldsymbol{z}_{(a,b),i}(a) - (1 - \lambda)\boldsymbol{z}_{(a,b),i}(b) \right)
$$

$$
+ C_g C_f \left( \frac{1}{K_B} \sum_{k'=K_A+1}^{K} \boldsymbol{z}_{(a,b),i}(k') - \lambda \boldsymbol{z}_{(a,b),i}(a) - (1 - \lambda)\boldsymbol{z}_{(a,b),i}(b) \right) + C_d.
$$

$$(30)$$

Therefore, integrating Eq. 30 with $(a, b) \in \mathbb{K}_A^2$ and $i \in [n_A]$, recalling that $\boldsymbol{w}_A = \frac{1}{K_A} \sum_{k=1}^{K_A} \boldsymbol{w}_k$ and $\boldsymbol{w}_B = \frac{1}{K_B} \sum_{k=K_A+1}^{K} \boldsymbol{w}_k$, we have

$$
\frac{1}{|\mathbb{K}_A^2| n_A} \sum_{(a,b) \in \mathbb{K}_A^2} \sum_{i=1}^{n_A} \mathcal{L}(\boldsymbol{W} \boldsymbol{h}_{(a,b),i}^{\lambda}, \boldsymbol{y}_{(a,b)})
$$

$$
\geq \frac{1}{|\mathbb{K}_A^2| n_A} \sum_{(a,b) \in \mathbb{K}_A^2} \sum_{i=1}^{n_A} C_g \left( \begin{array}{c} C_e \left( \lambda(\boldsymbol{h}_{a,i}\boldsymbol{w}_A - \boldsymbol{h}_{a,i}\boldsymbol{w}_a) + (1 - \lambda)(\boldsymbol{h}_{b,i}\boldsymbol{w}_A - \boldsymbol{h}_{b,i}\boldsymbol{w}_b) \right) \\ + C_f \left( \lambda(\boldsymbol{h}_{a,i}\boldsymbol{w}_B - \boldsymbol{h}_{a,i}\boldsymbol{w}_a) + (1 - \lambda)(\boldsymbol{h}_{b,i}\boldsymbol{w}_B - \boldsymbol{h}_{b,i}\boldsymbol{w}_b) \right) \end{array} \right)
$$

$$
\overset{a}{=} \frac{1}{K_A n_A} \sum_{k=1}^{K_A} \sum_{i=1}^{n_A} Cg \left[ C_e (\boldsymbol{h}_{k,i}\boldsymbol{w}_A - \boldsymbol{h}_{k,i}\boldsymbol{w}_k) + C_f (\boldsymbol{h}_{k,i}\boldsymbol{w}_B - \boldsymbol{h}_{k,i}\boldsymbol{w}_k) \right] + C_d
$$

$$
\overset{b}{=} \frac{C_g}{K_A} \sum_{k=1}^{K_A} \boldsymbol{h}_{k,i}^{\mathsf{T}} (C_e \boldsymbol{w}_A + C_f \boldsymbol{w}_B - \boldsymbol{w}_k) + C_d, \tag{31}
$$

where in $\overset{a}{=}$, we use $\sum_{(a,b) \in \mathbb{K}_A^2} \boldsymbol{h}_{(a,b),i}^{\lambda} = K_A \sum_{k=1}^{K_A} \boldsymbol{h}_{k,i}$, and in $\overset{b}{=}$, we introduce $\boldsymbol{h}_k := \frac{1}{n_k} \sum_{i=1}^{n_k} \boldsymbol{h}_{k,i}$ for $k \in [K]$ and use $C_e + C_f = 1$. Then, it is sufficient to bound $\sum_{k=1}^{K_A} \boldsymbol{h}_k^{\mathsf{T}} (C_e \boldsymbol{w}_A + C_f \boldsymbol{w}_B - \boldsymbol{w}_k)$. By the Cauchy-Schwarz inequality, we have

$$
\sum_{k=1}^{K_A} \boldsymbol{h}_k^{\mathsf{T}} (C_e \boldsymbol{w}_A + C_f \boldsymbol{w}_B - \boldsymbol{w}_k) \geq - \sqrt{\sum_{k=1}^{K_A} \|\boldsymbol{h}_k\|^2} \sqrt{\sum_{k=1}^{K_A} \|C_e \boldsymbol{w}_A + C_f \boldsymbol{w}_B - \boldsymbol{w}_k\|^2}
$$

$$
\overset{a}{\geq} - \sqrt{\sum_{k=1}^{K_A} \frac{1}{n_k} \sum_{i=1}^{n_k} \|\boldsymbol{h}_{k,i}\|^2} \sqrt{\sum_{k=1}^{K_A} \|C_e \boldsymbol{w}_A + C_f \boldsymbol{w}_B - \boldsymbol{w}_k\|^2}
$$

$$
\overset{b}{\geq} - \sqrt{K E_H} \sqrt{\sum_{k=1}^{K_A} \|C_e \boldsymbol{w}_A + C_f \boldsymbol{w}_B - \boldsymbol{w}_k\|^2}, \tag{32}
$$

where $\overset{a}{\geq}$ follows from Jensen's inequality $\|\boldsymbol{h}_k\|^2 \leq frac1n_k \sum_{i=1}^{n_k}\|\boldsymbol{h}_{k,i}\|^2$ for $k \in [K_A]$, and $\overset{b}{\geq}$ uses the constraint that $\sum_{k=1}^{K} \frac{1}{n_k} \sum_{i=1}^{n_k}\|\boldsymbol{h}_{k,i}\|^2 \leq E_H$. Moreover, we have $\sum_{k=1}^{K} \frac{1}{n_k} \sum_{i=1}^{n_k}\|\boldsymbol{h}_{k,i}\|^2 = E_H$ only if $\boldsymbol{h}_{k,i} = \boldsymbol{0}_p$ for all $k \in [K_A + 1, K]$. Plugging Eq. 32 to Eq. 31, we obtain $\overset{a}{\geq}$ in Eq. 29.

We then bound $\sum_{k=1}^{K_A}\|C_e\boldsymbol{w}_A + C_f\boldsymbol{w}_B - \boldsymbol{w}_k\|^2$. First, we have

$$\frac{1}{K_A} \sum_{k=1}^{K_A}\|C_e\boldsymbol{w}_A + C_f\boldsymbol{w}_B - \boldsymbol{w}_k\|^2$$

$$=\frac{1}{K_A} \sum_{k=1}^{K_A}\|\boldsymbol{w}_k\|^2 - 2\frac{1}{K_A} \sum_{k=1}^{K_A} \boldsymbol{w}_k \cdot (C_e\boldsymbol{w}_A + C_f\boldsymbol{w}_B) + \|C_e\boldsymbol{w}_A + C_f\boldsymbol{w}_B\|^2$$

$$\overset{a}{=}\frac{1}{K_A} \sum_{k=1}^{K_A}\|\boldsymbol{w}_k\|^2 - 2C_f^2\boldsymbol{w}_A^\intercal\boldsymbol{w}_B - C_e(2 - C_e)\|\boldsymbol{w}_A\|^2 + C_f^2\|\boldsymbol{w}_B\|^2$$

$$\tag{33}$$

where $\overset{a}{=}$ uses $\sum_{k=1}^{K_A} \boldsymbol{w}_A = K_A\boldsymbol{w}_A$. Then, using the constraint that $\sum_{k=1}^{K_A}\|\boldsymbol{w}_A\|^2 \leq KE_W$ yields that

$$\frac{1}{K_A} \sum_{k=1}^{K_A}\|\boldsymbol{w}_k\|^2 - 2C_f^2\boldsymbol{w}_A^\intercal\boldsymbol{w}_B - C_e(2 - C_e)\|\boldsymbol{w}_A\|^2 + C_f^2\|\boldsymbol{w}_B\|^2$$

$$\leq\frac{K}{K_A}E_W - \frac{1}{K_A} \sum_{k=K_A+1} K\|\boldsymbol{w}_k\|^2 - C_e(2 - C_e)\|\boldsymbol{w}_A + \frac{C_f^2}{C_e(2 - C_e)}\boldsymbol{w}_B\|^2 + \left(C_f^2 + \frac{C_f^4}{C_e(2 - C_e)}\right)\|\boldsymbol{w}_B\|^2$$

$$\overset{a}{=}\frac{K}{K_A}E_W - \left(1/K_R - C_f^2 - \frac{C_f^4}{C_e(2 - C_e)}\right)\|w_B\|^2 - \frac{1}{K_A} \sum_{k=K_A+1}^{K}\|\boldsymbol{w}_k - \boldsymbol{w}_B\|^2,$$

$$\tag{34}$$

where $\overset{a}{=}$ applies $\sum_{k=K_A+1} K\|\boldsymbol{w}_k\|^2 = K_B\|\boldsymbol{w}_B\|^2 + \sum_{k=K_A+1} K\|\boldsymbol{w}_k - \boldsymbol{w}_B\|^2$. Plugging Eq. 33 and Eq. 34 into $\overset{a}{\geq}$ in Eq. 29, we obtain $\overset{b}{\geq}$ in Eq. 29, completing the proof. $\square$

## C.4 PROOF OF THEOREM 3

**Definition 1.** *A K-simplex ETF is a collection of points in $\mathbb{R}^p$ specified by the columns of the matrix*

$$M^\star = \sqrt{\frac{K}{K-1}} P \left( I_K - \frac{1}{K} 1_K 1_K^\mathsf{T} \right)$$

*where $I_K \in \mathbb{R}^{K \times K}$ is the identity matrix, $1_K$ is the ones vector, and $P \in \mathbb{R}^{p \times K}$ ($p \geq K$) is a partial orthogonal matrix such that $P^\mathsf{T} P = I_K$.*

**Theorem 3.** *In the balanced case, although $(W^\lambda, H^\lambda)$ are linearly dependent on $(W, H)$ in Eq. 9, any global minimizer $W^\star \equiv [w_1^\star, ..., w_K^\star]$, $H^\star \equiv \left[ h_{k,i}^\star : 1 \leq k \leq K, 1 \leq i \leq n \right]$ of Eq. 9 with the cross-entropy loss obeys*

$$h_{k,i}^\star = C w_k^\star = C' m_k^\star \tag{35}$$

*for all $1 \leq i \leq n$, $1 \leq k \leq K$, where the constants $C = \sqrt{E_H/E_W}$, $C' = \sqrt{E_H}$, and the matrix $[m_1^\star, ..., m_K^\star]$ forms a K-simplex ETF specified in Definition 1*

Because there are multiplication of variables in the objective functions, Eq. 9 is non-convex. Thus, the KKT condition is not sufficient for optimality. To prove Theorem 3, we directly determine the global minimum of Eq. 9. During this procedure, one key step is to show that minimizing Eq. 9 is equivalent to minimize a symmetric quadratic function:

$$\sum_{i=1}^n \left[ \left( \sum_{k \in \mathbb{K}^2} h_{k,i}^\lambda \right)^\mathsf{T} \left( \sum_{k \in \mathbb{K}^2} w_k^\lambda \right) - K^2 \sum_{k \in \mathbb{K}^2} h_{k,i}^{\lambda \mathsf{T}} w_k^\lambda \right]$$

under suitable conditions. The detail is shown below.

*Proof.* By the concavity of $\log(\cdot)$, for any $z \in \mathbb{R}^{K^2}$, $k \in [K^2]$, constants $C_a$, $C_b > 0$, letting $C_c = \frac{C_b}{(C_a + C_b)(K^2 - 1)}$, we have

$$-\log \left( \frac{z(k)}{\sum_{k'=1}^{K^2} z(k')} \right) = -\log(z(k)) + \log \left( \sum_{k'=1}^{K^2} z(k') \right)$$

$$= -\log(z(k)) + \log \left( \frac{C_a}{C_a + C_b} \left( \frac{(C_a + C_b) z(k)}{C_a} \right) + C_c \sum_{k'=1, k' \neq k}^{K^2} \frac{z(k')}{C_c} \right). \tag{36}$$

Recognizing the equality

$$\frac{C_a}{C_a + C_b} + \underbrace{C_c + \cdots + C_c}_{K^2 - 1} = \frac{C_a}{C_a + C_b} + (K^2 - 1) \frac{C_b}{(C_a + C_b)(K^2 - 1)} = 1$$

and the concavity of $\log(\cdot)$, we see that the Jensen inequality gives

$$\log \left( \frac{C_a}{C_a + C_b} \left( \frac{(C_a + C_b) z(k)}{C_a} \right) + C_c \sum_{k'=1, k' \neq k}^{K^2} \frac{z(k')}{C_c} \right)$$

$$\geq \frac{C_a}{C_a + C_b} \log \left( \frac{(C_a + C_b) z(k)}{C_a} \right) + C_c \sum_{k'=1, k' \neq k}^{K^2} \log \left( \frac{z(k')}{C_c} \right). \tag{37}$$

Plugging this inequality into Eq. 36, we get

$$-\log \left( \frac{z(k)}{\sum_{k'=1}^{K^2} z(k')} \right) \geq -\log(z(k)) + \frac{C_a}{C_a + C_b} \log \left( \frac{(C_a + C_b) z(k)}{C_a} \right) + C_c \sum_{k'=1, k' \neq k}^{K^2} \log \left( \frac{z(k')}{C_c} \right)$$

$$= -\frac{C_a}{C_a + C_b} \left[ \log(z(k)) - \frac{1}{K^2 - 1} \sum_{k'=1, k' \neq k}^{K^2} \log(z(k')) \right] + C_d,$$

where the constant $C_d := \frac{C_a}{C_a+C_b} \log\left(\frac{C_a+C_b}{C_a}\right) + \frac{C_b}{C_a+C_b} \log(1/C_c)$. Note that in Eq. 36, $C_a$ and $C_b$ can be any positive numbers. To prove Theorem 3, we set $C_a := \exp(\sqrt{E_H E_W})$ and $C_b := \exp(-\sqrt{E_H E_W}/(K^2-1))$, which shall lead to the tightest lower bound for the objective of Eq. 9. Applying Eq. 36 to the objective, we have

$$\frac{1}{N} \sum_{k \in \mathbb{K}^2} \sum_{i=1}^{n} \mathcal{L}(\boldsymbol{W}^\lambda \boldsymbol{h}_{k,i}^\lambda, \boldsymbol{y}_k^\lambda)$$

$$\geq \frac{C_b}{(C_a + C_b)N(K^2-1)} \sum_{i=1}^{n} \left[ \left( \sum_{k \in \mathbb{K}^2} \boldsymbol{h}_{k,i}^\lambda \right)^\mathsf{T} \left( \sum_{k \in \mathbb{K}^2} \boldsymbol{w}_k^\lambda \right) - K^2 \sum_{k \in \mathbb{K}^2} \boldsymbol{h}_{k,i}^{\lambda\mathsf{T}} \boldsymbol{w}_k^\lambda \right] + C_d. \quad (38)$$

Defining $\bar{\boldsymbol{h}}_i^\lambda := \frac{1}{K^2} \sum_{k \in \mathbb{K}^2} \boldsymbol{h}_{k,i}^\lambda$ for $i \in [n]$, it follows from the simple inequality $2ab \leq a^2 + b^2$ that

$$\sum_{i=1}^{n} \left[ \left( \sum_{k \in \mathbb{K}^2} \boldsymbol{h}_{k,i}^\lambda \right)^\mathsf{T} \left( \sum_{k \in \mathbb{K}^2} \boldsymbol{w}_k^\lambda \right) - K^2 \sum_{k \in \mathbb{K}^2} \boldsymbol{h}_{k,i}^{\lambda\mathsf{T}} \boldsymbol{w}_k^\lambda \right]$$

$$= K^2 \sum_{i=1}^{n} \sum_{k \in \mathbb{K}^2} (\bar{\boldsymbol{h}}_i^\lambda - \boldsymbol{h}_{k,i}^\lambda)^\mathsf{T} \boldsymbol{w}_k^\lambda$$

$$\geq -\frac{K^2}{2} \sum_{k \in \mathbb{K}^2} \sum_{i=1}^{n} \|\bar{\boldsymbol{h}}_i^\lambda - \boldsymbol{h}_{k,i}^\lambda\|^2 / C_e - \frac{C_e N}{2} \sum_{k \in \mathbb{K}^2} \|\boldsymbol{w}_k^\lambda\|^2, \quad (39)$$

where we pick $C_e := \sqrt{E_H/E_W}$. The two terms in the right hand side of Eq. 39 can be bounded via the constrains of Eq. 9. Specifically, we have

$$\frac{C_e N}{2} \sum_{k \in \mathbb{K}^2} \|\boldsymbol{w}_k^\lambda\|^2 \leq \frac{K^2 N \sqrt{E_H E_W}}{2}, \quad (40)$$

and

$$\frac{K^2}{2} \sum_{k \in \mathbb{K}^2} \sum_{i=1}^{n} \|\bar{\boldsymbol{h}}_i^\lambda - \boldsymbol{h}_{k,i}^\lambda\|^2 / C_e \overset{a}{=} \frac{K^4}{2C_e} \sum_{i=1}^{N} \left( \frac{1}{K^2} \sum_{k \in \mathbb{K}^2} \|\boldsymbol{h}_{k,i}^\lambda\|^2 - \|\bar{\boldsymbol{h}}_i^\lambda\|^2 \right)$$

$$\leq \frac{K^2}{2C_e} \sum_{k \in \mathbb{K}^2} \sum_{i=1}^{N} \|\boldsymbol{h}_{k,i}^\lambda\|^2 \leq \frac{K^2 N \sqrt{E_H E_W}}{2}, \quad (41)$$

where $\overset{a}{=}$ uses the fact that $\mathbb{E}\|\boldsymbol{a} - \mathbb{E}[\boldsymbol{a}]\|^2 = \mathbb{E}\|\boldsymbol{a}\|^2 - \|\mathbb{E}[\boldsymbol{a}]\|^2$. Thus, plugging Eq. 39, Eq. 40, and Eq. 41 into Eq. 38, we have

$$\frac{1}{N} \sum_{k \in \mathbb{K}^2} \sum_{i=1}^{n} \mathcal{L}(\boldsymbol{W}^\lambda \boldsymbol{h}_{k,i}^\lambda, \boldsymbol{y}_k^\lambda) \geq -\frac{C_b}{(C_a+C_b)} \frac{K^2 \sqrt{E_H E_W}}{K^2-1} + C_d := L_0. \quad (42)$$

Now, we check the conditions that reduce Eq. 42 to an equality.

By the strict concavity of $\log(\cdot)$, Eq. 37 reduces to an equality only if

$$\frac{(C_a + C_b)\boldsymbol{z}(k)}{C_a} = \frac{\boldsymbol{z}(k')}{C_c}$$

for $k' \neq k$. Therefore, Eq. 38 reduces to an equality only if

$$\frac{(C_a + C_b)\boldsymbol{h}_{k,i}^{\lambda\mathsf{T}} \boldsymbol{w}_k^\lambda}{C_a} = \frac{\boldsymbol{h}_{k,i}^{\lambda\mathsf{T}} \boldsymbol{w}_{k'}^\lambda}{C_c}.$$

Recognizing $C_c = \frac{C_b}{(C_a+C_b)(K^2-1)}$ and taking the logarithm of both sides of the above equation, we obtain

$$\boldsymbol{h}_{k,i}^{\lambda\mathsf{T}} \boldsymbol{w}_k^\lambda = \boldsymbol{h}_{k,i}^{\lambda\mathsf{T}} \boldsymbol{w}_{k'}^\lambda + \log\left( \frac{C_a(K^2-1)}{C_b} \right),$$

for all $(k, i, k') \in \{(k, i, k') : k \in \mathbb{K}^2, k' \in \mathbb{K}^2, k' \neq k, i \in [n]\}$. Eq. 39 becomes equality if and only if

$$\bar{h}_i^{\lambda} - h_{k,i}^{\lambda} = -C_e w_k^{\lambda}, \quad k \in \mathbb{K}^2, \; i \in [n].$$

By the definition of $w_k^{\lambda}$ and $h_{k,i}^{\lambda}$, $\bar{h}_i^{\lambda} = \frac{1}{K^2} \sum_{k \in \mathbb{K}^2} h_{k,i}^{\lambda} = \frac{1}{K} \sum_{k=1}^{K} h_{k,i}$. Defining $\bar{h}_i := \frac{1}{K} \sum_{k=1}^{K} h_{k,i}$ and plugging this equality into the above equation, we get

$$\bar{h}_i - \lambda h_{a,i} - (1 - \lambda) h_{b,i} = -C_e (\lambda w_a + (1 - \lambda) w_b) \tag{43}$$

For $a \neq b \neq c$, we define

$$\bar{h}_i - \lambda h_{a,i} - (1 - \lambda) h_{c,i} = -C_e (\lambda w_a + (1 - \lambda) w_c) \tag{44}$$

$$\bar{h}_i - \lambda h_{b,i} - (1 - \lambda) h_{c,i} = -C_e (\lambda w_b + (1 - \lambda) w_c) \tag{45}$$

From the sum of Eq. 44 and Eq. 45, we get

$$h_{a,i} = h_{b,i} + C_e (w_a - w_b).$$

Plugging this equality to Eq. 43, we get

$$\bar{h}_i - h_{b,i} = -C_e w_b,$$

which is the same result of the balanced case where the number of classes is $K$. As a result, Eq. 39 becomes equality if and only if

$$\bar{h}_i - h_{k,i} = -C_e w_k, \quad k \in \mathbb{K}^2, \; i \in [n]. \tag{46}$$

The remainder of the proof is identical to that in (Fisher et al., 2024). However, for the sake of clarity, we present it here in full rather than omitting it.

Applying Lemma 4 shown in the below, we have $(W, H)$, which satisfies Eq. 35.

Reversely, it is easy to verify that Eq. 42 reduces to equality when $(W, H)$ admits Eq. 35. So, $L_0$ is the global minimum of Eq. 9 and Eq. 35 is the unique form for the minimizers. We complete the proof of Theorem 3. □

**Lemma 4.** *Suppose $(W, H)$ satisfies*

$$\bar{h}_i - h_{k,i} = -\sqrt{\frac{E_H}{E_W}} w_k, \quad k \in [K], \; i \in [n], \tag{47}$$

*and*

$$\frac{1}{K} \sum_{k=1}^{K} \frac{1}{n} \sum_{i=1}^{n} \|h_{k,i}\|^2 = E_H, \quad \frac{1}{K} \sum_{k=1}^{K} \|w_k\|^2 = E_W, \quad \bar{h}_i = 0_p, \; i \in [n], \tag{48}$$

*where $\bar{h}_i := \frac{1}{K} \sum_{k=1}^{K} h_{k,i}$ with $i \in [n]$. Moreover, there exists a constant $C$ such that for all $\{(k, i, k') : k \in [K], k' \in [K], k' \neq k, i \in [n]\}$, we have*

$$h_{k,i} \cdot w_k = h_{k,i} \cdot w_{k'} + C. \tag{49}$$

*Then, $(W, H)$ satisfies Eq. 35.*

*Proof.* Combining Eq. 47 with the last equality in Eq. 48, we have

$$W = \sqrt{\frac{E_W}{E_H}} [h_1, ..., h_K]^{\mathsf{T}}, \quad h_{k,i} = h_k, \; k \in [K], \; i \in [n].$$

Thus, it remains to show

$$W = \sqrt{E_W} (M^*)^{\mathsf{T}}, \tag{50}$$

where $M^*$ is a $K$-simplex ETF.

Plugging $h_k = h_{k,i} = \sqrt{\frac{E_W}{E_H}} w_k$ into Eq. 49, we have, for all $(k, k') \in \{(k, k') : k \in [K], k' \in [K], k' \neq k\}$,

$$\sqrt{\frac{E_W}{E_H}} \|w_k\|^2 = h_{k,i} \cdot w_k = h_{k,i} \cdot w_{k'} + C = \sqrt{\frac{E_W}{E_H}} w_k w_{k'} + C,$$

and

$$\sqrt{\frac{E_W}{E_H}}\|\boldsymbol{w}_{k'}\|^2 = \boldsymbol{h}_{k',i} \cdot \boldsymbol{w}_{k'} = \boldsymbol{h}_{k',i} \cdot \boldsymbol{w}_k + C = \sqrt{\frac{E_W}{E_H}}\boldsymbol{w}_{k'}\boldsymbol{w}_k + C.$$

Therefore, from $\frac{1}{K}\sum_{k=1}^{K}\|\boldsymbol{w}_k\|^2 = E_W$, we have $\|\boldsymbol{w}_k\| = \sqrt{E_W}$ and $\boldsymbol{h}_k\boldsymbol{w}_{k'} = C' := \sqrt{E_H E_W} - C$.

Furthermore, recalling that $\bar{\boldsymbol{h}}_i = \boldsymbol{0}_p$ for $i \in [n]$, we have $\sum_{k=1}^{K}\boldsymbol{h}_k = \boldsymbol{0}_p$, which further yields $\sum_{k=1}^{K}\boldsymbol{h}_k \cdot \boldsymbol{w}_{k'} = 0$ for $k' \in [K]$. Then, it follows from $\boldsymbol{h}_k\boldsymbol{w}_{k'} = C'$ and $\boldsymbol{h}_k\boldsymbol{w}_k = \sqrt{E_H E_W}$ that $\boldsymbol{h}_k\boldsymbol{w}_{k'} = -\sqrt{E_H E_W}/(K-1)$. Thus, we obtain

$$\boldsymbol{W}\boldsymbol{W}^\intercal = \sqrt{\frac{E_W}{E_H}}\boldsymbol{W}[\boldsymbol{h}_1, ..., \boldsymbol{h}_K] = E_W\left[\frac{K}{K-1}\left(\boldsymbol{I}_K - \frac{1}{K}\boldsymbol{1}_K\boldsymbol{1}_K^\intercal\right)\right],$$

which implies Eq. 50. We complete the proof. $\square$

## C.5 PROOF OF PROPOSITION 2

To prove that only mixed labels $(a, b)$ for the case where $a < b$ ensures Theorem 1 and Proposition 1, we demonstrate that the following statement is true.

**Proposition 2.** *Let $\mathbb{K}^<$ be the mixed label set where $a < b$ for all $(a, b) \in \mathbb{K}^2$ and $\boldsymbol{W}_{\mathbb{K}^<}^\lambda$ be the partial matrix of $\boldsymbol{W}^\lambda$ which has class vectors for mixed labels $(a, b) \in \mathbb{K}^<$.*

*Then, $\boldsymbol{W}$ is a $K$-simplex ETF if $\boldsymbol{W}_{\mathbb{K}^<}^\lambda$ is a $|\mathbb{K}^<|$-simplex ETF.*

For the simplicity, we remove the subscript $\mathbb{K}^<$ of $\boldsymbol{W}_{\mathbb{K}^<}^\lambda$ and $\boldsymbol{w}_{\mathbb{K}^<}^\lambda$ in the following proof.

*Proof.* Let $f(x; \alpha, \beta)$ be the probability density function of Beta distribution $D_\lambda(\alpha, \beta)$. For the mixup ratio $\lambda$ sampled from $D_\lambda(\alpha, \alpha)$, we have

$$
\begin{aligned}
\boldsymbol{w}_{(a,b)}^\lambda &= \mathbb{E}_\lambda \left( \lambda \boldsymbol{w}_a + (1 - \lambda) \boldsymbol{w}_b \right) \\
&\overset{a}{=} \frac{1}{2} \mathbb{E}_\lambda \left( (\lambda \boldsymbol{w}_a + (1 - \lambda) \boldsymbol{w}_b) + ((1 - \lambda) \boldsymbol{w}_a + \lambda \boldsymbol{w}_b) \right) \\
&= \frac{1}{2} (\boldsymbol{w}_a + \boldsymbol{w}_b),
\end{aligned}
\tag{51}
$$

where in $\overset{a}{=}$, we use $f(\lambda; \alpha, \alpha) = f(1 - \lambda; \alpha, \alpha)$.

From the definition of a simplex ETF, we get

$$
\sum_{(a,b) \in \mathbb{K}^<} \boldsymbol{w}_{(a,b)}^\lambda = 0
\tag{52}
$$

Plugging the equality of Eq. 51 into Eq. 52, we have

$$
\sum_{(a,b) \in \mathbb{K}^<} \boldsymbol{w}_{(a,b)}^\lambda = \frac{K - 1}{2} \sum_{i=1}^K \boldsymbol{w}_i = 0
$$

$$
\therefore \boldsymbol{w}_i = -\sum_{j \neq i}^K \boldsymbol{w}_j, \quad \forall i \in [K]
\tag{53}
$$

From the definition of $\mathbb{K}^<$, we can get $< i, j, k >$ for all $i \in [K]$, satisfying

$$
\boldsymbol{w}_i = \boldsymbol{w}_{\{i,j\}}^\lambda - \boldsymbol{w}_{\{j,k\}}^\lambda + \boldsymbol{w}_{\{i,k\}}^\lambda,
\tag{54}
$$

where $\{a, b\} = (a, b)$ if $a < b$ otherwise $(b, a)$ and $i \neq j \neq k$.

Now, we show that $\boldsymbol{w}_i^\top \boldsymbol{w}_{i'} = -\frac{1}{K-1}$ is true for all $i \neq i'$

$$
\begin{aligned}
\boldsymbol{w}_i^\top \boldsymbol{w}_{i'} &\overset{Eq.\ 54}{=} \left( \boldsymbol{w}_{\{i,j\}}^\lambda - \boldsymbol{w}_{\{j,k\}}^\lambda + \boldsymbol{w}_{\{i,k\}}^\lambda \right)^\top \left( \boldsymbol{w}_{\{i',j'\}}^\lambda - \boldsymbol{w}_{\{j',k'\}}^\lambda + \boldsymbol{w}_{\{i',k'\}}^\lambda \right) \\
&\overset{a}{=} -\frac{1}{K-1}
\end{aligned}
\tag{55}
$$

where in $\overset{a}{=}$, we use the property of the simplex ETF, i.e., $\left( \boldsymbol{w}_{(a,b)}^\lambda \right)^\top \boldsymbol{w}_{(a',b')}^\lambda = -\frac{1}{K-1}$ for all $(a, b) \neq (a', b')$. We complete the proof. $\square$

## D EXPERIMENTAL SETUP

**Implementation Details.** Our experiments follow the setups of Zhong et al. (2021) and Zhou et al. (2020) for CIFAR10-LT, ImageNet-LT, Places-LT, and iNaturalist2018 and Yang et al. (2022) for CIFAR100-LT. We employ ResNet32 for CIFAR10-LT, doubling the feature dimensions for CIFAR100-LT. For ImageNet-LT and iNaturalist2018, we use ResNet50, and for Places-LT, use ResNet152, respectively. To reproduce baseline comparisons, we adopt the same hyperparameter settings as in Zhong et al. (2021) and Zhou et al. (2020).

**Datasets.** Following Zhong et al. (2021); Zhou et al. (2020), we use the long-tailed variants of CIFAR10, CIFAR100, ImageNet (Russakovsky et al., 2015), Places365 (Zhou et al., 2017), and iNaturalist2018 (Cui et al., 2018).

*CIFAR10-LT.* 10 imbalanced classes, subsampled at exponentially decreasing rates from CIFAR10 (Zhong et al., 2021).

*CIFAR100-LT.* 100 imbalanced classes, constructed analogously to CIFAR10-LT.

*ImageNet-LT.* Derived from ImageNet for large-scale object classification. Class frequencies follow a Pareto distribution ($\alpha = 5$) with cardinalities from 5 to 1,280, totaling 115.8K images across 1,000 classes.

*Places-LT.* An extended version of Places, with class sizes ranging from 5 to 4,980, yielding 184.5K images from 365 classes.

*iNaturalist2018.* A large-scale real-world species classification dataset with extreme label imbalance, comprising 437,513 images from 8,142 categories.

**Architectures.** For CIFAR10-LT, we use ResNet32 (Zhong et al., 2021) with three residual blocks, producing feature dimensions of 16, 32, and 64, respectively. CIFAR100-LT doubles these dimensions. Differing from the standard ResNet architecture used for ImageNet, the ResNet32's first convolutional layer has a kernel size, stride, and padding of 3, 1, and 1, respectively. ResNet50 and 152 follow He et al. (2015).

**Hyperparameters.** For CIFAR10/100-LT, models are trained with mini-batch size 128 using SGD with momentum 0.9 and weight decay 2e-4 for 200 epochs. The learning rate is linearly warmed up from 0.02 and decayed by 0.1 at epochs 160 and 180. For ImageNet-LT and Places-LT, models are trained with SGD (momentum 0.9, weight decay 5e-4) and a cosine annealing scheduler. Mixup alpha is set per dataset: $\alpha = 1.0$ for CIFAR10/100-LT, $\alpha = 0.2$ for others.

**ETF+DR** (Yang et al., 2022). In Yang et al. (2022), it was proven that by fixing the classifier as a $K$-simplex ETF, NC is satisfied regardless of class balance, and that using this fixed ETF classifier along with a specialized loss (Dot-Regression; DR) improves model performance in imbalanced learning environments. Leveraging the advantages of the fixed ETF classifier, we hypothesized that our method could produce synergies with this approach, and we conducted experiments applying our method to this framework. However, a scale factor is necessary for the fixed ETF classifier, due to class vectors should be normalized. For this reason, we make a modified version of the fixed ETF classifier to apply our methods, named as *fixed Mixed-Singleton Weighted ETF classifier (MS-WETF)*.

The scale of class vectors is important for softmax cross-entropy loss. Thus, we remove the scale factor and add learnable parameter $s \in \mathbb{R}^K$ to control the scale of each class vectors.

$$W_{\text{WETF}} = s \cdot W_{\text{ETF}}$$

Then, we make $W_{\text{WETF}}$ as Mixed-Singleton classifier

$$W_{\text{MS-WETF, (a,b)}}^{\lambda} = [\lambda w_{\text{WETF},a} + (1 - \lambda) w_{\text{WETF},b}]_{(a,b) \in \mathbb{K}^2}$$

**Remix** (Chou et al., 2020). In (Chou et al., 2020), they pointed out that using the same mixing factor $\lambda$ for mixed samples in both last-layer features and their respective labels does not make sense under the imbalanced learning environments. As a result, Remix has been proposed to disentangle $\lambda$ as

below:

$$\tilde{\boldsymbol{x}}^{\mathrm{RM}} = \lambda_x \boldsymbol{x}_i + (1 - \lambda_x)\boldsymbol{x}_{\pi(i)},$$
$$\tilde{\boldsymbol{y}}^{\mathrm{RM}} = \lambda_y \boldsymbol{y}_{c_i} + (1 - \lambda_y)\boldsymbol{y}_{c_{\pi(i)}}, \forall (i, \pi(i)) \in \mathcal{I}^\lambda,$$

where $\lambda_x$ is sampled from the beta distribution and $\lambda_y$ is defined as below:

$$\lambda_y = \begin{cases} 0 & n_i/n_{\pi(i)} \geq \kappa \text{ and } \lambda < \tau; \\ 1 & n_i/n_{\pi(i)} \leq 1/\kappa \text{ and } 1 - \lambda < \tau; \\ \lambda & \text{otherwise} \end{cases}$$

Here $n_i$ and $n_{\pi(i)}$ denote the number of samples in the corresponding class from $\boldsymbol{x}_i$ and $\boldsymbol{x}_{\pi(i)}$. $\kappa$ and $\tau$ are two hyperparameters in Remix, and we used the same values as those employed in the original Remix implementation: $\kappa = 3$ and $\tau = 0.5$.

Unlike Remix, which controls the mixing factor $\lambda$, our method controls only the sample and label pairs. Owing to this independence from Remix, integrating our method with Remix simply requires replacing the original index pair set $\mathcal{I}^\lambda$ with the balanced mixed-label pair set $\tilde{\mathcal{I}}^\lambda$ obtained through BMLS. Also, when initializing the MS classifier, we used $\lambda_y$ from the same way to Remix.

**DBN-mix** (Baik et al., 2024). DBN-mix is a method that expands bilateral mixup (which is from BBN-mix (Zhou et al., 2020)) to double branches while one input sample $\boldsymbol{x}_i$ comes from random sampler and the other $\boldsymbol{x}_j$ comes from class-balanced sampler. Therefore, there are two mixed samples generated in each mini-batch as below:

$$\tilde{\boldsymbol{x}}^{\mathrm{cb}} = \lambda \boldsymbol{x}_i + (1 - \lambda)\boldsymbol{x}_j,$$
$$\tilde{\boldsymbol{x}}^{\mathrm{rb}} = (1 - \lambda)\boldsymbol{x}_i + \lambda \boldsymbol{x}_j,$$
$$\tilde{\boldsymbol{y}}^{\mathrm{cb}} = \lambda \boldsymbol{y}_i + (1 - \lambda)\boldsymbol{y}_j,$$
$$\tilde{\boldsymbol{y}}^{\mathrm{rb}} = (1 - \lambda)\boldsymbol{y}_i + \lambda \boldsymbol{y}_j,$$

where the mixed samples $\tilde{\boldsymbol{x}}^{\mathrm{cb}}$ and $\tilde{\boldsymbol{x}}^{\mathrm{rb}}$ are trained by their respective different classifiers.

In this setting, the loss from each branch $\mathcal{L}$ is computed separately as shown below, and the final loss $\mathcal{L}_{\mathrm{total}}$ is obtained by taking their weighted sum via a hyperparameter $\gamma$.

$$\mathcal{L}_{\mathrm{total}} = \gamma \cdot \mathcal{L}\left(\tilde{\boldsymbol{p}}^{\mathrm{cb}}, \tilde{\boldsymbol{y}}^{\mathrm{cb}}\right) + (1 - \gamma) \cdot \mathcal{L}\left(\tilde{\boldsymbol{p}}^{\mathrm{rb}}, \tilde{\boldsymbol{y}}^{\mathrm{rb}}\right),$$

where $\tilde{\boldsymbol{p}}$ is the logit of the mixed sample $\tilde{\boldsymbol{x}}$ and $\mathcal{L}$ denotes the cross-entropy loss. This loss is then used to train the classifiers of each branch and the shared backbone in an end-to-end manner.

In DBN-mix, two different samples—each drawn from a different sampler—are mixed together, which causes the mixed-label balance to break in both branches even if the class-balanced sampler is replaced with BMLS. To address this, we employ two samplers in parallel and configure each branch as follows:

$$\tilde{\boldsymbol{x}}^{\mathrm{cb}} = \lambda \boldsymbol{x}_i + (1 - \lambda)\boldsymbol{x}_{\pi(i)},$$
$$\tilde{\boldsymbol{x}}^{\mathrm{rb}} = \lambda \boldsymbol{x}_j + (1 - \lambda)\boldsymbol{x}_{\pi(j)},$$
$$\tilde{\boldsymbol{y}}^{\mathrm{cb}} = \lambda \boldsymbol{y}_i + (1 - \lambda)\boldsymbol{y}_{\pi(i)},$$
$$\tilde{\boldsymbol{y}}^{\mathrm{rb}} = \lambda \boldsymbol{y}_j + (1 - \lambda)\boldsymbol{y}_{\pi(j)},$$

for all $(i, \pi(i)) \in \mathcal{I}^\lambda$ and $(j, \pi(j)) \in \tilde{\mathcal{I}}^\lambda$, which denote a random sampler and BMLS, respectively.

In our experiments, we empirically identified appropriate values for the hyperparameter $\gamma$. As a result, we set $\gamma = 0.9$ for CIFAR10-LT and $\gamma = 0.5$ for CIFAR100-LT.

# E ADDITIONAL EXPERIMENTAL RESULTS

According to Liu et al. (2019), we also calculate top-1 test accuracy of three disjoint set: many, medium, and few classes. The classes included in each set for the respective datasets are described in Table 5. In the tables of experimental results about many, medium, and few classes, we report the mean and std of top-1 test accuracies as $mean_{std}$.

Table 5: The classes in Many/Medium/Few class sets.

|  | CIFAR10-LT | CIFAR100-LT | Places-LT | ImageNet-LT | iNaturalist2018 |
|---|---|---|---|---|---|
| Many | [0,2] | [0,35] | [0, 130] | [0, 389] | [0, 841] |
| Medium | [3,6] | [36,70] | [131, 287] | [390, 835] | [842, 4542] |
| Few | [7,9] | [71,99] | [288, 364] | [835, 999] | [4543, 8141] |

Table 6: Extension to the fixed ETF classifier on CIFAR10/100-LT datasets with various imbalance factors. The results are the mean of five repeated experiments with random seeds. Best in bold (†: the reported values are taken from Yang et al. (2022))

| Sampler | Clf. | $\mathcal{L}$ | CIFAR10-LT | | | | CIFAR100-LT | | | |
|---|---|---|---|---|---|---|---|---|---|---|
|  |  |  | imbalance factor | | | | imbalance factor | | | |
|  |  |  | 200 | 100 | 50 | 10 | 200 | 100 | 50 | 10 |
| random | ETF | CE† | 60.06 | 67.00 | 77.20 | 87.00 | N/A | N/A | N/A | N/A |
| random | ETF | DR† | 71.90 | 76.50 | 81.00 | 87.70 | 40.90 | 45.30 | 50.40 | N/A |
| random | ETF | DR | 71.58 | 76.82 | 81.25 | 87.59 | 41.20 | 45.07 | 50.71 | 63.08 |
| CBS | ETF | DR | 69.35 | 75.46 | 81.15 | 88.38 | 38.78 | 42.96 | 48.84 | 62.01 |
| CAS | ETF | DR | 69.17 | 76.16 | 80.81 | **88.61** | 38.91 | 43.18 | 49.05 | 62.50 |
| BMLS | ETF | DR | **77.77** | **80.38** | **84.30** | 87.91 | 39.54 | 43.60 | 49.54 | 62.06 |
|  |  | diff. | +6.19 | +3.56 | +3.05 | +0.32 | -1.66 | -1.47 | -1.17 | -1.02 |
| BMLS | MS-WETF | CE | 77.73 | 80.31 | 84.22 | 88.26 | **42.73** | **47.10** | **52.44** | **64.10** |
|  |  | diff. | +6.15 | +3.49 | +2.97 | +0.67 | +1.53 | +2.03 | +1.73 | +1.02 |

Table 7: Comparison experiments of samplers on the CIFAR10/100-LT dataset with various imbalance factors. The results are the mean of five repeated experiments with random seeds. Best in bold (CBS: Class-Balanced Sampler, CAS: Class-Aware Sampler, BMLS: Balanced Mixed Label Sampler)

| Method | CIFAR10-LT | | | | CIFAR100-LT | | | |
|---|---|---|---|---|---|---|---|---|
| | imbalance factor | | | | imbalance factor | | | |
| | 200 | 100 | 50 | 10 | 200 | 100 | 50 | 10 |
| *mixup* | | | | | | | | |
| Mixup (Zhang et al., 2018) | 67.30 | 72.80 | 78.60 | 87.70 | 38.70 | 43.00 | 48.10 | 58.20 |
| Remix (Zhang et al., 2022) | N/A | 73.00 | N/A | 88.50 | N/A | 41.40 | N/A | 59.50 |
| Remix+RS (Chou et al., 2020) | N/A | 76.23 | N/A | 87.70 | N/A | 41.13 | N/A | 58.62 |
| CMO (Park et al., 2021) | N/A | N/A | N/A | N/A | N/A | 43.90 | 48.30 | 59.50 |
| SBN-mix (Baik et al., 2024) | 69.87 | 76.33 | 81.04 | 89.84 | 40.30 | 45.07 | 50.39 | 62.37 |
| OTMix (Gao et al., 2023) | N/A | 78.30 | 83.40 | 90.20 | N/A | 46.40 | 50.70 | 61.60 |
| ETF+CE (Yang et al., 2022) | 60.06 | 67.00 | 77.20 | 87.00 | N/A | N/A | N/A | N/A |
| ETF+DR (Yang et al., 2022) | 71.90 | 76.50 | 81.00 | 87.70 | 40.90 | 45.30 | 50.40 | N/A |
| *2-stage or extra network* | | | | | | | | |
| BBN-mix (Zhou et al., 2020) | N/A | 79.82 | 82.18 | 88.32 | N/A | 42.56 | 47.02 | 59.12 |
| DBN-mix (Baik et al., 2024) | 79.58 | 83.47 | 86.82 | 90.87 | **46.21** | **51.04** | 54.93 | 64.98 |
| UniMix (Xu et al., 2021) | 78.48 | 82.75 | 84.32 | 89.66 | 42.07 | 45.45 | 51.11 | 61.25 |
| MiSLAS (Zhong et al., 2021) | N/A | 82.10 | 85.70 | 90.00 | N/A | 47.00 | 52.30 | 63.20 |
| CP-Mix (Yoon et al., 2025) | 78.34 | 82.44 | 85.08 | 89.87 | 43.56 | 48.20 | 52.12 | 61.91 |
| *class-balance loss* | | | | | | | | |
| CB+RS (Cao et al., 2019) | N/A | 70.55 | N/A | 86.79 | N/A | 33.44 | N/A | 55.06 |
| CB+RW (Cui et al., 2019) | N/A | 72.37 | N/A | 86.54 | N/A | 33.99 | N/A | 57.12 |
| CB+Focal (Cui et al., 2019) | N/A | 74.57 | N/A | 87.10 | N/A | 36.02 | N/A | 57.99 |
| LDAM (Cao et al., 2019) | N/A | 73.35 | N/A | 86.96 | N/A | 39.60 | N/A | 56.91 |
| LDAM+DRW (Cao et al., 2019) | N/A | 77.03 | N/A | 88.16 | N/A | 42.04 | N/A | 58.71 |
| *class-balance sampling* | | | | | | | | |
| CAS (Shen & Lin, 2016) | N/A | 68.40 | N/A | 86.90 | N/A | 31.90 | N/A | 55.00 |
| LOM (Zhang et al., 2022) | N/A | 74.20 | N/A | 89.40 | N/A | 41.50 | N/A | 59.90 |
| CAS+DRW (Shen & Lin, 2016) | N/A | 73.50 | N/A | 87.70 | N/A | 41.50 | N/A | 57.60 |
| LOM+DRW (Zhang et al., 2022) | N/A | 78.70 | N/A | 89.60 | N/A | 46.20 | N/A | 61.10 |
| *reproduced results and our method* | | | | | | | | |
| Mixup | 66.77 | 72.94 | 78.64 | 88.05 | 39.06 | 42.88 | 48.31 | 63.03 |
| +LOM | 70.17 | 76.63 | 81.15 | 89.24 | 39.61 | 44.24 | 49.99 | 63.90 |
| +CAS | 69.90 | 76.43 | 81.42 | 89.24 | 40.28 | 44.65 | 50.07 | 63.57 |
| +BMLS$_{MS}$ | 74.70 | 79.67 | 83.46 | 88.51 | 41.71 | 47.62 | 52.74 | 64.47 |
| diff. | +7.93 | +6.73 | +4.82 | +0.46 | +2.65 | +4.74 | +4.43 | +1.44 |
| ETF+DR | 71.58 | 76.82 | 81.25 | 87.59 | 41.20 | 45.07 | 50.71 | 63.08 |
| BMLS+WETF$_{MS}$+CE | 77.73 | 80.31 | 84.22 | 88.26 | 42.73 | 47.10 | 52.44 | 64.10 |
| diff. | +6.15 | +3.49 | +2.97 | +0.67 | +1.53 | +2.03 | +1.73 | +1.02 |
| Remix | 69.58 | 75.15 | 80.41 | 88.61 | 41.03 | 44.95 | 50.19 | 63.45 |
| +BMLS | 73.95 | 80.10 | 83.92 | 88.62 | 39.95 | 46.34 | 51.53 | 64.42 |
| +BMLS$_{MS}$ | 73.18 | 78.00 | 83.70 | 88.20 | 40.25 | 46.82 | 49.78 | 63.54 |
| diff. | +3.60 | +2.85 | +3.29 | -0.41 | -0.78 | +1.87 | -0.41 | +0.09 |
| DBN-mix | 77.40 | 82.40 | 86.05 | **91.01** | 40.71 | 45.52 | 50.47 | 62.68 |
| +BMLS$_{MS}$ | **79.73** | **84.30** | **87.28** | 90.93 | 44.42 | 49.08 | **55.41** | **65.42** |
| diff. | +2.33 | +1.90 | +1.23 | -0.08 | +3.71 | +3.56 | +4.94 | +2.74 |

Table 8: Experimental results on Many/Medium/Few classes in the CIFAR10/100-LT datasets.

| | Method | Clf. | CIFAR10-LT | | | | CIFAR100-LT | | | |
|---|---|---|---|---|---|---|---|---|---|---|
| | | | many | med | few | all | many | med | few | all |
| imb 200 | random | FC | $91.17_{3.65}$ | $69.99_{2.25}$ | $38.09_{6.32}$ | $66.77_{0.76}$ | $71.16_{0.52}$ | $35.22_{0.20}$ | $3.85_{0.47}$ | $39.06_{0.23}$ |
| | CBS | FC | $82.63_{2.72}$ | $69.29_{3.79}$ | $58.89_{3.94}$ | $70.17_{0.51}$ | $65.92_{0.51}$ | $39.44_{0.96}$ | $7.15_{0.50}$ | $39.61_{0.50}$ |
| | CAS | FC | $85.65_{3.74}$ | $67.67_{3.86}$ | $57.14_{4.43}$ | $69.90_{0.77}$ | $66.32_{0.55}$ | $40.54_{0.59}$ | $7.62_{0.28}$ | $40.28_{0.29}$ |
| | BMLS | FC | $90.49_{0.26}$ | $74.12_{1.21}$ | $54.43_{2.25}$ | $73.13_{0.67}$ | $65.29_{0.45}$ | $41.33_{0.76}$ | $7.09_{0.37}$ | $40.03_{0.38}$ |
| | BMLS | MS | $88.94_{0.32}$ | $72.97_{0.83}$ | $62.77_{1.30}$ | $74.70_{0.45}$ | $63.24_{0.57}$ | $44.86_{0.42}$ | $11.19_{0.76}$ | $41.71_{0.36}$ |
| imb 100 | random | FC | $93.39_{2.42}$ | $74.05_{2.03}$ | $50.99_{5.40}$ | $72.94_{0.68}$ | $72.09_{0.21}$ | $41.10_{0.40}$ | $8.77_{0.42}$ | $42.88_{0.15}$ |
| | CBS | FC | $90.89_{2.45}$ | $74.31_{2.99}$ | $65.46_{5.76}$ | $76.63_{0.41}$ | $67.07_{0.74}$ | $46.29_{0.72}$ | $13.43_{0.37}$ | $44.24_{0.14}$ |
| | CAS | FC | $90.54_{2.86}$ | $75.54_{1.95}$ | $63.51_{6.26}$ | $76.43_{0.60}$ | $68.28_{0.35}$ | $46.47_{0.34}$ | $13.12_{0.15}$ | $44.65_{0.26}$ |
| | BMLS | FC | $88.53_{1.01}$ | $77.84_{0.25}$ | $70.53_{1.27}$ | $78.85_{0.34}$ | $68.38_{0.27}$ | $46.89_{0.33}$ | $14.37_{0.88}$ | $45.20_{0.33}$ |
| | BMLS | MS | $89.14_{0.63}$ | $76.34_{0.62}$ | $74.63_{0.69}$ | $79.67_{0.21}$ | $66.31_{0.26}$ | $49.80_{0.57}$ | $21.80_{0.40}$ | $47.62_{0.25}$ |
| imb 50 | random | FC | $95.25_{0.23}$ | $78.52_{0.54}$ | $62.19_{1.04}$ | $78.64_{0.57}$ | $73.72_{0.18}$ | $48.62_{0.23}$ | $16.40_{0.93}$ | $48.31_{0.28}$ |
| | CBS | FC | $91.57_{3.17}$ | $79.61_{1.58}$ | $72.78_{4.05}$ | $81.15_{0.48}$ | $68.80_{0.46}$ | $52.97_{0.28}$ | $23.06_{0.56}$ | $49.99_{0.13}$ |
| | CAS | FC | $92.78_{0.43}$ | $79.28_{0.46}$ | $72.89_{0.96}$ | $81.42_{0.27}$ | $69.16_{0.61}$ | $52.71_{0.34}$ | $23.18_{0.41}$ | $50.07_{0.27}$ |
| | BMLS | FC | $91.86_{0.40}$ | $81.32_{0.36}$ | $76.63_{1.05}$ | $83.07_{0.43}$ | $69.77_{0.55}$ | $54.55_{0.28}$ | $26.83_{0.67}$ | $51.99_{0.26}$ |
| | BMLS | MS | $89.45_{0.15}$ | $79.29_{0.54}$ | $83.03_{0.50}$ | $83.46_{0.36}$ | $67.06_{0.51}$ | $55.30_{0.84}$ | $31.88_{1.39}$ | $52.74_{0.55}$ |
| imb 10 | random | FC | $94.79_{0.55}$ | $85.38_{0.27}$ | $84.86_{1.23}$ | $88.05_{0.27}$ | $76.06_{0.32}$ | $64.10_{0.63}$ | $45.56_{0.57}$ | $63.03_{0.17}$ |
| | CBS | FC | $93.95_{0.78}$ | $86.04_{0.57}$ | $88.81_{0.28}$ | $89.24_{0.37}$ | $72.42_{0.64}$ | $65.76_{0.45}$ | $51.08_{0.69}$ | $63.90_{0.37}$ |
| | CAS | FC | $94.14_{0.23}$ | $86.34_{0.24}$ | $88.21_{0.43}$ | $89.24_{0.18}$ | $72.59_{0.49}$ | $65.20_{0.47}$ | $50.40_{0.66}$ | $63.57_{0.26}$ |
| | BMLS | FC | $91.17_{0.40}$ | $87.04_{0.26}$ | $90.98_{0.59}$ | $89.46_{0.19}$ | $71.05_{0.70}$ | $68.93_{0.66}$ | $55.24_{0.47}$ | $65.72_{0.29}$ |
| | BMLS | MS | $91.63_{0.60}$ | $84.92_{0.63}$ | $90.18_{0.56}$ | $88.51_{0.19}$ | $71.61_{0.21}$ | $65.67_{1.20}$ | $54.17_{1.49}$ | $64.47_{0.24}$ |

Table 9: Experimental results on Many/Medium/Few classes in the Places-LT datasets.

| Method | Clf. | Places-LT | | | | Places-LT (FT) | | | |
|---|---|---|---|---|---|---|---|---|---|
| | | many | med | few | all | many | med | few | all |
| random | FC | $42.02_{0.76}$ | $15.79_{0.54}$ | $0.86_{0.12}$ | $22.06_{0.50}$ | $43.79_{0.29}$ | $20.45_{0.27}$ | $6.59_{0.26}$ | $25.90_{0.06}$ |
| CBS | FC | $38.65_{1.97}$ | $22.60_{1.20}$ | $5.69_{0.52}$ | $24.79_{0.13}$ | $41.31_{0.09}$ | $39.98_{0.17}$ | $25.11_{0.11}$ | $37.32_{0.07}$ |
| CAS | FC | $40.68_{0.33}$ | $20.08_{0.53}$ | $4.86_{0.50}$ | $24.26_{0.22}$ | $41.35_{0.08}$ | $40.06_{0.06}$ | $25.46_{0.17}$ | $37.44_{0.04}$ |
| BMLS | FC | $38.43_{0.21}$ | $27.80_{0.12}$ | $7.47_{0.26}$ | $27.33_{0.17}$ | $34.65_{0.04}$ | $43.79_{0.05}$ | $29.00_{0.08}$ | $37.39_{0.01}$ |
| BMLS | MS | $39.39_{0.32}$ | $27.01_{0.40}$ | $10.39_{0.12}$ | $27.95_{0.26}$ | $41.33_{0.09}$ | $40.14_{0.00}$ | $27.05_{0.15}$ | $37.81_{0.01}$ |

Table 10: Experimental results on Many/Medium/Few classes in the ImageNet-LT and iNaturalist2018 datasets.

| Method | Clf. | ImageNet-LT | | | | iNaturalist2018 | | | |
|---|---|---|---|---|---|---|---|---|---|
| | | many | med | few | all | many | med | few | all |
| random | FC | $67.76_{0.43}$ | $38.72_{0.50}$ | $9.33_{0.28}$ | $45.19_{0.43}$ | $77.55_{0.39}$ | $66.66_{0.38}$ | $59.49_{0.38}$ | $64.62_{0.31}$ |
| CBS | FC | $62.46_{0.91}$ | $44.55_{1.10}$ | $20.00_{0.92}$ | $47.49_{0.99}$ | $63.25_{0.22}$ | $68.36_{0.15}$ | $66.63_{0.18}$ | $67.06_{0.04}$ |
| CAS | FC | $63.04_{0.31}$ | $43.83_{0.34}$ | $19.53_{0.40}$ | $47.31_{0.33}$ | $63.99_{0.63}$ | $68.80_{0.02}$ | $67.10_{0.08}$ | $67.55_{0.09}$ |
| BMLS | FC | $62.35_{0.69}$ | $46.53_{0.43}$ | $23.08_{0.54}$ | $48.83_{0.55}$ | $64.44_{2.52}$ | $68.33_{0.37}$ | $66.19_{0.87}$ | $66.98_{0.19}$ |
| BMLS | MS | $59.03_{0.89}$ | $45.87_{1.01}$ | $24.86_{0.92}$ | $47.54_{0.94}$ | $51.73_{0.83}$ | $57.15_{0.14}$ | $57.18_{0.28}$ | $56.60_{0.18}$ |

Table 11: Experimental results of the ablation study on Many/Medium/Few classes in the CIFAR10/100-LT datasets. The results are the mean of five repeated experiments with random seeds.

| | Method | Clf. | CIFAR10-LT | | | | CIFAR100-LT | | | |
|---|---|---|---|---|---|---|---|---|---|---|
| | | | many | med | few | all | many | med | few | all |
| imb 200 | random | FC | $91.17_{3.65}$ | $69.99_{2.25}$ | $38.09_{6.32}$ | $66.77_{0.76}$ | $71.16_{0.52}$ | $35.22_{0.20}$ | $3.85_{0.47}$ | $39.06_{0.23}$ |
| | random | MS | $88.59_{0.19}$ | $53.77_{1.07}$ | $16.74_{1.04}$ | $53.11_{0.58}$ | $64.59_{0.75}$ | $28.43_{0.49}$ | $0.75_{0.15}$ | $33.42_{0.37}$ |
| | BMLS | FC | $90.49_{0.26}$ | $74.12_{1.21}$ | $54.43_{2.25}$ | $73.13_{0.67}$ | $65.29_{0.45}$ | $41.33_{0.76}$ | $7.09_{0.37}$ | $40.03_{0.38}$ |
| | BMLS | MS | $88.94_{0.32}$ | $72.97_{0.83}$ | $62.77_{1.30}$ | $74.70_{0.45}$ | $63.24_{0.57}$ | $44.86_{0.42}$ | $11.19_{0.76}$ | $41.71_{0.36}$ |
| imb 100 | random | FC | $93.39_{2.42}$ | $74.05_{2.03}$ | $50.99_{5.40}$ | $72.94_{0.68}$ | $72.09_{0.21}$ | $41.10_{0.40}$ | $8.77_{0.42}$ | $42.88_{0.15}$ |
| | random | MS | $89.47_{0.46}$ | $62.24_{2.21}$ | $41.15_{3.22}$ | $64.08_{1.59}$ | $67.29_{0.31}$ | $33.84_{0.61}$ | $2.78_{0.32}$ | $36.87_{0.24}$ |
| | BMLS | FC | $88.53_{1.01}$ | $77.84_{0.25}$ | $70.53_{1.27}$ | $78.85_{0.34}$ | $68.38_{0.27}$ | $46.89_{0.33}$ | $14.37_{0.88}$ | $45.20_{0.33}$ |
| | BMLS | MS | $89.14_{0.63}$ | $76.34_{0.62}$ | $74.63_{0.69}$ | $79.67_{0.21}$ | $66.31_{0.26}$ | $49.80_{0.57}$ | $21.80_{0.40}$ | $47.62_{0.25}$ |
| imb 50 | random | FC | $95.25_{0.23}$ | $78.52_{0.54}$ | $62.19_{1.04}$ | $78.64_{0.57}$ | $73.72_{0.18}$ | $48.62_{0.23}$ | $16.40_{0.93}$ | $48.31_{0.28}$ |
| | random | MS | $90.04_{0.63}$ | $64.80_{1.76}$ | $52.11_{1.06}$ | $68.56_{0.50}$ | $68.28_{0.69}$ | $42.17_{0.89}$ | $8.00_{0.52}$ | $41.66_{0.42}$ |
| | BMLS | FC | $91.86_{0.40}$ | $81.32_{0.36}$ | $76.63_{1.05}$ | $83.07_{0.43}$ | $69.77_{0.55}$ | $54.55_{0.28}$ | $26.83_{0.67}$ | $51.99_{0.26}$ |
| | BMLS | MS | $89.45_{0.15}$ | $79.29_{0.54}$ | $83.03_{0.50}$ | $83.46_{0.36}$ | $67.06_{0.51}$ | $55.30_{0.84}$ | $31.88_{1.39}$ | $52.74_{0.55}$ |
| imb 10 | random | FC | $94.79_{0.55}$ | $85.38_{0.27}$ | $84.86_{1.23}$ | $88.05_{0.27}$ | $76.06_{0.32}$ | $64.10_{0.63}$ | $45.56_{0.57}$ | $63.03_{0.17}$ |
| | random | MS | $91.54_{0.43}$ | $76.31_{1.02}$ | $75.25_{1.44}$ | $80.56_{0.76}$ | $71.91_{0.35}$ | $57.38_{0.90}$ | $37.03_{0.72}$ | $56.71_{0.48}$ |
| | BMLS | FC | $91.17_{0.40}$ | $87.04_{0.26}$ | $90.98_{0.59}$ | $89.46_{0.19}$ | $71.05_{0.70}$ | $68.93_{0.66}$ | $55.24_{0.47}$ | $65.72_{0.29}$ |
| | BMLS | MS | $91.63_{0.60}$ | $84.92_{0.63}$ | $90.18_{0.56}$ | $88.51_{0.19}$ | $71.61_{0.21}$ | $65.67_{1.20}$ | $54.17_{1.49}$ | $64.47_{0.24}$ |

Table 12: Experimental results of extension to the fixed ETF Classifier on Many/Medium/Few classes in the CIFAR10-LT dataset. The results are the mean of five repeated experiments with random seeds.

| | Method | Clf. | $\mathcal{L}$ | CIFAR10-LT | | | |
| --- | --- | --- | --- | --- | --- | --- | --- |
| | | | | many | med | few | all |
| imb 200 | random | ETF | DR | $84.13_{0.64}$ | $73.89_{0.92}$ | $55.94_{1.24}$ | $71.58_{0.39}$ |
| | CBS | ETF | DR | $81.05_{3.12}$ | $69.26_{2.29}$ | $57.77_{4.75}$ | $69.35_{0.38}$ |
| | CAS | ETF | DR | $87.67_{6.09}$ | $72.17_{0.94}$ | $46.67_{6.61}$ | $69.17_{0.67}$ |
| | BMLS | ETF | DR | $84.52_{0.47}$ | $74.15_{0.36}$ | $75.85_{0.66}$ | $77.77_{0.13}$ |
| | BMLS | MS-WETF | CE | $85.41_{0.71}$ | $74.96_{0.45}$ | $73.74_{0.72}$ | $77.73_{0.32}$ |
| imb 100 | random | ETF | DR | $83.75_{0.92}$ | $75.42_{0.30}$ | $71.75_{0.95}$ | $76.82_{0.20}$ |
| | CBS | ETF | DR | $88.89_{3.19}$ | $74.46_{2.41}$ | $63.37_{6.15}$ | $75.46_{0.37}$ |
| | CAS | ETF | DR | $91.03_{0.54}$ | $75.97_{0.44}$ | $61.55_{2.15}$ | $76.16_{0.56}$ |
| | BMLS | ETF | DR | $88.85_{0.16}$ | $77.51_{0.39}$ | $75.74_{0.42}$ | $80.38_{0.23}$ |
| | BMLS | MS-WETF | CE | $86.71_{0.88}$ | $76.28_{0.69}$ | $79.27_{1.39}$ | $80.31_{0.43}$ |
| imb 50 | random | ETF | DR | $85.45_{0.50}$ | $78.60_{0.28}$ | $80.59_{0.42}$ | $81.25_{0.18}$ |
| | CBS | ETF | DR | $91.41_{1.07}$ | $79.15_{1.05}$ | $73.57_{1.93}$ | $81.15_{0.37}$ |
| | CAS | ETF | DR | $91.02_{1.68}$ | $79.26_{1.09}$ | $72.68_{2.07}$ | $80.81_{0.22}$ |
| | BMLS | ETF | DR | $88.17_{0.24}$ | $80.21_{0.19}$ | $85.87_{0.20}$ | $84.30_{0.07}$ |
| | BMLS | MS-WETF | CE | $87.01_{0.89}$ | $80.36_{0.67}$ | $86.59_{0.29}$ | $84.22_{0.43}$ |
| imb 10 | random | ETF | DR | $89.67_{0.52}$ | $83.81_{0.28}$ | $90.54_{0.39}$ | $87.59_{0.18}$ |
| | CBS | ETF | DR | $92.79_{0.23}$ | $85.14_{0.38}$ | $88.28_{0.41}$ | $88.38_{0.25}$ |
| | CAS | ETF | DR | $92.87_{0.28}$ | $85.33_{0.60}$ | $88.72_{0.22}$ | $88.61_{0.21}$ |
| | BMLS | ETF | DR | $88.76_{0.93}$ | $85.08_{1.00}$ | $90.83_{0.79}$ | $87.91_{0.24}$ |
| | BMLS | MS-WETF | CE | $91.27_{0.32}$ | $85.89_{0.20}$ | $88.40_{0.42}$ | $88.26_{0.04}$ |

Table 13: Experimental results of extension to the fixed ETF Classifier on Many/Medium/Few classes in the CIFAR100-LT dataset. The results are the mean of five repeated experiments with random seeds.

| | Method | Clf. | $\mathcal{L}$ | CIFAR100-LT | | | |
| --- | --- | --- | --- | --- | --- | --- | --- |
| | | | | many | med | few | all |
| imb 200 | random | ETF | DR | $68.23_{0.59}$ | $42.05_{0.52}$ | $6.63_{0.29}$ | $41.20_{0.18}$ |
| | CBS | ETF | DR | $63.90_{1.17}$ | $38.98_{0.81}$ | $7.36_{0.77}$ | $38.78_{0.25}$ |
| | CAS | ETF | DR | $64.10_{0.66}$ | $38.86_{0.68}$ | $7.68_{0.31}$ | $38.91_{0.43}$ |
| | BMLS | ETF | DR | $63.81_{0.48}$ | $39.09_{0.69}$ | $9.94_{0.54}$ | $39.54_{0.45}$ |
| | BMLS | MS-WETF | CE | $65.58_{0.70}$ | $45.26_{0.54}$ | $11.32_{0.52}$ | $42.73_{0.41}$ |
| imb 100 | random | ETF | DR | $69.85_{0.40}$ | $47.22_{0.35}$ | $11.72_{0.81}$ | $45.07_{0.25}$ |
| | CBS | ETF | DR | $65.43_{0.88}$ | $44.78_{0.94}$ | $12.88_{0.91}$ | $42.96_{0.25}$ |
| | CAS | ETF | DR | $66.04_{0.40}$ | $44.73_{0.34}$ | $12.93_{0.35}$ | $43.18_{0.18}$ |
| | BMLS | ETF | DR | $65.59_{0.18}$ | $44.49_{0.45}$ | $15.21_{0.49}$ | $43.60_{0.22}$ |
| | BMLS | MS-WETF | CE | $63.44_{0.32}$ | $51.15_{0.87}$ | $21.92_{0.72}$ | $47.10_{0.47}$ |
| imb 50 | random | ETF | DR | $70.56_{0.39}$ | $53.52_{0.65}$ | $22.69_{0.70}$ | $50.71_{0.24}$ |
| | CBS | ETF | DR | $67.73_{0.54}$ | $51.15_{0.13}$ | $22.59_{0.50}$ | $48.84_{0.16}$ |
| | CAS | ETF | DR | $67.87_{0.55}$ | $51.58_{0.62}$ | $22.63_{0.78}$ | $49.05_{0.36}$ |
| | BMLS | ETF | DR | $66.21_{0.58}$ | $51.02_{0.49}$ | $27.06_{0.59}$ | $49.54_{0.39}$ |
| | BMLS | MS-WETF | CE | $67.02_{0.90}$ | $54.66_{0.62}$ | $31.66_{0.41}$ | $52.44_{0.40}$ |
| imb 10 | random | ETF | DR | $72.76_{0.29}$ | $64.48_{0.50}$ | $49.39_{0.36}$ | $63.08_{0.21}$ |
| | CBS | ETF | DR | $70.89_{0.43}$ | $63.73_{0.42}$ | $48.90_{0.49}$ | $62.01_{0.19}$ |
| | CAS | ETF | DR | $71.13_{0.45}$ | $63.89_{0.34}$ | $50.12_{0.45}$ | $62.50_{0.27}$ |
| | BMLS | ETF | DR | $68.95_{1.20}$ | $64.83_{0.71}$ | $50.18_{1.26}$ | $62.06_{0.22}$ |
| | BMLS | MS-WETF | CE | $68.81_{0.40}$ | $64.95_{0.46}$ | $57.24_{0.28}$ | $64.10_{0.25}$ |

## F ADDITIONAL EXPERIMENTAL RESULTS FOR REBUTTAL

This page provided additional experimental results for rebuttal. These contents will be included in the main paper or appendix depending on the review.

### F.1 COMPARISON EXPERIMENTS FOR REMIX

Table 14: Comparison experiments of Remix on CIFAR10/100-LT datasets with various imbalance factors. The results are the mean of five repeated experiments with random seeds. Best in bold (CBS: Class-Balanced Sampler, CAS: Class-Aware Sampler, BMLS: Balanced Mixed Label Sampler, †: the reported values are taken from Chou et al. (2020), which used different experimental settings. ∗: the reproduced result of Remix on our experimental settings.)

| Method | CIFAR10-LT | | | | CIFAR100-LT | | | |
|---|---|---|---|---|---|---|---|---|
| | imbalance factor | | | | imbalance factor | | | |
| | 200 | 100 | 50 | 10 | 200 | 100 | 50 | 10 |
| Remix† | N/A | 75.36 | N/A | 88.15 | N/A | 41.94 | N/A | 59.36 |
| Remix$_{RS}$† | N/A | 76.23 | N/A | 87.70 | N/A | 41.13 | N/A | 58.62 |
| Remix* | 69.58 | 75.15 | 80.41 | 88.61 | **41.03** | 44.95 | 50.19 | 63.45 |
| +CBS | 71.39 | 76.72 | 82.03 | **89.39** | 39.95 | 43.72 | 49.46 | 63.49 |
| +CAS | 71.36 | 77.28 | 82.00 | 89.37 | 40.21 | 44.91 | 49.83 | 63.26 |
| +BMLS | **73.95** | **80.10** | **83.92** | 88.62 | 39.95 | 46.34 | **51.53** | **64.42** |
| +BMLS$_{MS}$ | 73.18 | 78.00 | 83.70 | 88.20 | 40.25 | **46.82** | 49.78 | 63.54 |

Table 15: Experimental results of Remix on Many/Medium/Few classes in the CIFAR10/100-LT datasets. The results are the mean of five repeated experiments with random seeds. (†: the reported values are taken from Chou et al. (2020), which used different experimental settings. ∗: the reproduced result of Remix on our experimental settings.)

| | Method | CIFAR10-LT | | | | CIFAR100-LT | | | |
|---|---|---|---|---|---|---|---|---|---|
| | | many | med | few | all | many | med | few | all |
| imb 200 | Remix† | N/A | N/A | N/A | N/A | N/A | N/A | N/A | N/A |
| | Remix$_{RS}$† | N/A | N/A | N/A | N/A | N/A | N/A | N/A | N/A |
| | Remix* | **92.31**$_{3.87}$ | 71.40$_{1.23}$ | 44.43$_{7.91}$ | 69.58$_{0.99}$ | **70.43**$_{0.23}$ | 39.83$_{1.02}$ | 5.99$_{0.42}$ | **41.03**$_{0.31}$ |
| | +CBS | 90.15$_{1.32}$ | 72.19$_{1.88}$ | 51.56$_{5.63}$ | 71.39$_{0.87}$ | 63.48$_{1.25}$ | 41.62$_{0.83}$ | 8.70$_{1.00}$ | 39.95$_{0.17}$ |
| | +CAS | 88.35$_{4.44}$ | 71.63$_{1.52}$ | 54.02$_{8.11}$ | 71.36$_{0.91}$ | 64.44$_{0.36}$ | 41.06$_{0.71}$ | 9.11$_{0.24}$ | 40.21$_{0.35}$ |
| | +BMLS | 80.43$_{7.52}$ | **73.66**$_{1.38}$ | **67.86**$_{7.14}$ | **73.95**$_{0.48}$ | 61.38$_{3.17}$ | **42.97**$_{0.88}$ | **9.70**$_{4.35}$ | 39.95$_{0.55}$ |
| | +BMLS$_{MS}$ | 89.47$_{0.44}$ | 72.17$_{0.54}$ | 58.23$_{1.19}$ | 73.18$_{0.22}$ | 64.88$_{0.30}$ | 40.37$_{0.77}$ | 9.55$_{0.46}$ | 40.25$_{0.32}$ |
| imb 100 | Remix† | N/A | N/A | N/A | 75.36 | N/A | N/A | N/A | 41.94 |
| | Remix$_{RS}$† | N/A | N/A | N/A | 76.23 | N/A | N/A | N/A | 41.13 |
| | Remix* | **93.70**$_{0.60}$ | 76.23$_{0.67}$ | 55.17$_{1.41}$ | 75.15$_{0.23}$ | **71.16**$_{0.41}$ | 45.58$_{0.85}$ | 11.67$_{0.80}$ | 44.95$_{0.37}$ |
| | +CBS | 91.31$_{0.63}$ | 76.54$_{1.27}$ | 62.37$_{1.84}$ | 76.72$_{0.62}$ | 64.42$_{0.30}$ | 46.74$_{0.62}$ | 14.38$_{0.32}$ | 43.72$_{0.29}$ |
| | +CAS | 90.76$_{0.81}$ | 76.72$_{0.85}$ | 64.55$_{1.22}$ | 77.28$_{0.43}$ | 66.21$_{0.43}$ | 47.48$_{0.34}$ | 15.36$_{0.80}$ | 44.91$_{0.20}$ |
| | +BMLS | 89.23$_{2.64}$ | **77.78**$_{1.46}$ | **74.05**$_{1.18}$ | **80.10**$_{0.36}$ | 64.88$_{0.47}$ | 48.86$_{0.46}$ | **20.28**$_{0.46}$ | 46.34$_{0.29}$ |
| | +BMLS$_{MS}$ | 91.25$_{0.64}$ | 74.78$_{0.75}$ | 69.05$_{1.84}$ | 78.00$_{0.36}$ | 67.21$_{0.47}$ | **48.92**$_{0.20}$ | 18.97$_{0.77}$ | **46.82**$_{0.30}$ |
| imb 50 | Remix† | N/A | N/A | N/A | N/A | N/A | N/A | N/A | N/A |
| | Remix$_{RS}$† | N/A | N/A | N/A | N/A | N/A | N/A | N/A | N/A |
| | Remix* | **94.25**$_{0.49}$ | 79.20$_{0.31}$ | 68.16$_{1.33}$ | 80.41$_{0.25}$ | **72.03**$_{0.51}$ | 51.73$_{0.15}$ | 21.22$_{0.84}$ | 50.19$_{0.24}$ |
| | +CBS | 91.47$_{0.71}$ | 79.68$_{0.52}$ | 75.73$_{1.57}$ | 82.03$_{0.34}$ | 67.05$_{0.30}$ | 52.47$_{0.59}$ | 23.98$_{0.31}$ | 49.46$_{0.26}$ |
| | +CAS | 92.08$_{0.32}$ | 79.89$_{0.64}$ | 74.72$_{1.14}$ | 82.00$_{0.48}$ | 67.88$_{0.63}$ | 52.42$_{0.24}$ | 24.28$_{0.33}$ | 49.83$_{0.20}$ |
| | +BMLS | 90.63$_{0.44}$ | **81.30**$_{0.33}$ | 80.70$_{1.15}$ | **83.92**$_{0.38}$ | 64.89$_{0.40}$ | **53.63**$_{0.57}$ | **32.41**$_{0.53}$ | **51.53**$_{0.40}$ |
| | +BMLS$_{MS}$ | 89.71$_{0.49}$ | 80.09$_{0.57}$ | **82.49**$_{0.87}$ | 83.70$_{0.16}$ | 67.16$_{1.86}$ | 51.46$_{0.82}$ | 26.17$_{1.78}$ | 49.78$_{0.43}$ |
| imb 10 | Remix† | N/A | N/A | N/A | 88.15 | N/A | N/A | N/A | 59.36 |
| | Remix$_{RS}$† | N/A | N/A | N/A | 87.70 | N/A | N/A | N/A | 58.62 |
| | Remix* | **94.85**$_{0.65}$ | 85.44$_{0.43}$ | 86.59$_{0.42}$ | 88.61$_{0.18}$ | **75.03**$_{0.58}$ | 63.77$_{0.40}$ | 48.70$_{0.85}$ | 63.45$_{0.40}$ |
| | +CBS | 93.75$_{0.19}$ | 85.79$_{0.55}$ | 89.83$_{0.43}$ | **89.39**$_{0.17}$ | 70.47$_{0.50}$ | 65.90$_{0.38}$ | 51.91$_{0.57}$ | 63.49$_{0.16}$ |
| | +CAS | 93.64$_{0.72}$ | **86.18**$_{0.61}$ | 89.36$_{0.84}$ | 89.37$_{0.24}$ | 70.72$_{0.43}$ | 65.45$_{0.41}$ | 51.37$_{0.63}$ | 63.26$_{0.10}$ |
| | +BMLS | 89.75$_{0.43}$ | 85.66$_{0.17}$ | **91.45**$_{0.22}$ | 88.62$_{0.11}$ | 69.28$_{1.67}$ | **68.32**$_{1.15}$ | **53.68**$_{1.36}$ | **64.42**$_{0.30}$ |
| | +BMLS$_{MS}$ | 92.26$_{0.17}$ | 84.59$_{0.34}$ | 88.95$_{0.21}$ | 88.20$_{0.15}$ | 70.59$_{0.42}$ | 65.11$_{0.32}$ | 52.88$_{0.61}$ | 63.54$_{0.26}$ |

### F.2 ABLATION STUDY INCLUDING K2 CLASSIFIER

Table 16: Ablation study on CIFAR10/100-LT datasets with various imbalance factors including $K^2$ classifier (notated as $K^2$ on the table). The results are the mean of five repeated experiments with random seeds. Best in bold (CBS: Class-Balanced Sampler, CAS: Class-Aware Sampler, BMLS: Balanced Mixed Label Sampler)

| Sampler | Clf. | CIFAR10-LT | | | | CIFAR100-LT | | | |
|---|---|---|---|---|---|---|---|---|---|
| | | imbalance factor | | | | imbalance factor | | | |
| | | 200 | 100 | 50 | 10 | 200 | 100 | 50 | 10 |
| *Sampler* | | | | | | | | | |
| random | FC | 66.77 | 72.94 | 78.64 | 88.05 | 39.06 | 42.88 | 48.31 | 63.03 |
| BMLS | FC | **73.13** | **78.85** | **83.07** | **89.46** | **40.03** | **45.20** | **51.99** | **65.72** |
| *Classifier* | | | | | | | | | |
| random | MS | 53.11 | 64.08 | 68.56 | 80.56 | 33.42 | 36.87 | 41.66 | 56.71 |
| BMLS | $K^2$ | 34.86 | 39.01 | 42.20 | 51.60 | 7.90 | 8.72 | 9.22 | 16.41 |
| BMLS | MS | **74.70** | **79.67** | **83.46** | **88.51** | **41.71** | **47.62** | **52.74** | **64.47** |

Table 17: Experimental results of the ablation study including $K^2$ classifier (notated as $K^2$ on the table) on Many/Medium/Few classes in the CIFAR10/100-LT datasets. The results are the mean of five repeated experiments with random seeds.

| | Method | Clf. | CIFAR10-LT | | | | CIFAR100-LT | | | |
|---|---|---|---|---|---|---|---|---|---|---|
| | | | many | med | few | all | many | med | few | all |
| imb 200 | *Sampler* | | | | | | | | | |
| | random | FC | $\textbf{91.17}_{3.65}$ | $69.99_{2.25}$ | $38.09_{6.32}$ | $66.77_{0.76}$ | $\textbf{71.16}_{0.52}$ | $35.22_{0.20}$ | $3.85_{0.47}$ | $39.06_{0.23}$ |
| | BMLS | FC | $90.49_{0.26}$ | $\textbf{74.12}_{1.21}$ | $\textbf{54.43}_{2.25}$ | $\textbf{73.13}_{0.67}$ | $65.29_{0.45}$ | $\textbf{41.33}_{0.76}$ | $\textbf{7.09}_{0.37}$ | $\textbf{40.03}_{0.38}$ |
| | *Classifier* | | | | | | | | | |
| | random | MS | $88.59_{0.19}$ | $53.77_{1.07}$ | $16.74_{1.04}$ | $53.11_{0.58}$ | $\textbf{64.59}_{0.75}$ | $28.43_{0.49}$ | $0.75_{0.15}$ | $33.42_{0.37}$ |
| | BMLS | $K^2$ | $67.94_{11.93}$ | $29.79_{6.92}$ | $8.55_{5.98}$ | $34.86_{0.92}$ | $14.69_{0.75}$ | $6.91_{0.63}$ | $0.67_{0.17}$ | $7.90_{0.13}$ |
| | BMLS | MS | $\textbf{88.94}_{0.32}$ | $\textbf{72.97}_{0.83}$ | $\textbf{62.77}_{1.30}$ | $\textbf{74.70}_{0.45}$ | $63.24_{0.57}$ | $\textbf{44.86}_{0.42}$ | $\textbf{11.19}_{0.76}$ | $\textbf{41.71}_{0.36}$ |
| imb 100 | *Sampler* | | | | | | | | | |
| | random | FC | $\textbf{93.39}_{2.42}$ | $74.05_{2.03}$ | $50.99_{5.40}$ | $72.94_{0.68}$ | $\textbf{72.09}_{0.21}$ | $41.10_{0.40}$ | $8.77_{0.42}$ | $42.88_{0.15}$ |
| | BMLS | FC | $88.53_{1.01}$ | $\textbf{77.84}_{0.25}$ | $\textbf{70.53}_{1.27}$ | $\textbf{78.85}_{0.34}$ | $68.38_{0.27}$ | $\textbf{46.89}_{0.33}$ | $\textbf{14.37}_{0.88}$ | $\textbf{45.20}_{0.33}$ |
| | *Classifier* | | | | | | | | | |
| | random | MS | $\textbf{89.47}_{0.46}$ | $62.24_{2.21}$ | $41.15_{3.22}$ | $64.08_{1.59}$ | $\textbf{67.29}_{0.31}$ | $33.84_{0.61}$ | $2.78_{0.32}$ | $36.87_{0.24}$ |
| | BMLS | $K^2$ | $58.35_{3.36}$ | $34.31_{5.73}$ | $25.93_{5.20}$ | $39.01_{0.90}$ | $14.15_{0.60}$ | $9.36_{0.75}$ | $1.21_{0.12}$ | $8.72_{0.43}$ |
| | BMLS | MS | $89.14_{0.63}$ | $\textbf{76.34}_{0.62}$ | $\textbf{74.63}_{0.69}$ | $\textbf{79.67}_{0.21}$ | $66.31_{0.26}$ | $\textbf{49.80}_{0.57}$ | $\textbf{21.80}_{0.40}$ | $\textbf{47.62}_{0.25}$ |
| imb 50 | *Sampler* | | | | | | | | | |
| | random | FC | $\textbf{95.25}_{0.23}$ | $78.52_{0.54}$ | $62.19_{1.04}$ | $78.64_{0.57}$ | $\textbf{73.72}_{0.18}$ | $48.62_{0.23}$ | $16.40_{0.93}$ | $48.31_{0.28}$ |
| | BMLS | FC | $91.86_{0.40}$ | $\textbf{81.32}_{0.36}$ | $\textbf{76.63}_{1.05}$ | $\textbf{83.07}_{0.43}$ | $69.77_{0.55}$ | $\textbf{54.55}_{0.28}$ | $\textbf{26.83}_{0.67}$ | $\textbf{51.99}_{0.26}$ |
| | *Classifier* | | | | | | | | | |
| | random | MS | $\textbf{90.04}_{0.63}$ | $64.80_{1.76}$ | $52.11_{1.06}$ | $68.56_{0.50}$ | $\textbf{68.28}_{0.69}$ | $42.17_{0.89}$ | $8.00_{0.52}$ | $41.66_{0.42}$ |
| | BMLS | $K^2$ | $59.75_{7.70}$ | $37.92_{0.59}$ | $30.37_{7.19}$ | $42.20_{0.89}$ | $13.25_{1.08}$ | $10.40_{0.49}$ | $2.81_{0.96}$ | $9.22_{0.39}$ |
| | BMLS | MS | $89.45_{0.15}$ | $\textbf{79.29}_{0.54}$ | $\textbf{83.03}_{0.50}$ | $\textbf{83.46}_{0.36}$ | $67.06_{0.51}$ | $\textbf{55.30}_{0.84}$ | $\textbf{31.88}_{1.39}$ | $\textbf{52.74}_{0.55}$ |
| imb 10 | *Sampler* | | | | | | | | | |
| | random | FC | $\textbf{94.79}_{0.55}$ | $85.38_{0.27}$ | $84.86_{1.23}$ | $88.05_{0.27}$ | $\textbf{76.06}_{0.32}$ | $64.10_{0.63}$ | $45.56_{0.57}$ | $63.03_{0.17}$ |
| | BMLS | FC | $91.17_{0.40}$ | $\textbf{87.04}_{0.26}$ | $\textbf{90.98}_{0.59}$ | $\textbf{89.46}_{0.19}$ | $71.05_{0.70}$ | $\textbf{68.93}_{0.66}$ | $\textbf{55.24}_{0.47}$ | $\textbf{65.72}_{0.29}$ |
| | *Classifier* | | | | | | | | | |
| | random | MS | $91.54_{0.43}$ | $76.31_{1.02}$ | $75.25_{1.44}$ | $80.56_{0.76}$ | $\textbf{71.91}_{0.35}$ | $57.38_{0.90}$ | $37.03_{0.72}$ | $56.71_{0.48}$ |
| | BMLS | $K^2$ | $57.78_{1.43}$ | $46.29_{2.03}$ | $52.52_{1.76}$ | $51.60_{0.99}$ | $17.09_{1.33}$ | $17.71_{0.71}$ | $13.97_{0.83}$ | $16.41_{0.51}$ |
| | BMLS | MS | $\textbf{91.63}_{0.60}$ | $\textbf{84.92}_{0.63}$ | $90.18_{0.56}$ | $\textbf{88.51}_{0.19}$ | $71.61_{0.21}$ | $\textbf{65.67}_{1.20}$ | $54.17_{1.49}$ | $\textbf{64.47}_{0.24}$ |

### F.3 An Empirical Study on Mixup Alpha

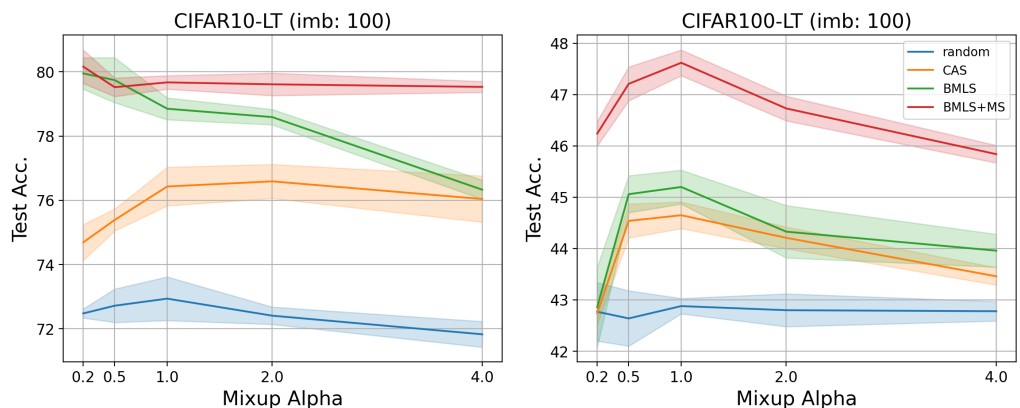

Figure 5: The change of test accuracy of each sampler on CIFAR10/100-LT (imb: 100)

Table 18: Ablation study on CIFAR10/100-LT datasets (imbalance factor: 100) with various mixup alpha values. The results are the mean of five repeated experiments with random seeds. Best in bold (CAS: Class-Aware Sampler, BMLS: Balanced Mixed Label Sampler)

| Method | CIFAR10-LT | | | | | CIFAR100-LT | | | | |
| | mixup alpha | | | | | mixup alpha | | | | |
| | 0.2 | 0.5 | 1.0 | 2.0 | 4.0 | 0.2 | 0.5 | 1.0 | 2.0 | 4.0 |
|---|---|---|---|---|---|---|---|---|---|---|
| random | 72.48 | 72.72 | 72.94 | 72.41 | 71.83 | 42.77 | 42.64 | 42.88 | 42.80 | 42.78 |
| CAS | 74.69 | 75.39 | 76.43 | 76.59 | 76.04 | 42.74 | 44.54 | 44.65 | 44.21 | 43.46 |
| BMLS | 79.95 | **79.74** | 78.85 | 78.59 | 76.33 | 42.86 | 45.06 | 45.20 | 44.33 | 43.96 |
| BMLS$_{MS}$ | **80.16** | 79.52 | **79.67** | **79.61** | **79.53** | **46.24** | **47.21** | **47.62** | **46.73** | **45.84** |

Table 19: Experimental results of Remix on Many/Medium/Few classes in the CIFAR10/100-LT datasets. The results are the mean of five repeated experiments with random seeds. (†: the reported values are taken from Chou et al. (2020), which used different experimental settings. ∗: the reproduced result of Remix on our experimental settings.)

| | Method | CIFAR10-LT | | | | CIFAR100-LT | | | |
| | | many | med | few | all | many | med | few | all |
|---|---|---|---|---|---|---|---|---|---|
| $\alpha = 0.2$ | random | **92.90**$_{1.54}$ | 72.28$_{2.47}$ | 52.34$_{4.80}$ | 72.48$_{0.14}$ | **71.69**$_{0.65}$ | 41.39$_{0.71}$ | 8.51$_{0.68}$ | 42.77$_{0.57}$ |
| | CAS | 89.05$_{4.00}$ | 72.29$_{3.03}$ | 63.53$_{7.84}$ | 74.69$_{0.56}$ | 66.31$_{0.40}$ | 43.65$_{0.57}$ | 12.39$_{0.45}$ | 42.74$_{0.22}$ |
| | BMLS | 89.73$_{0.57}$ | **76.84**$_{0.54}$ | 74.33$_{1.83}$ | 79.95$_{0.49}$ | 65.51$_{1.01}$ | 43.53$_{0.93}$ | 13.92$_{0.71}$ | 42.86$_{0.78}$ |
| | BMLS$_{MS}$ | 87.15$_{0.79}$ | 75.39$_{0.72}$ | **79.53**$_{1.80}$ | **80.16**$_{0.52}$ | 64.73$_{0.50}$ | **48.93**$_{0.62}$ | **20.06**$_{0.24}$ | **46.24**$_{0.24}$ |
| $\alpha = 0.5$ | random | 91.53$_{3.38}$ | 73.06$_{3.07}$ | 53.47$_{6.92}$ | 72.72$_{0.52}$ | **72.42**$_{0.10}$ | 40.62$_{0.90}$ | 8.12$_{0.78}$ | 42.64$_{0.54}$ |
| | CAS | **91.74**$_{2.24}$ | 74.91$_{2.60}$ | 59.66$_{4.84}$ | 75.39$_{0.34}$ | 68.63$_{0.25}$ | 45.83$_{0.33}$ | 13.08$_{0.71}$ | 44.54$_{0.33}$ |
| | BMLS | 90.96$_{0.58}$ | **78.39**$_{0.40}$ | 70.34$_{2.05}$ | **79.74**$_{0.70}$ | 67.86$_{0.68}$ | 46.46$_{0.31}$ | 15.06$_{1.23}$ | 45.06$_{0.36}$ |
| | BMLS$_{MS}$ | 88.88$_{1.02}$ | 75.65$_{0.56}$ | 75.31$_{1.21}$ | 79.52$_{0.29}$ | 64.13$_{1.12}$ | **50.41**$_{0.82}$ | **22.34**$_{0.45}$ | **47.21**$_{0.33}$ |
| $\alpha = 1.0$ | random | **93.39**$_{2.42}$ | 74.05$_{2.03}$ | 50.99$_{5.40}$ | 72.94$_{0.68}$ | **72.09**$_{0.21}$ | 41.10$_{0.40}$ | 8.77$_{0.42}$ | 42.88$_{0.15}$ |
| | CAS | 90.54$_{2.86}$ | 75.54$_{1.95}$ | 63.51$_{6.26}$ | 76.43$_{0.60}$ | 68.28$_{0.35}$ | 46.47$_{0.34}$ | 13.12$_{0.25}$ | 44.65$_{0.26}$ |
| | BMLS | 88.53$_{1.01}$ | **77.84**$_{0.25}$ | 70.53$_{1.27}$ | 78.85$_{0.34}$ | 68.38$_{0.27}$ | 46.89$_{0.33}$ | 14.37$_{0.88}$ | 45.20$_{0.33}$ |
| | BMLS$_{MS}$ | 89.14$_{0.63}$ | 76.34$_{0.62}$ | **74.63**$_{0.69}$ | **79.67**$_{0.21}$ | 66.31$_{0.26}$ | **49.80**$_{0.57}$ | **21.80**$_{0.40}$ | **47.62**$_{0.25}$ |
| $\alpha = 2.0$ | random | **93.97**$_{0.29}$ | 73.29$_{0.41}$ | 49.65$_{1.54}$ | 72.41$_{0.27}$ | **71.92**$_{0.39}$ | 41.45$_{0.71}$ | 8.30$_{0.23}$ | 42.80$_{0.32}$ |
| | CAS | 88.30$_{3.08}$ | 75.95$_{2.20}$ | 65.74$_{5.88}$ | 76.59$_{0.53}$ | 66.38$_{0.68}$ | 47.04$_{0.27}$ | 13.27$_{0.46}$ | 44.21$_{0.21}$ |
| | BMLS | 88.17$_{0.57}$ | **77.24**$_{1.02}$ | 70.80$_{1.53}$ | 78.59$_{0.24}$ | 66.73$_{1.07}$ | 47.30$_{0.71}$ | 12.92$_{0.89}$ | 44.33$_{0.51}$ |
| | BMLS$_{MS}$ | 84.22$_{0.41}$ | 74.66$_{0.72}$ | **81.59**$_{0.31}$ | **79.61**$_{0.35}$ | 65.15$_{1.60}$ | **49.53**$_{1.26}$ | **20.47**$_{0.41}$ | **46.73**$_{0.24}$ |
| $\alpha = 4.0$ | random | **93.18**$_{0.30}$ | 71.64$_{0.97}$ | 50.75$_{0.43}$ | 71.83$_{0.40}$ | **71.84**$_{0.31}$ | 41.72$_{0.37}$ | 7.99$_{0.76}$ | 42.78$_{0.19}$ |
| | CAS | 87.37$_{3.19}$ | 75.20$_{2.05}$ | 65.84$_{6.09}$ | 76.04$_{0.72}$ | 64.18$_{0.54}$ | 47.25$_{0.25}$ | 13.15$_{1.00}$ | 43.46$_{0.17}$ |
| | BMLS | 86.59$_{2.05}$ | **77.49**$_{0.78}$ | 64.52$_{2.86}$ | 76.33$_{0.30}$ | 64.02$_{0.24}$ | 47.23$_{0.53}$ | 15.09$_{0.48}$ | 43.96$_{0.32}$ |
| | BMLS$_{MS}$ | 86.57$_{0.79}$ | 73.88$_{0.86}$ | **80.03**$_{0.89}$ | **79.53**$_{0.17}$ | 61.20$_{1.02}$ | **50.70**$_{1.28}$ | **20.89**$_{0.55}$ | **45.84**$_{0.17}$ |

