# OpenReview forum: "Balancing Mixed Labels: Mixup meets Neural Collapse in Imbalanced Learning"
_ICLR.cc/2026/Conference — ICLR 2026 Conference Withdrawn Submission_

### Official Review · Reviewer_ucbY · 2025-10-27

**Soundness:** 2
**Presentation:** 2
**Contribution:** 2
**Rating:** 2
**Confidence:** 4

**Summary:**

This work addresses minority collapse in imbalanced learning by proposing to balance mixed-label samples generated by Mixup. The authors introduce a Balanced Mixed Label Sampler (BMLS) and a Mixed-Singleton Classifier (MS), demonstrating that their approach yields performance gains, particularly for minority classes,

**Strengths:**

1. The paper addresses a pertinent issue in imbalanced learning by investigating the often-overlooked factor of mixed-label balance in conjunction with Mixup.
2. The theoretical analysis, grounded in the Layer-Peeled Model and Neural Collapse, provides a structured lens through which to analyze the problem.

**Weaknesses:**

1. Insufficient and Outdated Empirical Comparisons:
i) The empirical evaluation primarily benchmarks against methods up to 2022 (e.g., LOM, CAS), omitting comparisons with more recent and potent state-of-the-art techniques. This selective baseline choice casts doubt on the claimed effectiveness and fails to situate the contribution within the current landscape.
ii) Crucially, several highly relevant and strong baselines, such as Remix [a] (which explicitly manipulates mixed labels for balance) and other well-established methods (e.g., [b-c]), are either absent or relegated to a non-comparative grayed-out section in Table 7.
2. Theoretical Analysis Lacks Rigor and Practical Alignment:
i)  Key components of the theoretical framework (e.g., Lemma 2, Proposition 1) are not self-contained but instead heavily rely on citations without verifying their direct applicability in the novel context of mixed labels. This undermines the theoretical contribution's solidity.
ii) The convexity assumption of the loss function contradicts the non-convex nature of training deep networks with cross-entropy loss. The paper does not adequately justify how the semidefinite relaxation (Eq. 11) bridges this gap.
iii) The analysis is based on the simplified Layer-Peeled Model, but the experiments employ standard deep networks. The paper does not provide evidence that the behaviors analyzed in LPM faithfully approximate those of the deep models used in practice.
3. The core innovation of interpreting mixed labels as singletons is not conclusively validated. The performance gain of MS could stem merely from increasing the classifier's capacity to K^2 units. A controlled ablation, using a classifier with K^2 randomly initialized or fixed ETF units, is needed to isolate the benefit of the specific mixed-singleton construction.
4. The mixup coefficient λ is a critical hyperparameter that governs the interpolation between samples and labels. It is surprising that the paper contains no sensitivity analysis on λ or the parameter α of the Beta distribution it is sampled from. The performance and the proposed theoretical claims (e.g., independence from the mixup ratio) could be highly sensitive to this parameter. The robustness of BMLS and MS across different λ regimes remains an open and unverified question.

References:
[a] Chou et al. Remix: rebalanced mixup. ECCV'2020.
[b] Cui et al. Parametric contrastive learning. ICCV'2021.
[c] Yang et al. Inducing neural collapse in imbalanced learning: Do we really need a learnable classifier at the end of deep neural network? NeurIPS'2022.

**Questions:**

1. Why were more recent and directly relevant strong baselines (e.g., [a-c]) not included in the primary comparisons? Can the authors provide results against these methods?
2. Could the authors provide more detailed proofs or intuitive justification for the applicability of Lemma 2 and Proposition 1 in the mixed-label setting, which is central to this work?
3. How does the semidefinite relaxation (Eq. 11) handle the non-convexity inherent in training deep networks? Is there empirical evidence that the solutions of the LPM align with the features and classifiers learned by the deep models in the experiments?
4. Can the authors conduct an ablation study using a classifier with K^2 singleton units not constructed via mixup? This would help decouple the effect of increased classifier capacity from the specific mixed-singleton interpretation.
5. What is the sensitivity of the proposed methods to the mixup hyperparameters (λ or α)? Please provide a sensitivity analysis to demonstrate the robustness of your findings across different settings.

---

> ### Author Response · Authors · 2025-11-25
> **Part 1 for W1 and Q1**
>
> ### W1 & Q1. Insufficient and Outdated Empirical Comparisons
> **i)** We appreciate the reviewer’s effort to consider more recent methods when evaluating the contribution of our proposed approach. As pointed out, the primary baselines we targeted—LOM(2022) and CAS(2016)—are not among the most recent methods. To address this, as shown in *Table 7* of *Appendix E*, we additionally referenced recent Mixup-based methods such as OTMix(2023), SBN-mix(2024), DBN-mix(2024), and CP-mix(2025). Although we did not directly reproduce all of these methods, we compared our results with the performance reported in their respective reference papers. Notably, our proposed method outperformed other Mixup-based approaches.
>
> In this work, we intentionally restricted our comparison to **one-stage training frameworks with a single architecture.** Therefore, methods such as CP-mix that rely on two-stage training or require extra networks were treated as non-target baselines and grayed out in the table to avoid direct comparison.
>
> **ii)** In the same spirit, contrastive learning approaches were excluded from our comparison scope, and consequently, PaCo[b] was also considered out of scope.
>
> Unfortunately, because we initially omitted a citation for a fixed ETF classifier[c] in the *Table 5* in *Section 5.4*, it may have appeared that the fixed ETF classifier was not considered, even though we already referenced its reported performance and incorporated it into our comparison. We have now explicitly added the citation in the *Table 5* to clarify that the fixed ETF classifier's results were taken directly from the reference paper.
>
> Finally, we additionally reproduced and compared Remix[a]. As shown in *Tables 14 and 15* in *Appendix F.1*, our proposed method outperforms the baseline Remix as well as Remix combined with other class-balanced samplers. Moreover, because our method is not incompatible with Remix but can be jointly applied, we integrated our method into Remix. The integrated model showed improved performance over the original Remix, providing further rationale for the *Integration with other Mixup-based methods* pathway described in our future work (*Section 7* in the revised manuscript).

---

> ### Author Response · Authors · 2025-11-25
> **Part 2 for W2, Q2, and Q3**
>
> ### W2-1 & Q2. Clarify the omitted mathematical statements
> To address the reviewer’s concern regarding the omitted proofs of *Lemma 2* and *Proposition 1* in the theoretical framework, we have provided the detailed proofs of them in *Appendix C.3 (lines 1208–1544)* and *Appendix C.2*, respectively.
>
> ---
>
> ### W2-2 & Q3. Convexity assumption
> We appreciate the opportunity to discuss the reviewer’s concern regarding the convexity assumption of the loss function contradicting the non-convex nature of training deep networks with cross-entropy loss. As supporting evidence, we note that [A1] has shown that minimizing an indefinite quadratic function can be solved via an semidefinite program (SDP) formulation. Building on this result, [A2] relaxed the Layer-Peeled Model with cross-entropy loss—originally formulated as a quadratically constrained quadratic program—into an SDP, and proved that the solution of the relaxed SDP coincides with that of the original non-convex program when the inequality constraint becomes an equality.
>
> Following the same reasoning, our paper proves that the minimizer of the convex program obtained via SDP relaxation (*Eq. 11*) recovers the solution of LPM$\_{\lambda}$ (*Eq. 5*) when the equality constraint holds (see *Appendix C.1*).
>
> A potential concern may be the applicability of SDP beyond the LPM setting, particularly for deep networks with multiple layers. However, [A3] demonstrates that the safety of an abstracted network—sufficient for ensuring the safety of the original deep network—can be analyzed using semidefinite programming. As further discussed in [A4], SDP relaxations of quadratically constrained quadratic programs generally provide approximate solutions with bounded gaps for deep networks.
>
> ---
>
> ### W2-3. The practical alignment between LPM and standard deep networks
> Neural Collapse (NC), first identified in the terminal phase of training in [A5], characterizes the geometric properties that emerge between last-layer features and the classifier in ideally trained deep networks. Leveraging these theoretical benefits, subsequent studies have employed NC to improve model performance across various domains (e.g., exemplar-free class-incremental learning[A7], imbalanced learning[c], knowledge distillation[A8], semantic segmentation[A9]).
> Therefore, as argued in [A2], *the presence of neural collapse in the Layer-Peeled Model offers evidence of the effectiveness of the LPM as a tool for analyzing neural networks*, and LPM has since been adopted as an analytical model in later NC-based researches[c, A6].
>
> ---
>
> [A1] Sturm and Zhang, On Cones of Nonnegative Quadratic Function, Mathematics of Operations Research 2023
>
> [A2] Fang et al., Exploring deep neural networks via layer-peeled model: Minority collapse in imbalanced training, PNAS 2021
>
> [A3] Fazlyab et al., Probabilistic Verification and Reachability Analysis of Neural Networks via Semidefinite Programming, IEEE CDC 2019
>
> [A4] Azuma et al., Tight Semidefinite Relaxations for Verifying Robustness of Neural Networks, arXiv 2025
>
> [A5] Papyan et al., Prevalence of neural collapse during the terminal phase of deep learning training, PNAS 2020
>
> [A6] Fisher et al., Pushing Boundaries: Mixup's Influence on Neural Collapse, ICLR 2024
>
> [A7] Yang et al., Neural Collapse Inspired Feature-Classifier Alignment for Few-Shot Class-Incremental Learning, ICLR 2023
>
> [A8] Zhang et al., Neural Collapse Inspired Knowledge Distillation, AAAI 2025
>
> [A9] Zhong et al., Understanding Imbalanced Semantic Segmentation Through Neural Collapse, CVPR 2023

---

> ### Author Response · Authors · 2025-11-25
> **Part 3 for W3, Q4, W4, and Q5**
>
> ### W3 & Q4. $K^2$ classifier
> We thank the reviewer for the sharp insight. We agree with the reviewer’s perspective and conducted an ablation study on the $K^2$ classifier to verify that the performance gain of our proposed method MS does not simply stem from an increase in classifier capacity. The results of this experiment are presented in *Tables 16 and 17* in *Appendix F.2*.
> As shown in the results, the $K^2$ classifier performs worse than MS alone, and even worse than when MS is combined with a random sampler. This degradation occurs because the use of a $K^2$ classifier drastically reduces the number of samples available to learn each class vector, leading to underfitting due to insufficient class-vector learning.
> Through this experiment, we empirically confirm that **the performance improvement of MS is not attributable to increased classifier capacity, but rather to the effect of the linear interpolation between class vectors induced by mixup ratio $\lambda$.**
>
> ---
>
> ### W4 & Q5. An empirical study on Mixup alpha $\alpha$
> We thank the reviewer for raising this concern. To address this issue, we conducted an empirical study across various Mixup alphas ($\\alpha \in \\{0.2, 0.5, 1.0^*, 2.0, 4.0\\}$, where * denotes the ratio used in our main experiments). The results are presented in *Figure 5* and *Tables 18 and 19* in *Appendix F.3*.
> As shown in the experiments, we observe that **for any positive choice of $\\alpha$, incorporating our proposed method consistently outperforms both the random sampler and CAS.** While hyperparameter tuning of the Mixup alpha $\\alpha$ affects the magnitude of improvement, it does not alter the conclusion itself—namely, that the performance improvements do not critically depend on the specific value of $\alpha$.

---

> ### Comment · Reviewer_ucbY · 2025-11-27
> **Response to the Authors‘ Rebuttal**
>
> Thank you for the revision and rebuttal. I appreciate the added material, but I still have the following concerns:
> 1. Empirical evaluation and SOTA positioning are still not convincing.
> Although you cite several recent Mixup/long-tailed methods, most are not reimplemented under a unified training and evaluation protocol, so the comparisons remain indirect. In this highly recipe-sensitive setting, cross-paper reported numbers cannot reliably support superiority claims. The primary tables are still centered on outdated baselines (e.g., CAS/LOM), and strong recent methods are excluded via a narrow “one-stage + single-architecture” scope, which is misaligned with the paper’s broad practical claims. As a result, the true SOTA standing of the proposed approach remains unclear.
> 2. Theory–practice alignment is still weak.
> While the missing proofs are now provided and Eq. 11 is rigorous within the LPM abstraction, the conclusions rely on a simplified, convexity-style LPM/SDP analysis, whereas the experiments use standard deep networks trained with non-convex cross-entropy. Prior NC/LPM citations do not substitute for direct evidence that the mixed-label dynamics predicted by LPM faithfully track real deep-network behavior in this setting. This gap makes the theoretical claims more suggestive than confirmatory.

---

> ### Author Response · Authors · 2025-12-03
>
> ### R1. Empirical evaluation and SOTA positioning are still insufficient
>
> Although the reproducing and comparing experiments with ETF+DR and Remix requested in the initial comment have been conducted, the reviewer again raised concerns about our empirical evaluation. We interpret this as stemming from the fact that, **although we argue that BMLS+MS is effective “from theory to remedy,” our empirical comparisons were limited to a constrained setting (one-stage, single-architecture) and to a restricted set of methods (primarily class-balanced samplers such as CAS and LOM).**
>
> We agree with this reviewer’s concern about SOTA positioning. In response, we additionally included experiments comparing our approach against DBN-mix, the strongest-performing method among our baselines. **As shown in Table 2, BMLS$\_{\text{MS}}$, when integrated with DBN-mix, improves the performance of DBN-mix by up to 4.94%p**. These results demonstrate that **our proposed method achieves SOTA-competitive performance, thereby addressing both the reviewer’s concerns regarding empirical evaluation and SOTA positioning.** The results for DBN-mix, as well as those for Remix and ETF+DR, have been added to the main table (**Table 2**), and the corresponding discussion and implementation details have been updated in **Section 5.2** and **Appendix D**.
>
> At the same time, we emphasize that, despite these strong empirical results, our work should not be viewed as purely empirical. As summarized in the *“Official Comment by Authors > An Executive Summary of Mathematical Statements (2/2) > Remark of our theoretical contribution,”* we reveal that BMLS is grounded in solid theoretical justification and, consequently, exhibits superior minority collapse mitigation compared with other samplers in empirical evaluations. Furthermore, through integration experiments with a range of Mixup-based methods—including Remix, ETF+DR, and DBN-mix—we demonstrate that **our proposed approach integrates seamlessly with them, suggesting it can serve as an effective sampler and classifier to achieve SOTA-level performance.**
>
> ---
>
> ### R2. Theory-practice misalignment
> As we understand it, the reviewer’s concerns center around two key themes: **shallow network** and **convexity**.
>
> **Shallow network (the concern about LPM).**
> Even though the LPM analyzes only the relationship between the last-layer features and the classifier, *the presence of NC in LPM offers evidence of the effectiveness of the LPM as a tool for analyzing neural networks* [A2]. As noted in our earlier response, this is already a well-established fact, and the stream of research on LPM analysis has been developed based on this premise [c, A6].
> Therefore, **the presence of NC in LPM$\_{\lambda}$ and LPM$\_{\lambda}$-MS also provides evidence supporting their effectiveness as analytical tools for understanding deep neural networks.**
>
> **Convexity (the concern about SDP).**
> When analyzing the LPM together with the non-convex cross-entropy loss, the SDP relaxation enables the use of a convex optimization program. As mentioned earlier, **the solution of the relaxed SDP coincides with that of the original non-convex program when the inequality constraint becomes an equality**, showing that the relaxation is valid for analyzing the non-convex objective. Furthermore, as noted in our response to the initial comment, *SDP relaxations of quadratically constrained quadratic programs often yield approximate solutions with bounded gaps even in the context of deep networks* [A4]. Therefore, the SDP relaxation in the LPM with CE loss is also an effective tool for deep networks.
>
> Building on these observations, we conclude that **analyzing the presence of NC in LPM$\_{\lambda}$ and LPM$\_{\lambda}$-MS via the SDP relaxation provides a practical theoretical approach for understanding deep neural networks.**
>
> Consequently, the methods we develop—**BMLS** and **MS**, inspired by LPM$\_{\lambda}$ and LPM$\_{\lambda}$-MS—can likewise be considered empirically valid for deep networks, in the same way prior NC/LPM-based work has [c, A7-9].
>
> +Separately, and beyond the scope of NC/LPM/SDP, we would emphasize that many methods inspired by theoretical analyses conducted under relaxed settings have been successfully applied to modern deep networks, which are too complex to analyze directly, yet exhibit strong empirical performance (e.g., mini-batch SGD).

---

> ### Author Response · Authors · 2025-12-03
> **Summary of the Discussion**
>
> Based on the reviewer’s response, we conclude that the concerns related to **W2-1 & Q2**, **W3 & Q4**, and **W4 & Q5** have been adequately addressed.
> * **W2-1 & Q2**: The reviewer said *"the missing proofs are now provided and Eq. 11 is rigorous within the LPM abstraction"*, indicating that the reviewer acknowledged that their concern regarding this point has been resolved.
> * **W3 & Q4** and **W4 & Q5**: The reviewer did not raise any additional or remaining concerns regarding these points.
>
> | Comment                                                             | Added or modified in revision              | Discussed                                                                                                                                                                                                                                                                                                                                                                                                                                                                       |
> | ------------------------------------------------------------------- | ------------------------------------------ | ------------------------------------------------------------------------------------------------------------------------------------------------------------------------------------------------------------------------------------------------------------------------------------------------------------------------------------------------------------------------------------------------------------------------------------------------------------------------------- |
> | **W1, Q1, and R1.** Insufficient and Outdated Empirical Comparisons | Table 2, Section 5.2, Appendix D           | As shown in **Table 2**, BMLS$\_{\text{MS}}$, when integrated with DBN-mix, improves the performance of DBN-mix by up to 4.94%p. Through integration experiments with a range of Mixup-based methods, including Remix, ETF+DR, and DBN-mix, we demonstrate that our proposed approach integrates seamlessly with them, suggesting it can serve as an effective sampler and classifier to achieve SOTA-level performance.                                                         |
> | **W2-1 & Q2.** Clarify the omitted mathematical statements          | Appendix C.2 and C.3                       | We have provided the detailed proofs of **Lemma 2** and **Proposition 1** in **Appendix C.3 (lines 1208–1544)** and **Appendix C.2**, respectively.                                                                                                                                                                                                                                                                                                                             |
> | **W2-2, W2-3, Q3, and R2.** Theory-practice misalignment            |                                            | We addressed the reviewer’s concern by presenting multiple previously validated references demonstrating why LPM and SDP relaxations serve as effective analytical tools for understanding deep networks. Building on these foundations, we provided a justified rationale that our analysis—examining the presence of NC in LPM$\_{\lambda}$ and LPM$\_{\lambda}$-MS via the SDP relaxation—constitutes a practical theoretical approach for understanding deep neural networks. |
> | **W3 & Q4.** $K^2$ classifier                                       | Tables 16 and 17 in Appendix F.2           | Through the ablation study including $K^2$ classifier, we empirically confirm that the performance improvement of MS is not attributable to increased classifier capacity, but rather to the effect of the linear interpolation between class vectors induced by mixup ratio $\lambda$.                                                                                                                                                                                         |
> | **W4 & Q5.** An empirical study on Mixup alpha                      | Figure 5, Tables 18 and 19 in Appendix F.3 | As shown in the empirical study on Mixup alpha, we observe that for any positive choice of $\alpha$, incorporating our proposed method consistently outperforms both the random sampler and CAS, which means the performance improvements do not critically depend on the specific value of $\alpha$.                                                                                                                                                                           |

---

### Official Review · Reviewer_a1Sm · 2025-10-31

**Soundness:** 2
**Presentation:** 1
**Contribution:** 3
**Rating:** 2
**Confidence:** 3

**Summary:**

This paper addresses the problem of minority class disappearance in imbalanced learning settings. The authors analyze Mixup from the perspective of neural collapse (NC) and propose BMLS and MS as methods to tackle this issue and enhance performance. They demonstrate through experiments that the proposed architecture achieves improved performance compared to existing approaches.

**Strengths:**

The paper analyzes Mixup*from the perspective of neural collapse (NC) and proposes a new architecture to address the problem of minority class disappearance. This approach leads to improved performance.

**Weaknesses:**

A major concern lies in the clarity of the claims and the appropriateness of the theoretical justification presented in the paper.

- It is difficult to clearly understand what the main claim of this paper is. The work appears to aim at improving Mixup from the perspective of neural collapse (NC). In this context, the proof presented in Section 5 seems to show that the proposed architecture satisfies the NC-related properties that the authors consider desirable. There appear to be two possible interpretations:
    (a) The claim that satisfying these NC properties leads to improved performance is already well established, and the authors are showing that their proposed architecture indeed satisfies them — in which case the proof can be seen as providing theoretical guarantees.
    (b) The authors themselves claim that these NC properties are beneficial and then show that their architecture satisfies them — in which case the argument functions more as motivation and background for the proposed architecture.

    These two interpretations differ significantly in terms of their scientific contribution, and it is important to make clear which one the paper intends. At present, I interpret the work as closer to (b). In that case, recognizing both the novelty of the proposed method and the value of theoretical guarantees simultaneously risks falling into circular reasoning, and therefore cannot be straightforwardly accepted.

- The intended effect of the proposed method is also unclear. While the disappearance of minority classes is identified as the main challenge, the experiments report only mean accuracy. It is not clear how this metric is defined, and more importantly, mean accuracy alone does not demonstrate improvements specifically for the minority classes. To substantiate the claimed contribution, the paper should report results that directly assess performance on the minority classes. As it stands, it is difficult to judge whether the proposed method truly achieves the effect it claims.

- Section 5 presents only a sketch of a proof, introducing new concepts without clearly stating what is actually being proven. As a result, it is very difficult to grasp the main claim or the logical structure of the argument.

**Questions:**

Please clarify which of the two cases — **(a)** or **(b)** — represents the main claim of the paper.

Can you also demonstrate through experiments how the proposed method affects the minority classes specifically?

Finally, please explicitly state what is being proven in Section 5 — i.e., clearly formulate the statement or proposition that the proof is intended to establish.

---

> ### Author Response · Authors · 2025-11-25
> **Part 1 for W1 and W3**
>
> ### W1 & W3. Clarity of the claims and Appropriateness of the theoretical justification
> We thank the reviewer for their thoughtful comments and for taking the time to understand our claim. We acknowledge that the structure of the original manuscript—built around extending existing theory to the Mixup setting, identifying a new problem, analyzing its cause, and proposing an effective solution—may have obscured the clarity of the main claim and its theoretical justification. To address this, we have reorganized the logical flow as outlined below and made explicit the theoretical foundations supporting each step. Accordingly, we have made minor structural revisions to the paper.
>
> 1. We first demonstrate that **minority collapse still occurs under Mixup when the frequency of mixed labels is not balanced** (*Theorem 1*).
> 2. Although existing class-balanced samplers partially mitigate the minority collapse of Mixup by balancing singleton-label frequencies, they fail to resolve the issue due to the inherent randomness of Mixup.
> 3. To obtain empirical evidence for this limitation, we examined per-label frequencies generated in each epoch and observed an **epoch-wise label imbalance phenomenon** (*Figure 2*).
> 4. Through a mixed-label frequency control experiment (*Figure 3*), we further empirically verified that this imbalance substantially weakens the mitigation of minority collapse under Mixup.
> 5. To address this issue, we propose the ***B**alanced **M**ixed **L**abel **S**ampler (BMLS)*, which balances mixed-label frequencies across epochs (*Section 3*).
> 6. Both theoretically and empirically, we demonstrate that **aligning mixed-label frequencies across epochs mitigates minority collapse** (*Proposition 1* and *Figure 4*).
> 7. Moreover, our analysis shows that the minority collapse of Mixup is determined solely by the frequencies of singleton and mixed labels, independent of the Mixup ratio.
> 8. Leveraging this insight, we introduce the ***M**ixed-**S**ingleton (MS) classifier*, which treats mixed labels as singleton labels when learning class vectors (*Section 3*).
> 9. Compared with a conventional singleton classifier implemented as a fully connected layer, our approach achieves superior performance, particularly improving accuracy on minority classes (*Table 1*).
>
> These revisions will be reflected in the updated main paper. Importantly, these changes do not add new contributions; rather, they reorganize the presentation of the existing problem, its theoretical and empirical validation, and the proposed solutions.
>
> Additionally, to reduce confusion in *Section 5* (renamed *Section 4* in the revised manuscript), we clarify the connection between the proposed methods and their theoretical justification directly within Section 3 (see *Lines 132–136* and *Lines 147–150*). In the revised Section 4, after each theorem and proposition, we also move explicit remarks (*Lines 251–256* and *Lines 320–323*) at the end of each subsection to reinforce the main claim and highlight the logical structure of our argument.

---

> ### Author Response · Authors · 2025-11-25
> **Part 2 for Q1, Q3, W2, and Q2**
>
> ### Q1. Clarify which of the cases (a) or (b) represents the main claim
> [A1] proposes the Layer-Peeled Model (LPM), an analytical model that uses only the last-layer features and the classifier, and demonstrates that LPM satisfies the NC properties. Building on this model, [A1] theoretically proves that minority collapse arises in imbalanced settings and that oversampling minority classes mitigates this issue. Following the same logical structure, as outlined in *Section 4.1, Proof Sketch* in the revised manuscript, we extend this framework to the Mixup setting.
> As a result, the main claim of our paper is that _minority collapse also occurs in LPM with Mixup_, and we establish this by leveraging the NC properties, particularly the simplex ETF structure. Therefore, to clearly summarize our claim in a single sentence: **we theoretically prove, using NC, that the imbalance frequency of mixed labels induces minority collapse in LPM with Mixup under imbalanced learning**.
> For this reason, our proposed method is not directly related to NC; that is, it neither benefits from NC properties nor aims to induce NC. For further clarification regarding the relationship or logical flow between our main claim and the proposed method, please refer to our responses to W1 and W3 above.
>
> ---
>
> ### Q3. Explicitly state what is being proven in *Section 4. Theoretical analysis*
> Since detailed answers to this question have already been provided in our responses to W1 and W3, we summarize here only the key remarkable points of the mathematical statements covered in the theoretical analysis section, avoiding unnecessary repetition:
> * *Lemma 1 + Theorem 1:* **Minority collapse still occurs under Mixup when the frequency of mixed labels is not balanced**.
> * *Proposition 1*: **Balancing the frequency of mixed labels across epochs mitigates minority collapse.**
> * *Theorem 2*: The properties and behaviors observed in the LPM **remain unchanged even when the Mixed-Singleton classifier is applied**, thereby theoretically confirming that **BMLS remains effective in settings that incorporate MS.**
>
> ---
>
> ### W2 & Q2. Detailed criteria and descriptions of the evaluation results reported in the tables
> We are pleased to provide a detailed explanation regarding this point. We believe the reviewer’s concern arose because, although the experimental results referred the reader to the appendix, the original manuscript did not explicitly state what information the appendix contained. To address this, we added a clarification in the beginning of *Section 5, Experimental Results*—specifically at *Lines 360–361*—indicating which evaluation criteria and details can be found in the appendix for each table.
>
> To elaborate on the reviewer’s question: following the criteria used in [A2], we categorize each class into one of three groups—*many*, *median*, and *few*. The mean accuracy reported in each table corresponds to the average top-1 test accuracy across different seeds within each category. If a table does not distinguish categories, the mean accuracy refers to the top-1 test accuracy averaged across *all* classes. Accordingly, the standard deviation associated with each mean accuracy can also be computed, and these values are reported in the tables in *Appendix E*.
>
> This categorization additionally allows us to quantitatively assess model performance on minority classes (*i.e.*, classes belonging to the *few* category). Indeed, as shown in Tables 1, 8–13, 15, 17, and 19, our proposed method consistently improves accuracy on the *few* category, demonstrating its effectiveness to mitigate the minority collapse.
>
> [A1] Fang et al., Exploring deep neural networks via layer-peeled model: Minority collapse in imbalanced training, PNAS 2021
>
> [A2] Liu et al., Large-Scale Long-Tailed Recognition in an Open World, CVPR 2019

---

> ### Author Response · Authors · 2025-12-03
> **Summary of the discussion**
>
> Added or modified in revision
> * **Lines 50-77, 132-136, 147-150, 251-256, 320-323** for **W1 & W3**
> * **Lines 360–361** for **W2 & Q2**
>
> | Comment                                                                                         | Discussed                                                                                                                                                                                                                                                                                                                                                                                                                                                                                                                              |
> | ----------------------------------------------------------------------------------------------- | -------------------------------------------------------------------------------------------------------------------------------------------------------------------------------------------------------------------------------------------------------------------------------------------------------------------------------------------------------------------------------------------------------------------------------------------------------------------------------------------------------------------------------------- |
> | **W1 & W3.** Clarity of the claims and Appropriateness of the theoretical justification         | We have reorganized the logical flow and made explicit the theoretical foundations supporting each step. Accordingly, we have made minor structural revisions to the paper.                                                                                                                                                                                                                                                                                                                                                            |
> | **Q1.** Clarify which of the cases (a) or (b) represents the main claim                         | To clearly summarize our claim in a single sentence: we theoretically prove, using NC, that the imbalance frequency of mixed labels induces minority collapse in LPM with Mixup under imbalanced learning.                                                                                                                                                                                                                                                                                                                             |
> | **Q3.** Explicitly state what is being proven in Section 4. Theoretical analysis                | We summarize the key remarkable points of the mathematical statements covered in the theoretical analysis section.                                                                                                                                                                                                                                                                                                                                                                                                                     |
> | **W2 & Q2.** Detailed criteria and descriptions of the evaluation results reported in the table | The reviewer’s concerns regarding the interpretation of mean accuracy and the perceived absence of results for the few category arose because these points were not sufficiently emphasized in the original manuscript, even though the information was already included. We resolved this issue by adding a clear statement at the beginning of **Section 5**, indicating where the evaluation criteria and detailed descriptions for each table can be found in the appendix (see **Appendix E** and **Tables 1, 5, 8–13, 15, 17, 19**). |

---

### Official Review · Reviewer_GkCW · 2025-10-31

**Soundness:** 3
**Presentation:** 3
**Contribution:** 3
**Rating:** 6
**Confidence:** 3

**Summary:**

The paper addresses minority collapse in long-tailed learning under Mixup.
 It argues that collapse stems from imbalanced frequencies of mixed labels (pair-level imbalance) rather than class imbalance alone.
 To fix this, the authors propose: 1. Balanced Mixed Label Sampler (BMLS) – equalizes the occurrence of every label pair (a,b); 2. Mixed-Singleton Classifier (MS) – treats each mixed label as a new singleton class.
 They provide theoretical analysis via the Layer-Peeled Model (LPM) showing that pair-level balance mitigates collapse, and validate the claim empirically on multiple benchmarks.

**Strengths:**

1. Originality: The paper introduces a new perspective — that pair-level imbalance (in mixed labels) is the true driver of collapse under Mixup.

2. Quality: The theoretical analysis seems rigorous and well-motivated. Experiments align well with theory and are conducted carefully across multiple benchmarks.

3. Clarity: Well-organized paper with clear motivation–theory–experiment flow.

**Weaknesses:**

**1. Theoretical Limitations:**
 The claim that MS is “equivalent to an LPM with K^2 classes” is only approximate. Since mixed-class weights W^\lambda_{(a,b)} = \lambda w_a + (1-\lambda) w_b are not independent, the coupling among weights might alter the geometric equilibrium. A formal treatment of this dependency would strengthen the theory.


**2. Assumption Robustness:**  The main theoretical result assumes perfectly uniform pair sampling (P(a,b) constant). In practice, BMLS approximates this stochastically per mini-batch, which may not fully satisfy uniformity. A robustness analysis for imperfect balance or bounded variance would make the theoretical claims more convincing.


**3. Scalability** :  While the authors note scalability as a theoretical limitation—due to the quadratic growth of label pairs with KKK—this also manifests empirically: the method performs strongly on smaller datasets (e.g., CIFAR-LT) but yields weaker gains on large-scale benchmarks such as ImageNet-LT, Places-LT, and iNaturalist2018.
This suggests that the pair-level balancing principle, though theoretically sound, may not fully scale in practice. Further analysis of this behavior or mechanisms to preserve balance under high-class regimes would strengthen the method’s generalization impact.

**Questions:**

Please refer to the weaknesses above.

1. Could the authors clarify whether the K^2-class equivalence of MS is purely theoretical or affects actual classifier behavior?


2. How sensitive is the theoretical guarantee to deviations from perfectly uniform pair sampling? Have the authors observed performance degradation when the pair distribution is only approximately balanced?


3. Could the authors elaborate on why the method’s gains diminish on large-scale datasets? Do they attribute this to computational constraints, representation saturation, or limitations of pair-level balancing itself?

---

> ### Author Response · Authors · 2025-11-25
> **Part 1 for W1, Q1, W2, and Q2**
>
> ### W1. Theoretical limitations about the linear dependency of mixed-class weights $W^\lambda_{(a,b)}$
> We thank the reviewer for their sharp insight and are pleased to provide a detailed proof on this point. To thoroughly verify the claim, we conducted a theoretical analysis of the LPM$\_{\lambda}$-MS model. As a results, we conclude that **LPM$\_{\lambda}$-MS has the same global minimum as LPM in the balanced case with $K$ classes, due to the linear interpolation property of $W^\lambda\_{(a,b)}$**, as stated in *Lines 301–305* of the revised manuscript. This result implies that LPM$_{\lambda}$-MS also satisfies the NC properties; consequently, as in the imbalanced case of LPM (without Mixup; $K$-classes), minority collapse occurs, and oversampling of mixed labels demonstrably mitigates it. The full details and proof are provided in *Appendix C.4* of the revised manuscript. This content will be incorporated into the main paper. These revisions do not affect the logical development, experiments, or the demonstrated effectiveness of the proposed method MS; rather, they further strengthen the theoretical verification.
>
> ---
>
> ### Q1. Clarify whether $K^2$ class equivalence of MS is purely theoretical or affects actual classifier behavior
> Building upon our response to W1, we additionally demonstrate that **the properties and behaviors observed in the LPM$_{\lambda}$ remain unchanged even when MS is applied, thereby theoretically confirming that BMLS remains effective in settings that incorporate MS.** Furthermore, our empirical validation shows that BMLS+MS substantially improves the model’s accuracy on minority classes, as reported in *Table 1* of *Section 5.1*. A deeper analysis of how MS influences the actual classifier behavior to further alleviate minority collapse under imbalanced settings involves the Mixup loss formulation, which lies outside the primary scope of our work—focused mainly on the sampler—and is therefore deferred to future work.
>
> ---
>
> ### W2. Assumption Robustness about the uniformity in BMLS
> To guarantee perfectly uniform pair sampling for mixed labels, **we pre-define all label pairs before sampling.** For example, suppose the dataset contains 6 samples from class 0, 4 samples from class 1, and 2 samples from class 2. In this case, we construct exactly four pairs for each mixed label (0,1), (0,2), and (1,2). Therefore, every 0-th class sample is used at least once, and two of them are randomly used one additional time, resulting in a fully specified set of sample–label pairs. By sampling from this pre-constructed set, we ensure perfectly uniform pair sampling. Empirically, as shown in the rightmost heatmap of Figure 2(a), the mixed-label frequencies across epochs in BMLS exhibit a standard deviation close to zero. The deviation is not exactly zero because, for fair comparison, we match the total number of training samples by randomly sampling a small number of additional pairs corresponding to the remainder. For instance, in CIFAR10-LT with imbalance factor 100, the training set size is 12,406, which is not divisible by the mixed-label size of 45. In this case, each of the 45 label pairs is generated 275 times, and we additionally sample 31 extra pairs to match the dataset size. Therefore, if we ignore this dataset-size constraint required for fair comparison, the assumption of perfectly uniform pair sampling holds even under mini-batch-based stochastic training.
>
> ---
>
> ### Q2. The sensitivity of mixed-label uniformity
> We acknowledge that the structure of the original manuscript may have made it difficult to clearly highlight the *mixed-label frequency control experiment*, which directly evaluates the sensitivity of mixed-label uniformity. To address this, we have fully separated the theoretical analysis from the empirical validation in the revised version. The answer to the reviewer’s question can now be found in *Section 5.1, Epoch-wise Label Imbalance*. Briefly, our results show that the model is indeed sensitive to mixed-label uniformity: as the degree of perfectly uniform mixed-label sampling deteriorates, the model’s performance in imbalanced settings degrades correspondingly, as demonstrated in *Figure 3*.

---

> ### Author Response · Authors · 2025-11-25
> **Part 2 for W3 and Q3**
>
> ### W3. Scalability
> We appreciate the reviewer’s comments regarding the empirical limitations. However, we would like to clarify a minor point: scalability is not an issue in the theoretical setting. This is because one can simply equalize the number of samples for all mixed labels through controlled sampling. In contrast, as discussed in *Section 7, Scalability*, the limitation arises only in empirical settings due to the constraint of matching the total number of training samples for fair comparison.
> Nonetheless, we agree with the reviewer that a solution is required when the number of classes becomes very large. As a naive remedy, we proposed increasing the number of training samples without enforcing the fair-comparison constraint, as introduced in *Section 7*. To empirically assess the effectiveness of this approach, we conducted simple experiments on ImageNet-LT using an $N$-times expanded training set.
>
> | Method | $N$ | many           | median         | few            | all            |
> | ------ | --- | -------------- | -------------- | -------------- | -------------- |
> | BMLS   | x1  | $59.03_{0.89}$ | $45.87_{1.01}$ | $24.86_{0.92}$ | $47.54_{0.94}$ |
> |        | x2  | $59.65_{0.89}$ | $46.67_{0.09}$ | $25.27_{0.20}$ | $48.20_{0.08}$ |
> |        | x4  | $60.54_{0.68}$ | $48.32_{0.49}$ | $28.81_{0.20}$ | $49.76_{0.48}$ |
>
> where $N= \times 2$  indicates that the model was trained on twice as many samples as in the original training set. As shown in our experimental results, simply increasing the number of training samples helps alleviate the scalability issue. However, as noted in [A1], excessive oversampling may introduce overfitting; thus, we expect that an appropriate trade-off—left for future work—will enable us to effectively address the scalability limitation.
>
> ---
>
> ### Q3. Elaborating the scalability problem
> **BMLS.** As proven in *Proposition 1*, BMLS mitigates minority collapse by increasing the number of samples in minority classes up to the level of those in majority classes. However, when the number of classes becomes extremely large—for example, in iNaturalist2018 we used 437,513 images, whereas the number of mixed labels becomes $K^2 = 66{,}292{,}164$ with $K = 8{,}142$—such balancing becomes infeasible, inevitably resulting in epoch-wise label imbalance. Consequently, the minority collapse mitigation effect of BMLS is weakened, leading to degraded performance.
>
> **MS.** In general, effective classifier training requires a sufficient number of training samples; otherwise, underfitting occurs. While this is not an issue when the number of classes is small, in datasets such as iNaturalist2018 the number of samples available to train each mixed-class vector becomes at most one. This leads to severe underfitting and, accordingly, performance degradation—a phenomenon also observed in our ablation study with the $K^2$ classifier. (See *Table 4* in the revised manuscript.)
>
> Taken together, the ideal condition is that **every mixed label has at least one sample pair and that the MS classifier receives enough training samples to avoid underfitting.** Since it is impossible to train with an unlimited number of samples, this may be considered as a computational constraint; attempting to resolve it under a fixed sample budget may even induce representation saturation. However, this issue is not a fundamental limitation of pair-level balancing itself. We expect that integration with other Mixup-based methods may address this problem under realistic constraints.
>
> [A1] Fang et al., Exploring deep neural networks via layer-peeled model: Minority collapse in imbalanced training, PNAS 2021

---

> > ### Author Response · Authors · 2025-12-03
> > **Summary of the Discussion**
> >
> > Added or modified in revision: **Lines 301-305** and **Appendix C.4** for **W1**
> >
> > | Comment                                                                                                           | Discussed                                                                                                                                                                                                                                                                                                                                       |
> > | ----------------------------------------------------------------------------------------------------------------- | ----------------------------------------------------------------------------------------------------------------------------------------------------------------------------------------------------------------------------------------------------------------------------------------------------------------------------------------------- |
> > | **W1.** Theoretical limitations about the linear dependency of mixed-class weights                                | By incorporating the reviewer’s point into the proof of **Theorem 3** in **Appendix C.4**, we address the concern by proving that the MS classifier still converges to a simplex ETF even in the presence of linear dependency.                                                                                                                 |
> > | **Q1.** Clarify whether $K^2$ class equivalence of MS is purely theoretical or affects actual classifier behavior | From the conclusion of **W1**, we demonstrate that the properties and behaviors observed in the LPM remain unchanged even when MS is applied, thereby theoretically confirming that BMLS remains effective in settings that incorporate MS. In addition, we empirically verify the actual behavior of MS through experiments (see **Table 1**). |
> > | **W2.** Assumption robustness about the uniformity in BMLS                                                        | We address the reviewer’s concern by emphasizing that uniformity is preserved through the use of pre-defined mixed-label pairs, and by providing a detailed explanation with concrete examples to illustrate how this ensures uniform sampling.                                                                                                 |
> > | **Q2.** The sensitivity of mixed-label uniformity                                                                 | We address this concern by reiterating the finding from our epoch-wise label control experiment: the model is highly sensitive to mixed-label uniformity. This directly resolves the reviewer’s point, as shown in **Figure 3.**                                                                                                                |
> > | **W3.** Scalability                                                                                               | We emphasize that the scalability issue arises from empirical constraints, not from any theoretical limitation. Furthermore, we demonstrate through experiments that increasing the number of training samples effectively alleviates this scalability issue, thereby resolving the reviewer’s concern.                                         |
> > | **Q3.** Elaborating the scalability problem                                                                       | We elaborate on the scalability problem by detailing the limitations of BMLS and MS individually and presenting the ideal conditions under which the proposed methods operate most effectively. As a result, we provide a clearer and more comprehensive explanation of the scalability concern.                                                |

---

### Author Response · Authors · 2025-11-25

### Global Response to All Reviewers
We appreciate the reviewers’ thoughtful and constructive comments, which have significantly improved our work. In this revision, we have incorporated additional experiments and theoretical proofs, resulting in a more thorough empirical and theoretical validation of our proposed methods, as well as demonstrating stronger minority-collapse mitigation compared with other strong baselines.

The additional results in the revised manuscript directly address the main concerns about the clarity of our main claims and practical benefits.
* **Improving *Representation*:** to elaborate our logic structure and connection between theorems and proposed methods, we summarize the logical flow at the end of introduction and modify the following contents accordingly (a1Sm)
* **Clarify the omitted proofs (Lemma 2 and Proposition 1):** we clarify the omitted proofs of Lemma 2 and Proposition 1 to resolve the concerns about the theoretical contribution's solidity. (ucBY)
* **New theoretical analysis (Theorem 3):** As the theoretical analysis on the $K^2$ class equivalence of MS, we demonstrate that the properties and behaviors observed in the LPM$\_{\lambda}$ remain unchanged even when MS is applied, thereby theoretically confirming that BMLS remains effective in settings that incorporate MS. (GkCW)
* **Additional ablation study (Table 4, 16-17):** we perform an ablation study using a $K^2$ classifier to disentangle the effect of increased classifier capacity from that of MS and show that MS consistently outperforms it across all evaluation axes. (ucbY)
* **Additional baselines (Table 5, 14-15):** we conducted additional experiments comparing our approach with a fixed ETF classifier and Remix, and show that our proposed methods successfully integrate with these baselines by improving the model performance on minority classes. (ucbY)
* **An Emprical Study on Mixup alpha $\alpha$:** we clarified the sensitivity of our proposed methods to the mixup alpha $\alpha$ and demonstrate that for any positive choice of $\alpha$, incorporating our proposed method consistently outperforms both the random sampler and CAS. (ucbY)
---
### Acknowledged Contributions
Reviewers explicitly recognized the novelty, theoretical rigor, and empirical effectiveness of our work:

1. **Novel perspective on minority collapse in Mixup**
    “introduces a new perspective — that pair-level imbalance (in mixed labels) is the true driver of collapse under Mixup” (GkCW),
    “addresses a pertinent issue in imbalanced learning by investigating the often-overlooked factor of mixed-label balance” (ucbY).

2. **Theoretical analysis via Neural Collapse and Layer-Peeled Model**
    “theoretical analysis seems rigorous and well-motivated” (GkCW),
    “grounded in the Layer-Peeled Model and Neural Collapse, provides a structured lens” (ucbY).

3. **Empirical validation and performance improvement**
    “experiments align well with theory and are conducted carefully across multiple benchmarks” (GkCW),
    “achieves improved performance compared to existing approaches” (a1Sm),
    “yields performance gains, particularly for minority classes” (ucbY).

4. **Clarity and presentation quality**
    “well-organized paper with clear motivation–theory–experiment flow” (GkCW).
---
### Summary of the main claim and the logical flow
In this work, we extend the analytical model, Layer-Peeled Model (LPM), to the setting of Mixup under imbalanced learning and, by applying the previously established proof sketch, we prove that **minority collapse still occurs under Mixup when the frequency of mixed labels is not balanced** (*Theorem 1*). Building on this result, we analyze existing class-balanced samplers and empirically identify **epoch-wise label imbalance** as a critical factor that degrades model performance in imbalanced learning. To address this issue, we propose **BMLS**, and we validate its effectiveness in mitigating the minority collapse of Mixup both theoretically (*Proposition 1*) and empirically (*Figures 2–4*).

Furthermore, motivated by the insight that **treating mixed labels as singleton labels could further enhance minority-collapse mitigation**, we introduce **MS**. Before applying MS together with BMLS, we theoretically verify that **the properties and behaviors observed in the LPM remain unchanged even when the Mixed-Singleton classifier is applied** (*Theorem 2*), thereby confirming that **BMLS remains effective in settings that incorporate MS**.

Finally, applying MS on top of BMLS, we observe consistent performance improvements—particularly higher accuracy on minority classes—thereby empirically validating the effectiveness of MS (*Table 1*).

In the remainder of the rebuttal, we provide point-by-point responses to all reviewers.

---

> ### Author Response · Authors · 2025-11-26
> **An Executive Summary of Mathematical Statements (1/2)**
>
> To reduce the burden on reviewers during the short discussion and rebuttal period, we provide below a concise summary of the theorems and proofs covered in this paper, as well as the main theoretical contribution of our work.
>
> ### Section 4.2. Balancing mixed labels mitigate the minority collapse of Mixup
>
> 1. (**Eq. 5**) Define a Layer define a Layer Peeled Model with Mixup (LPM$\_{\lambda}$)
> 2. (**Eq. 11** in *Appendix B*) SDP relaxation of the non-convex LPM$\_{\lambda}$.
> 	* This procedure is beneficial to prove the following theorems by transforming a quadraticallly constrained quadratic progem into a semidefinite program, which generally provides approximate solutions with bounded gaps for deep networks.
> 3. (**Lemma 1.**) Define a minimizer of last-layer features and class vectors as **Eq. 6**
> 	* From **Lemma 1**, we prove that this minimizer becomes a minimizer of Eq. 5 and all the solutions of **Eq. 5** are in the form of **Eq. 6** when the equality is satisfied.
> 	* In other words, by showing that **Eq. 6** yields the minimum of **Eq. 5** as a direct consequence of **Lemma 1**, we provide a theoretical rationale for estimating LPM$_{\lambda}$ via the SDP relaxation.
> 4. (Proof of **Lemma 1** in *Appendix C.1*)
> 	* (Lines 878-906) Define a minimizer of **Eq. 6** with constraints and assume the global minimum of **Eq. 11** as $L\_{0}$ (**Eq. 12**)
> 	* (Lines 908-919) Show that the global minimum of **Eq. 11**  becomes that of **Eq. 5** and (Lines 920-946) prove it by the contradiction.
> 5. (**Theorem 1.**) Minority collapse occurs in LPM$_{\lambda}$.
> 	* By showing that, as the imbalanced ratio diverges to positive infinity, the distances between the class vectors of minority classes converge to zero, we prove that minority collapse also occurs in the Mixup setting under imbalanced learning.
> 6. (Proof of **Theorem 1** in *Appendix C.3*)
> 	* (Lines 1087-1105) Define the optimization program about the limit case: only majority classes (**Eq. 16**)
> 	* (Lines 1107-1121) Introduce **Lemma 2**, which shows that the global minimum of **Eq. 16** is determined solely by constant terms and does not depend on either the last-layer features or the class vectors. (the proof of **Lemma 2** is deferred below)
> 	* (Lines 1124-1138) Separate the objective function into two parts: major(A) and minor classes(B)
> 	* (Lines 1139-1161) Define each minimizer of the optmization program of A (Eq. 17) and B (Eq. 18)
> 	* (Lines 1162-1170) Define their minimizers by **Lemma 2**
> 	* (Lines 1171-1206) Building on the minimizers, prove **Theorem 1** by the contradiction, showing that the negation of Theorem 1 leads to a false statement.
> 7. (Proof of **Lemma 2** in *Appendix C.3*)
> 	* (Lines 1208-1245) Define useful constants and equations and prove that **Eq. 16** can be bounded only from the constants by consider three cases buliding on the result of **Lemma 3** (the proof of **Lemma 3** is deferred below)
> 	* (Lines 1246-1292) Case 1: $K\_A=1$ (there exists only one majority class)
> 	* (Lines 1293-1331) Case 2: $K\_A > 1$ and $C=C(K\_A, K\_R, E\_H, E\_W) < 1$ where $C$ is the constant depending on $K\_A, K\_R, E\_H$, and $E\_W$.
> 	* (Lines 1332-1401) Case 3: $K\_A > 1$ and $C \geq 1$
> 	* (Lines 1402) By proving Eq. 16 can be bounded in all three cases, complete the proof
> 8. (Proof of **Lemma 3** in *Appendix C.3*)
> 	* (Lines 1404-1544) Derive the lower bound of **Eq. 16** by defining useful constants and using Caushy-Schwarz inequality and Jensen's inequality.
> 9. (**Proposition 1.**) Balancing mixed labels mitigates minority collapse in LPM$\_{\lambda}$
> 	* First, define LPM$\_{\lambda}$ in imbalanced case where the number of samples in minority classes are increased through oversampling. (**Eq. 7**)
> 	* By showing that oversampling minority classes reduces **Eq. 7** to the balanced case (**Eq. 5**) considered in **Lemma 1**, we demonstrate that balancing mixed labels mitigates minority collapse in LPM$_{\lambda}$.
> 10. (Proof of **Proposition 1** in *Appendix C.2*)
> 	* **Proposition 1** follows from the same argument used in **Lemma 1**, with the only difference being that the analysis is separated into a majority part and a minority part according to the sample counts of each class.

---

> ### Author Response · Authors · 2025-11-26
> **An Executive Summary of Mathematical Statements (2/2)**
>
> ### Section 4.3. Enhancing minority collapse mitigation via Singleton Interpretation
> 1. (**Eq. 9**) Define a Layer define a Layer Peeled Model with Mixup and MS classifier (LPM$\_{\lambda}$-MS)
> 2. (**Theorem 3** in *Appendix C.4*.) Prove that LPM$\_{\lambda}$-MS has the same global minimum with that of the LPM in balanced case where the number of classes is $K$ due to the linear interpolation property of $W^\lambda\_{(a,b)}$. As a result, the LPM$\_{\lambda}$-MS also satisfies NC properties.
> 3. (Proof of **Theorem 3** in *Appendix C.4*)
> 	* (Lines 1581-1631) Define a symmetric quadratic function to directly determine the global minimum of **Eq. 9**
> 	* (Lines 1632-1660) Show that minimizing **Eq. 9** is equivalent to minimize the symmetric quadratic function (**Eq. 38**) and define the lower bound of it (**Eq. 42**)
> 	* (Lines 1661-1697) Check the conditions that reduce **Eq. 42** to equality. In this procedure, we get the same result of the balanced case where the number of classes is $K$.
> 	* (Lines 1698-1702) Applying **Lemma 4**, we have a minimizer satisfies **Theorem 3**. As a result, the global minimum of **Eq. 9** and **Eq. 35** is the unique form for the minimizers
> 4. (**Lemma 4** in *Appendix C.4*) Prove that when **Eq. 39** reduces to equality, classifier becomes a $K$-simplex ETF while class means being aligned to each respecitve class vector.
> 5. (Proof of **Lemma 4** in *Appendix C.4*)
> 	* (Lines 1702-1719) Based on the result of **Theorem 3** and constraints of **Eq. 9**, demonstrate that last-layer features and their repective class means are aligned.
> 	* (Lines 1720-1741) Based on the above result, prove that classifier becomes a $K$-simplex ETF
> 6. (**Theorem 2.**) Prove that the properties and behaviors observed in the LPM remain unchanged even when the Mixed-Singleton classifier is applied, thereby confirming that BMLS remains effective in settings that incorporate MS.
>
> ---
>
> ### Remark of our theoretical contribution
>
> Our theoretical contribution does not lie in proposing a novel proof sketch or theoretical framework. Rather, it lies in extending previously established results—those proven through Neural Collapse and the Layer-Peeled Model—to the Mixup setting under imbalanced learning, thereby analyzing a novel aspect of the problem (*Could the frequency balance of mixed labels be a critical factor in minority collapse?*) that has not been previously explored. Through this extension, we identify a new issue (*minority collapse arises from the frequency imbalance of mixed labels*) (**Lemma 1** + **Theorem 1**) and then propose a theoretically validated method (*BMLS*) (**Proposition 1**) to address it. Moreover, even when applying an intuitively motivated approach (*MS* classifier) to BMLS, we further provide theoretical verification of BMLS's effectiveness (**Theorem 2**). Overall, our results reveal that *BMLS is grounded in solid theoretical justification* and, consequently, *exhibits superior minority-collapse mitigation compared with other samplers in empirical evaluations*.

---

### Author Response · Authors · 2025-12-03
**Rebuttal Summary for AC**

Thank you for your efforts in coordinating the review process. Below is a concise summary of the reviews and our rebuttal

---
### Before rebuttal

All reviewers(`GkCW`, `a1Sm`, `ucbY`) acknowledged **the empirical validation and performance improvement of our proposed methods**. `GkCW` and `ucbY` also recognize that we tackle **the overlooked minority collapse in Mixup problem** and highlight that our paper **theoretically well grounded via NC and LPM**.

---
### During rebuttal

`GkCW` and `a1Sm` did not respond before the OpenReview issue.

`ucbY` has responded, but some concerns about SOTA positioning and Theory-practice alignment remain.

Our key responses are summarized below.

**Reviewer** `GkCW`
* **W1 (Theoretical limitation to $K^2$)**: We incorporate the reviewer’s point into the proof of **Theorem 3** in **Appendix C.4**.
* **W2 (Assumption robustness about the uniformity in BMLS)**: We emphasize that uniformity is preserved through the use of pre-defined mixed-label pairs.
* **W3 (Scalability)**: Additional experiments for scalability in the official comment.
* **Q1 (Clarify $K^2$ class equivalence of MS)**: In addition to the theoretical rationale from the response to **W1**, we reiterate the finding from experiments in **Table 1**: MS affects actual classifier behavior while increasing the accuracy on minority classes.
* **Q2 (The sensitivity of mixed-label uniformity)**: We reiterate the finding from our epoch-wise label control experiment: the model is highly sensitive to mixed-label uniformity in **Figure 3**
* **Q3 (Elaborating the scalability problem)**: We provide a clearer and more comprehensive explanation of the scalability concern.

**Reviewer** `a1Sm`
* **W1, W3, Q1, and Q3 (Clarity of the claims and Appropriateness of the theoretical justification**: We have reorganized the logical flow and made explicit the theoretical foundations supporting each step. (See **Lines 50-77, 132-136, 147-150, 251-256, 320-323**)
* **W2 & Q2 (Detailed criteria and descriptions of the evaluation results)**: the information about them was already included in the original manuscript. We resolve this concern by adding a clear statement about them.

**Reviewer** `ucbY`

Based on the reviewer’s response, we conclude that the concerns related to **W2-1 & Q2**, **W3 & Q4**, and **W4 & Q5** have been adequately addressed.
* **W2-1 & Q2**: The reviewer said *"the missing proofs are now provided and Eq. 11 is rigorous within the LPM abstraction"*, indicating that the reviewer acknowledged that their concern regarding this point has been resolved.
* **W3 & Q4** and **W4 & Q5**: The reviewer did not raise any additional or remaining concerns regarding these points.
* **W1, Q1, and R1 (Insufficient and Outdated Empirical Comparisons)**: Results added in **Table 2**
* **W2-2, W2-3, Q3, and R2 (Theory-practice misalignment)**: We provided a justified rationale with multiple previously validated references demonstrating why LPM and SDP relaxations serve as effective analytical tools for understanding deep networks.

---
### Closing remark

Since covering all reviewer requests resulted in substantial additions to the revised manuscript—including approximately 13 pages of new theorems and proofs, as well as about 4 pages of three types of additional experiments—we propose the following guidelines to help reduce the burden on the AC during the final decision process and to facilitate an efficient understanding of the reviews and rebuttal/discussion conducted for this paper.

1. Read the **global comment by Authors** (located directly below this comment). To maintain fairness, we did not modify this comment so that it remains exactly as it was prior to the OpenReview issue.
2. "*An Executive Summary of Mathematical Statements*" does not need to be read in its entirety; however, we kindly ask that you read the **Remark of our theoretical contribution** at the end of that comment.
3. Refer to the **Summary of the Discussion** provided at the end of each reviewer’s comment. If further clarification or a more profound understanding is needed, we would appreciate it if you could consult the corresponding author's response.

In addition, we highly recommend checking **Table 2**. As shown in Table 2, we demonstrate that our proposed method (**BMLS**) and classifier (**MS**) integrate seamlessly with other mixup-based methods and improve their performance. These experimental results suggest that **BMLS and MS have the potential to serve as an effective sampler and classifier, facilitating the development of new state-of-the-art methods.**

We would like once again to express our sincere appreciation for the AC’s efforts.

---

### Note · Authors · 2026-05-01

I have read and agree with the venue's withdrawal policy on behalf of myself and my co-authors.

---

### Meta-Review · Area_Chair_x1sf · 2025-12-31

**Summary:**

This paper investigates "minority collapse" in imbalanced learning under Mixup, proposing that pair-level label imbalance is a key driver of performance degradation. The authors introduce a sampler (BMLS) and a classifier (MS) grounded in Neural Collapse (NC) and the Layer-Peeled Model (LPM).

The authors provided an exceptionally high-volume rebuttal, including new proofs and additional experiments on ImageNet-LT and DBN-mix. While the novelty of the "pair-level imbalance" perspective is acknowledged, the paper suffers from two major flaws:
1.  The reliance on simplified LPM and SDP relaxations is viewed by the reviewers as insufficient to explain the behavior of non-convex deep networks trained with cross-entropy.
2.  The comparison with SOTA methods remains "indirect" (using cross-paper reported numbers), which is insufficient in a field where results are highly sensitive to specific training "recipes."

**Reviewer Concerns:**

### Concerns Addressed by the Rebuttal
*   **Missing Proofs (ucbY/a1Sm):** The authors successfully provided the proofs for Lemma 2 and Proposition 1, which were previously cited without verification.
*   **Classifier Capacity (ucbY):** The authors conducted an ablation study showing that the Mixed-Singleton (MS) classifier outperforms a standard $K^2$ capacity classifier, suggesting the benefit isn't just due to increased parameters.
*   **Minority Metrics (a1Sm):** The authors added specific "few-category" accuracy results, which were missing in the original submission.

### Outstanding Concerns
*   Reviewer ucbY correctly points out that comparing against recent methods (DBN-mix, Remix) using numbers from original papers rather than a unified re-implementation makes the "SOTA-competitive" claims unreliable.
*   The reviewers remain unconvinced that the presence of NC in the simplified LPM translates to evidence of effectiveness in standard deep networks. This "gap" makes the theoretical contribution feel decoupled from the empirical reality of the proposed method.
*   Despite reorganization, the paper still struggles to define whether the NC properties are a *motivation* for the method or a *theoretical guarantee* of its success, leading to concerns of circular reasoning.

**Reviewer Scores:**

Reviewers are more likely to keep their score unchanged.

---

### Decision · Program_Chairs · 2026-01-26

Reject